# Time and Zonally-varying Atmospheric Waveguides - Climatologies and Connections to Quasi-Stationary Waves

Rachel H. White[1] and Lualawi Mareshet Admasu[1]

[1]Department of Earth, Ocean and Atmospheric Sciences, University of British Columbia, Vancouver, BC, Canada

**Correspondence:** Rachel H. White (rwhite@eoas.ubc.ca)

**Abstract.** Atmospheric waveguides have been linked to amplified quasi-stationary Rossby waves and thus to extreme weather events in the mid-latitudes. Here, we compare different methods of calculating time- and spatially-varying waveguides, including different ways of separating the waveguides (background flow) from waves. We compare waveguides from potential vorticity (PV) gradients ('PV-waveguides') with barotropic waveguides based on what is known as the stationary wavenumber, or $K_S$ ('KS-waveguides'), which is calculated from the zonal wind. The PV-waveguides use a PV-rolling-zonalization method to calculate the background flow. Conversely, the background flow for the KS-waveguides is calculated using time- and zonal-filtering. We isolate the impacts of the background flow methodology from the different waveguide definitions by calculating PV-waveguides using the same background flow calculation method as the KS-waveguides. There are notable differences between the waveguides identified by the two definitions. KS-waveguides are more frequent in summer than in winter, whilst PV-waveguides are more frequent in winter, regardless of the method of background flow calculation. Composites of days with high spatially averaged waveguide strength over particular regions show distinct differences between the two waveguide definitions. Strong KS-waveguides in some regions are associated with a double jet structure, consistent with previous research; this structure is rarely present for strong PV-waveguides. The presence of high geopotential heights occurs with the double jet anomaly, consistent with atmospheric blocking creating the KS-waveguide conditions through the influence on local zonal winds, highlighting that this methodology does not sufficiently separate nonlinear perturbations (i.e. blocking) from the background flow (i.e. waveguides). Significant positive correlations exist between local waveguide strength and the amplitude of quasi-stationary waves; these correlations are stronger, and more widespread, for PV-waveguides than for KS-waveguides, and strongest when the rolling-zonalization background flow method is used. This study adds further caution against using KS-waveguides on time- and/or zonally- varying scales, and we recommend rolling-zonalized PV-waveguides for the study of waveguides and their connections to quasi-stationary atmospheric waves.

## 1 Introduction

The circulation associated with large scale atmospheric Rossby waves (Rossby, 1939) can have a strong influence on the weather we experience at the Earth's surface, particularly in the extra-tropics. Indeed, high-amplitude Rossby waves can be associated with extreme weather (e.g. Bosart et al., 1996; Screen and Simmonds, 2014; Chen et al., 2015; Hoskins and Woollings, 2015; Wirth et al., 2018; White et al., 2022), including temperature extremes (e.g. Marengo et al., 2002; Teng et al., 2013;

Parker et al., 2014; McKinnon et al., 2016; Petoukhov et al., 2016; Wolf et al., 2018; Röthlisberger et al., 2019; Jiménez-Esteve et al., 2022), as well as extreme precipitation and related flooding (e.g. Blackburn et al., 2008; Schubert et al., 2011; Hirata and Grimm, 2016; Petoukhov et al., 2016; Vries and Jan, 2021). The connection to extremes is often particularly strong for waves that become quasi-stationary, i.e. have near-zero phase speed, and thus exhibit circulation anomalies that persist over one region for many days (Wolf et al., 2018; Röthlisberger et al., 2019; Ali et al., 2021). The mechanisms underlying this quasi-stationary and amplifying behaviour are not yet fully understood (e.g. Screen and Simmonds, 2013; White et al., 2022; Fei and White, 2023), and there is particular uncertainty around how such mechanisms might be impacted by anthropogenic climate change and related Arctic amplification (Francis and Vavrus, 2012; Coumou et al., 2018; Blackport and Screen, 2020). It has been suggested that jet configurations that create atmospheric 'Rossby waveguides' (e.g. Martius et al., 2010) may play an important role in the amplification of Rossby waves, and any future changes in such waves (Hoskins and Woollings, 2015; Petoukhov et al., 2016; Mann et al., 2017; Xu et al., 2021; White et al., 2022). Waveguides provide a region where wave energy is more meridionally confined, and thus wave dissipation may be low; they may therefore provide conditions for high-amplitude quasi-stationary waves (QSWs) to develop (e.g. Wolf et al., 2022). Nakamura and Huang (2018) suggest that atmospheric blocking, a non-linear amplification (and/or breaking) of Rossby waves, is related to advection of more upstream wave energy than the downstream region has the capacity to transport, akin to a traffic jam developing on a highway. Upstream waveguides could potentially play a role in this mechanism by reducing meridional dispersion of the upstream wave energy, and thus acting as the 'highway' leading to the 'traffic jam'. Improved understanding of atmospheric waveguides and their role in the existence and amplification of QSWs is therefore of great interest.

Atmospheric Rossby waveguides ocurr when the background circulation provides a preferred, typically zonally oriented, pathway for the propagation of Rossby waves. This requires strong meridional gradients in vorticity, and thus waveguides are typically associated with atmospheric jets (Hoskins and Karoly, 1981; Hoskins and Ambrizzi, 1993; Martius et al., 2010; Manola et al., 2013; Giannakaki and Martius, 2016; Branstator and Teng, 2017; Wirth et al., 2018). There are two methods of defining atmospheric waveguides from the background flow (Wirth et al., 2018). One looks for strong horizontal gradients in (isentropic) potential vorticity (PV) or in $ln(PV)$ (e.g. Martius et al., 2010), hereafter referred to as 'PV-waveguides'. This method identifies regions where meridional displacements can lead to the strongest perturbations in PV, i.e. the strongest Rossby waves. The other waveguide definition uses a barotropic 'stationary wavenumber' ($K_S$), or 'refractive index' of the background flow (Hoskins and Karoly, 1981), and the particular spatial distributions of $K_S$ that are theorized to create waveguide conditions (Hoskins and Karoly, 1981; Hoskins and Ambrizzi, 1993; Ambrizzi, 1994), hereafter 'KS-waveguides'. This method identifies regions where Rossby wave energy may be more strongly meridionally confined, and therefore may have a higher probability of high amplitude waves; however, the connection between waveguides and waves is less direct than for PV-waveguides. Recent studies have suggested that the KS-waveguide definition may provide useful insights into the behaviour of QSWs (e.g., Petoukhov et al., 2013). The validity of KS-waveguide theory on the timescales relevant for extreme weather events has recently been brought into question (see, e.g., Wirth, 2020; Wirth and Polster, 2021), including the lack of validity of the underlying assumptions (limitations articulated clearly in the original papers). The theory has, however, previously provided qualitatively useful insights into the behaviour of waves in both idealized simulations and with realistic flow conditions

(Hoskins and Karoly, 1981; Hoskins and Ambrizzi, 1993; Hsu and Lin, 1992; Hoskins and Woollings, 2015; White et al., 2017).

The theory of atmospheric waveguides has been used to understand the propagation of Rossby waves on a range of timescales, from climatological or seasonal means (Hoskins and Ambrizzi, 1993; Ding and Wang, 2005; Branstator, 2014) to waves associated with extreme events with durations between one week to one month (Petoukhov et al., 2013). One of the key requirements of waveguide theory, regardless of the method of defining waveguides, is that there is a clear separation between 'waves' and the 'background flow' on which the waves propagate (i.e., the flow which may or may not create waveguide conditions). Without a clear separation, much care must be taken in the interpretation of results (Andrews, 1985; Held et al., 2002), especially with the non-linear flows often associated with extreme events (White et al., 2022). A relatively common approach for the separation of waves and background flow is to take the longitudinal (zonal) mean as the background flow, and waves as deviations from this zonal mean (e.g. Hoskins and Karoly, 1981; Petoukhov et al., 2013, 2016). An alternative approach is to take a time mean circulation as the background flow, in which climatological or seasonal means are considered as the background flow (e.g. Hoskins and Ambrizzi, 1993). This works well for understanding climatological pathways of waves; however, when using waveguides to understand extreme events, which by definition occur for a limited period, or for using waveguides as potential sources of predictability (e.g. White et al., 2022), it may be helpful to understand how a time-varying background flow affects the probability of high-amplitude QSWs developing. Further complicating this separation of waves and background flow, waves can themselves feedback onto and impact the 'background' zonal wind (Limpasuvan and Hartmann, 2000; Lorenz and Hartmann, 2003).

In this work, the two waveguides datasets we analyse use different ways of defining the background flow: the KS-waveguides are calculated using a Butterworth time-filter and a Fourier zonal-filter. KS-waveguides on time-filtered zonal mean data have been used in several recent papers on extreme events (Petoukhov et al., 2016; Kornhuber et al., 2016; Rousi et al., 2022), and this work extends this to zonally asymmetric data. Wirth and Polster (2021) have, however, shown that isolated high amplitude non-linear perturbations can themselves *create* zonal mean conditions with the appearance of a waveguide. This argument can also apply to zonally-filtered data. To avoid this issue, Polster and Wirth (2023) develop a method of calculating a 'locally-zonalized' flow for detection of the PV-waveguides on isentropic surfaces, an extension of the zonally symmetric zonalization method (Nakamura and Zhu, 2010; Nakamura and Solomon, 2011; Methven and Berrisford, 2015). The zonalization method cannot be applied directly to the zonal wind data required for the KS-waveguides. Calculating KS-waveguides on zonalized background flow would therefore require inverting zonalized PV to produce zonalized zonal winds; this PV inversion on daily data would be a complex task. Given that daily zonal wind on upper tropospheric pressure levels are available directly from many CMIP6 climate model simulations, KS-waveguides on time- and zonally-filtered zonal winds are easier to calculate for future climates than PV-waveguides, or KS-waveguides on zonalized data. The interpolation of daily pressure level data onto the isentropic surfaces required for the PV-waveguides could potentially introduce significant numerical noise, adding further complexity to the PV-waveguide calculation for CMIP data. There are, however, some potentially critical limitations of the KS-waveguide theory, as discussed by Wirth (2020) and Wirth and Polster (2021), who recommend the PV-waveguide approach instead. It is therefore important to compare these two waveguide approaches, to determine whether KS-waveguides

can provide clear information about the true 'waveguidability' of the atmosphere. To separate the impacts of the different waveguide definitions from the impacts of the different background flow definitions, we additionally calculate PV-waveguides using the same background flow methodology as the KS-waveguides, i.e. a Butterworth time-filter and Fourier zonal-filter.

In this paper we present an algorithm to objectively define KS-waveguides on zonally- and time-varying flow. The waveguide dataset produced by this algorithm is then compared to PV-waveguides calculated following the method of Polster and Wirth (2023), as well as PV-waveguides calculated using the Polster and Wirth (2023) waveguide definition, but with the background flow calculated using the same time- and zonal- filters as the KS-waveguides. One limitation of the KS-waveguides, articulated by Wirth (2020), is when the theory is interpreted in a binary manner: either a waveguide exists and waves are 100% trapped within the waveguide, or there is no waveguide and thus 0% wave confinement; in reality a range of 'waveguidability' exists, depending on the strength of the meridional gradients within the jet. Motivated by this, in this study we extend the binary approach, using the stationary wavenumber to define a metric of 'waveguide strength', allowing a continuous range of 'waveguidability' for the KS-waveguides once a waveguide is present. We show maps of climatological waveguide frequency for all waveguide definitions, the atmospheric circulation conditions typically associated with strong local waveguides, and the associations between strong waveguides and quasi-stationary atmospheric waves. Our results ultimately strengthen existing concerns about the use of KS-waveguides, and we conclude with recommending rolling-zonalized PV-waveguides for future waveguide studies.

## 2  Data and pre-processing

ERA5 re-analysis data (Hersbach et al., 2020) are used in this study as an approximation of the observed state of the atmosphere for the calculation of waveguides and QSWs. Temperature, and zonal and meridional winds from 1980-2022 are downloaded at 6-hourly temporal resolution, before being averaged into daily means. The data are regridded to a regular latitude-longitude grid using the Climate Data Store API (Hersbach et al., 2023). The resolution of these grids are $1 \times 1\,^\circ$ for KS-waveguides and $1.5 \times 1.5\,^\circ$ for PV-waveguides; in Section 5.1 we show that the correlations between waveguides and QSWs are not sensitive to the underlying resolution of the data. We use horizontal winds on the $300hPa$ pressure level for both the KS-waveguides and the QSWs, as in previous studies (Hoskins and Karoly, 1981; Hoskins and Ambrizzi, 1993; Petoukhov et al., 2013; Wolf et al., 2018). This is slightly higher than the equivalent barotropic level of $425hPa$ estimated by Held et al. (1985), and so the KS-waveguides are also calculated at $500hPa$, with the correlation with QSWs shown in Section 5.1. In most of this work, PV-waveguides are defined on $330K$ for winter (DJF for the Northern Hemisphere, NH; JJA for the Southern Hemisphere, SH; following Polster and Wirth, 2023) and on $345K$ for summer (JJA for NH; DJF for SH); Section 5.1 shows the sensitivity to this choice for the QSW correlations.

### 2.1  Background flow

As discussed in the introduction, filtering of the atmospheric circulation fields is required to separate the waves from the slowly-varying background flow that may act as a waveguide. This problem is non-trivial, and there is no one single definition

of a slowly-varying background flow. In this work, in addition to the two different waveguide definitions, we also explore the impact of two different methods of calculating a background flow.

For the PV-waveguides we follow the local zonalization approach of Polster and Wirth (2023). Zonalization is a method of straightening out the wavy PV contours, essentially zonally smoothing data by conserving PV whilst creating a zonally symmetric PV profile with values descending monotonically from north to south (Nakamura and Zhu, 2010). Polster and Wirth (2023) extended this methodology to produce a zonally asymmetric background flow by performing a rolling zonalization on data from a limited longitude range. In this work we use the code provided by Polster (2023), with a $60°$ longitude window; the sensitivity of the QSW correlations to this longitudinal window is shown in Section 5.1. Further details of the zonalization method can be found in Polster and Wirth (2023).

For the KS-waveguides we follow Branstator (1983), and calculate background flow conditions by filtering the zonal wind using a fast Fourier Transform, and retaining only zonal wavenumbers $k = 0$, 1 and 2. Using these wavenumbers provides some spatial scale separation from the QSWs, discussed in the following section, which retain only zonal wavenumbers 4-15. Sensitivity tests in which zonal wavenumber 3 is also retained when calculating the waveguides shows no sensitivity of the main conclusions to this choice (results are shown in Section 5.1). As noted by Wirth and Polster (2021), calculation of the zonal mean does not remove all the effects of non-linear perturbations such as blocking events, and similar arguments can be made for the Fourier filtering applied here. For the KS-waveguides a time filter is also applied to further remove the effects of transient waves from the background flow. A 15-day low-pass Butterworth filter is used, with 15 days chosen to be compatible with the time filtering commonly used for QSWs (e.g. Wolf et al., 2018; Röthlisberger et al., 2019). As with the zonal filtering, this does not necessarily remove all traces of non-linear perturbations such as blocking, even if they do occur on timescales shorter than that of the filter.

To allow a separation of the impacts of the different background flow separation methods from the different waveguide definitions, we also calculate PV-waveguides from PV that, instead of being smoothed with the rolling zonalization method, has been smoothed using the same zonal Fourier filter and Butterworth time filter as the KS-waveguides. For simplicity, we refer to this method as the 'Fourier filter', in contrast to the 'rolling zonalization' method.

## 2.2 Quasi-stationary waves

Quasi-stationary waves can lead to extreme weather, and thus the potential connections to atmospheric waveguides are worth exploring. Here, a method to extract the envelope amplitude of QSWs is used, following Wolf et al. (2018). This method uses a temporal filter on meridional wind data to isolate quasi-stationary anomalies, applies a fast Fourier transform in the zonal direction to select wavenumbers of synoptic-scales, and then uses a Hilbert transform to calculate the amplitude of the wave envelope of these quasi-stationary synoptic-scale waves, following Zimin et al. (2003). Specifically, we use a 15-day low-pass Butterworth filter on daily mean $300hPa$ meridional wind data, $v$. A smoothed daily climatology is then calculated using a Savitzky-Golay filter of order 1, with a 51-day window, and anomalies from this climatology, $v'$, are calculated. Whilst previous studies have used a latitude-dependent wavelength filter (e.g. Wolf et al., 2018), here a fixed wavenumber range of wavenumbers 4-15 is used at all latitudes to maintain a similar separation from the background flow at each latitude. Wolf and

Wirth (2015) note that the semi-geostrophic nature of Rossby waves often results in the Hilbert transform separating a single Rossby wavepacket into fragments; for simplicity, a semi-geostrophic transformation is not applied here — when looking at QSWs, either composites are calculated, or the envelope field is averaged over 20° longitude, reducing the impacts of the fragmentation.

In this work, both the KS-waveguides and the QSWs employ the same 15-day low-pass filter; the KS-waveguides and waves are therefore separated only by their spatial scale, with waveguides using $k \leq 2$, and the QSWs $4 \leq k \leq 15$. Key results are repeated with the QSWs defined as $6 \leq k \leq 15$ to increase the degree of separation (see Section 5.1).

## 3 Waveguide identification

To identify PV-waveguides, we analyse isentropic horizontal gradients of the natural logarithm of the smoothed (either rolling zonalization or Fourier filtered) Ertel PV ($\|\nabla ln(PV)\|$). The zonalization and PV gradients are all calculated using the code of Polster (2023). To study the frequency of PV-waveguides we use threshold values of $\|\nabla ln(PV)\|$, including the threshold of $1.2 \times 10^{-6} m^{-1}$ introduced by Polster and Wirth (2023).

To identify KS-waveguides, we follow the definition of Hoskins and Karoly (1981) and Hoskins and Ambrizzi (1993), which is based on the linearized barotropic PV equation, with the assumption that meridional gradients in PV will dominate in the background flow. This approach first calculates a diagnostic called the *stationary wavenumber*, $K_S$ and then defines a waveguide for wavenumber $k$ as a region of finite $K_S > k$ bounded to both the north and south by 'turning points' (TPs) or 'turning latitudes' where $K_S = k$. The stationary wavenumber is calculated as:

$$K_S = a(\beta_M/U_M)^{1/2} \tag{1}$$

where $a$ is the radius of the Earth, $\beta_M$ is the meridional gradient of absolute vorticity, transformed onto a Mercator projection, and $U_M$ is the Mercator zonal wind, defined as $U_M = \frac{U}{cos\phi}$, where $\phi$ is latitude. Here, $U$ is the temporally and zonally filtered zonal wind, as described in Section 2.1. Following Hoskins and Karoly (1981), and making use of the identities for differentiation on the Mercator projection given in Appendix A, $\beta_M$ can be calculated as:

$$\beta_M = \frac{2\Omega cos^2\phi}{a} - \frac{cos\phi}{a^2} \frac{\partial}{\partial \phi} \frac{1}{cos\phi} \frac{\partial}{\partial \phi} (U_M cos^2\phi) \tag{2}$$

where $\Omega$ is the rotation rate of Earth. Once $K_S$ has been calculated on the temporally and zonally filtered $U$, an algorithm identifies waveguides for waves of integer wavenumber $k$ from 4 to 9. At each longitude and time, the algorithm looks for regions where two TPs exist, such that $k < K_S < \infty$ between the TPs and $K_S < k$ immediately outside of the TPs. This is illustrated in Fig. 1, in which $K_S$ for a particular day and longitude is shown, with TPs identified for wavenumbers $k = 4, 5, 6$, and the hatching illustrating the waveguides for these different $k$. For each $k$ in $4 \leq k \leq 9$, if a waveguide is detected based on $K_S$, a minimum threshold is applied to the following: the zonal wind $U$ within the waveguide; the maximum waveguide strength, $\max(w_S) = \max(K_S - k)$ across all latitudes within the waveguide; and the waveguide width, $w_w$ (i.e. the meridional distance between the TPs). These criteria limit detection to substantial waveguides within westerly flow. The following

thresholds are used, with waveguides removed from the dataset if they do not meet all criteria: $U \geq 0.5m/s$ throughout the waveguide; $w_s \geq 1$; $w_w \geq 5°$ latitude; results are found to be insensitive to changes in these thresholds of up to 50%. The two hemispheres are treated separately, and the algorithm only looks for TPs between 20-85° N/S. This therefore includes much of the sub-tropical jets, but reduces the influence of the subjective choice of the latitude at which to start identifying waveguides on waveguide frequency and strength in the extra-tropics.

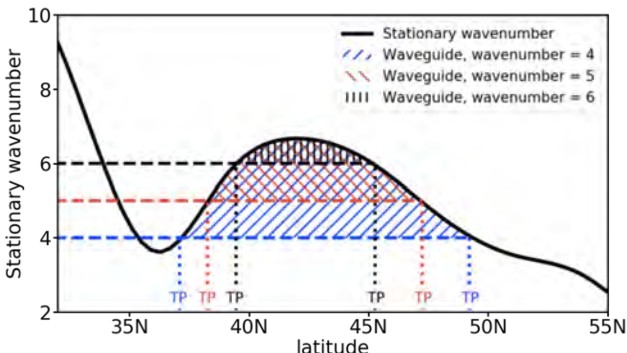

**Figure 1.** Illustration of the KS-waveguide detection algorithm for an example day and longitude. The black thick line denotes the stationary wavenumber $K_S$ as a function of latitude. The 'turning points' are marked as 'TP' for wavenumbers $k =$ 4, 5 and 6. Shading illustrates where KS-waveguides exist for these different wavenumbers.

## 4 Climatological waveguide statistics

To get a broad understanding of where we expect to see waveguides, Fig. 2 shows the climatological median value of $K_S$ (left side), the climatological mean of $\|\nabla ln(PV)\|$ in the time- and Fourier-filtered PV data (centre), and the climatological mean of $\|\nabla ln(PV)\|$ in the rolling-zonalization PV data (right side). For $K_S$ the median is used instead of the mean to avoid noise from near-infinite values; nan values are set to 0 before the median is taken. Despite the non-linearities in $K_S$, the amplitude and spatial distribution seen in Fig. 2 are consistent with $K_S$ calculated on climatological mean $U$ presented in Hoskins and Ambrizzi (1993) and Ambrizzi et al. (1995).

Figure 2 shows that $K_S$ is typically higher in summer than in winter. Conversely, the climatological values of $\|\nabla ln(PV)\|$ (Fig. 2 b, c, e and f) follow the strength of the jet more closely (regardless of the method of calculating the background flow), with typically stronger values in winter than in summer. As waveguides are associated with jets, there is a general expectation that stronger jets, which typically occur in winter (see black contours in Fig. 2), should lead to stronger waveguides. As highlighted by Manola et al. (2013), in addition to jet strength, the jet width, or rather 'narrowness' of the jet is also key for high values of $K_S$. Indeed, returning to Eq. 1, note that the magnitude of $U$ appears in the denominator of the equation for $K_S$, whilst only the second meridional gradient of $U$, i.e. $\partial^2 U/\partial\phi^2$, appears in the numerator, within the $\beta_M$ term (see Eq. 2). Idealised tests of calculating $K_S$ on linear multiples of different realistic $U$ profiles confirms that, unless the planetary vorticity

(first term on the right hand side of Eq. 2) is negligible, then if $U$ increases in strength but the latitudinal shape of the $U$ profile (i.e. the jet half-width) remains constant, $K_S$ decreases in magnitude. This result, of stronger $K_S$ in summer than winter, is also seen in previous literature of $K_S$ calculated on climatological $U$ (Hoskins and Ambrizzi, 1993; Ambrizzi et al., 1995; Hoskins and Woollings, 2015), but is in contrast to the results from the PV-waveguides.

The impact of the choice of background flow calculation method can be seen by comparing the centre and right-hand columns. There are some differences in the zonal asymmetry, but overall, the centre panel is more similar to the right-hand column than the left-hand column, indicating that differences in waveguide definition (KS vs PV gradient) dominate over differences in the background flow methodology for these climatological values. Note that seasonal shifts in latitude for the PV-waveguides occur not only because of seasonal latitudinal shifts in the jets, but also because we are evaluating the PV gradient on different isentropes in summer vs winter.

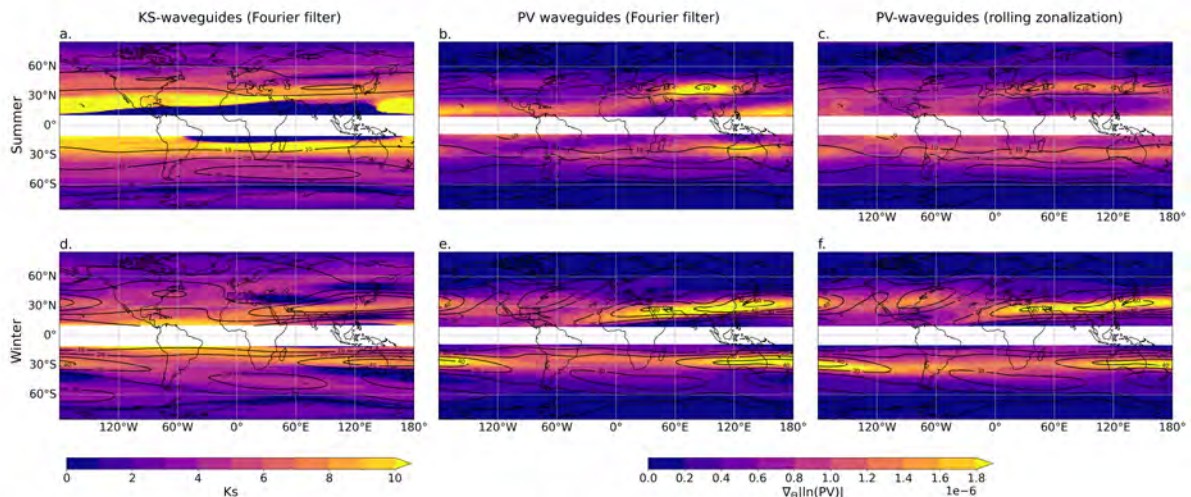

**Figure 2.** Seasonal climatologies of waveguide fields. Left: median $K_S$ (colours) and mean $U$ (black contours) at $300\ hPa$ for temporally and zonally filtered data for a. summer (JJA for NH and DJF for SH) and d. winter (DJF for NH; JJA for SH). Centre: mean isentropic $\|\nabla ln(PV)\|$ $(m^{-1})$ on Fourier-filtered PV (colours) and climatological unfiltered $300\ hPa$ $U$ (black contours) for b. summer ($345K$ isentrope) and e. winter ($330K$ isentrope). Right: mean isentropic $\|\nabla ln(PV)\|$ $(m^{-1})$ with rolling zonalization (colours) and climatological unfiltered $300\ hPa$ $U$ (black contours) for c. summer ($345K$ isentrope), and f. winter ($330K$ isentrope). Values between 10S to 10N are masked out to highlight the discontinuity in months between summer/winter in each hemisphere.

### 4.1 Waveguide frequency

The definition of KS-waveguides provides a specific binary definition of whether a waveguide exists for a given zonal wavenumber $k$: either two turning latitudes exist for a given $k$, or they do not. Conversely, for PV-waveguides, there is a continuity of gradients, with increasing waveguidability for increasing gradient strength (Wirth, 2020), and no dependence on wavenumber.

A 'frequency' of PV-waveguides can be calculated by setting a threshold value of $\|\nabla ln(PV)\|$, above which a waveguide is said to exist (e.g. Polster and Wirth, 2023), but the subsequent waveguide frequency will be highly dependent on the threshold chosen. No direct quantitative comparison of waveguide frequency between the two waveguide metrics is therefore possible. We can, however, compare the zonal distribution, and seasonal and hemispheric variability in waveguide frequency across the different waveguide definitions. Fig. 3 shows the climatological waveguide frequency (fraction of days that each gridpoint lies within a waveguide) of KS-waveguides for values of $k = 5$ and 7 (left column), and the frequency of PV-waveguides for two thresholds of $\|\nabla ln(PV)\|$ for Fourier-filtered PV-waveguides (centre column) and rolling-zonalization PV-waveguides (right column). The first threshold, $1.2 \times 10^{-6} m^{-1}$, follows Polster and Wirth (2023), and the second, $1.5 \times 10^{-6} m^{-1}$, provides a stricter definition.

There are some broad similarities in waveguide frequency distribution between the two waveguide definitions. For both KS- and PV-waveguides, high frequencies are largely constrained to the sub-tropics and mid-latitudes, as expected based on the location of the climatological jets (although the latitude of PV-waveguides is very sensitive to the isentrope selected). Waveguides tend to be more frequent over the Pacific and over Asia (NH) and the Indian Ocean (SH), consistent with the location of the strongest/narrowest jets (see black contours). There are also, however, noticeable differences between KS- and PV-waveguides, and between the PV-waveguides with different background flow methodologies. Consistent with the values of $K_S$ shown in Fig. 2, the frequency of KS-waveguides is, in many regions, higher in summer than in winter. In contrast, PV waveguides are, in almost all regions, more frequent in winter, regardless of the method of background flow calculation. The frequency of KS-waveguides is typically lower in the SH than in the NH, particularly in summer, consistent with a slightly lower $K_S$ in the SH relative to the NH (see Fig. 2). In contrast, the frequency of the rolling-zonalization PV-waveguides is more symmetric between the hemispheres. Again, this emphasises that, for the KS-waveguides, the magnitude of $U$ in the denominator of the equation for $K_S$ plays an important role, whilst for the PV-waveguides, the horizontal gradients are the only important factor. The frequency of the Fourier-filtered PV-waveguides is broadly similar to the rolling-zonalization PV waveguides, but there is more hemispheric asymmetry, particularly in summer, suggesting that the method of calculating the background flow plays some role in the enhanced waveguide frequency in the NH relative to the SH. In winter, the Fourier-filtered PV-waveguides are concentrated in the sub-tropics, and do not reach as far poleward as the Fourier-filtered PV-waveguides over the Pacific, North America and into the Atlantic. The method of calculating the background flow therefore has some influence on waveguide frequency.

Given the two waveguide definitions have large differences in the relative climatological waveguide frequency between different hemispheres and seasons, it is clear that the two waveguide methods cannot both be accurately quantifying the waveguid-ability of the atmosphere. Existing concerns over the KS-waveguide methodology suggest that the PV-waveguide climatology is likely a more accurate description of the atmosphere's waveguidability; in the following sub-sections we further explore the differences in the waveguide definitions.

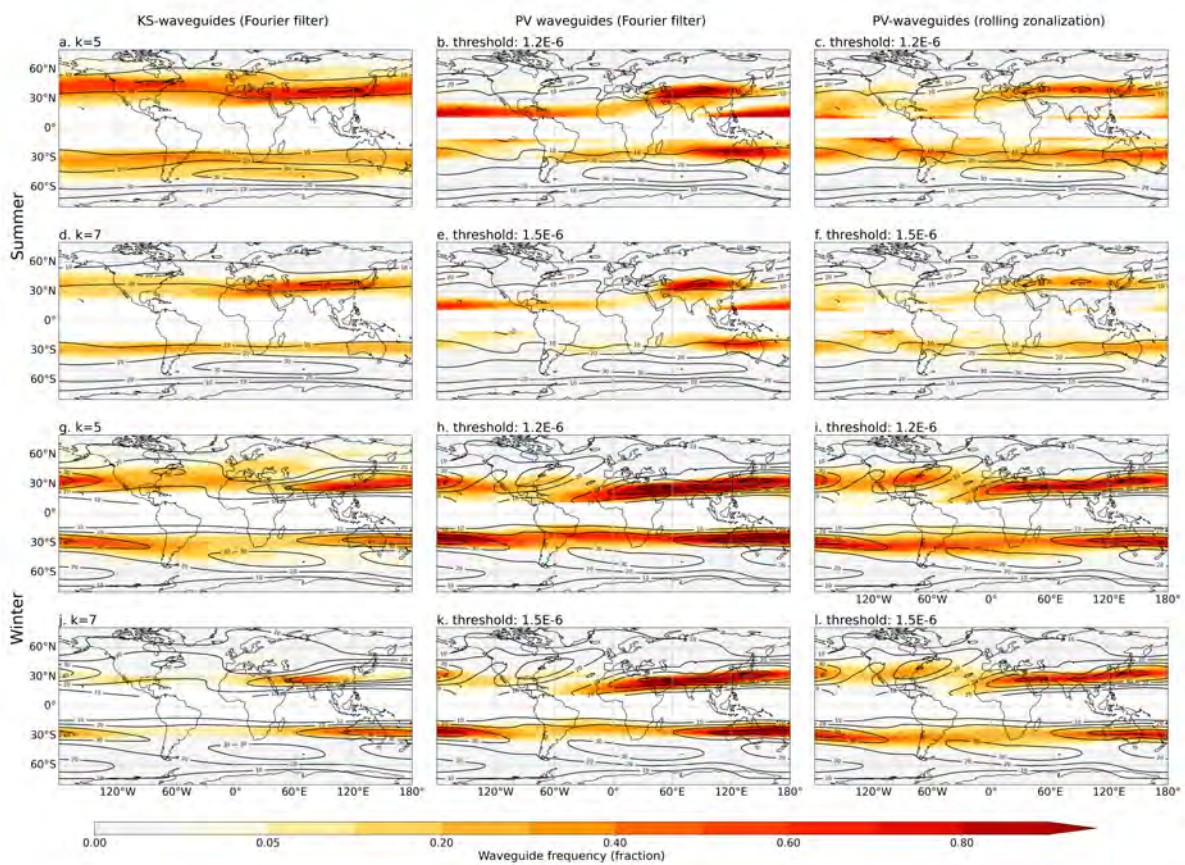

**Figure 3.** Seasonal waveguide frequency for summer (a-f) and winter (g-l), for KS-waveguides (left) at 300 $hPa$, and Fourier-filter PV-waveguides (centre) and rolling-zonalization PV-waveguides (right), on the $345K$ isentrope for summer and the $330K$ isentrope for winter. KS-waveguides are shown for zonal wavenumbers $k = 5$ (a, g) and $k = 7$ (d, j); PV-waveguides are shown for thresholds $1.2 \times 10^{-6}$ (b, c, h, i) and $1.5 \times 10^{-6}$ (e, f, k, l). Because of the dependence of the KS-waveguides on $k$ and the PV-waveguides on threshold, these frequencies cannot be directly compared between the methods. Note the non-linear progression of the color bar at the smallest frequencies. The seasonal mean zonally filtered $U$ at 300 $hPa$ is shown in black contours in the left column, with the centre and right columns showing the same seasonal climatological $U$ at 300 $hPa$, but without the zonal filter.

## 4.2 Waveguide composites

To study the conditions associated with strong waveguides, a single 'total waveguide strength', $W_S$, is defined for each day and each gridpoint. For KS-waveguides this is the sum of the individual waveguide strengths across all wavenumbers $4 \leq k \leq 9$ for which a waveguide exists, i.e. $W_S = \sum_{k=4}^{9} w_s = \sum_{k=4}^{9} (K_S - k)$. We use the sum of waveguide strengths for all values of $k$ as the waves may be a mix of different zonal wavenumbers; however, results are insensitive to using the maximum $w_S$ over $4 \leq k \leq 9$ instead. For PV-waveguides the total waveguide strength is simply the magnitude of the PV gradient, i.e.

$W_S = \|\nabla ln(PV)\|$. To avoid selecting too many days from one event, the waveguides are sub-sampled to every 5th day. Area-weighted regional averages of the sub-sampled total waveguide strengths are made over different regions, and composites are created for days for which the regionally averaged total waveguide strength exceeds the 90th percentile for that region and season. We refer to these as 'strong waveguide days'. This is approximately 70 days for each composite, all separated by at least 5 days. All anomaly fields are relative to a smoothed seasonal cycle, calculated by applying an order-1 Savitzky-Golay

filter to a daily climatology. Values are masked to $p < 0.05$ using bootstrap resampling of random days with 500 resamples. We show plots for NH summer, as this has been the focus of several previous papers studying waveguides and extreme weather, with SH summer for a hemispheric comparison. Plots for winter are provided in Appendix B.

### 4.2.1   Zonal winds

Figures 4 (NH) and 5 (SH) show the composite anomalous unfiltered zonal wind for strong waveguide days for four distinct
localized regions (panels a-l) plus a region extending around all longitudes (panels m, n and o). These composites demonstrate the anomalous background wind conditions associated with particularly strong waveguides in each region. The boxes for the regional waveguide averages are shown in dashed black lines, and two contours of the anomalous total waveguide strength are shown in magenta. The anomalous waveguide field is masked to 0 equatorward of $35°$ N/S to reduce noise in the contoured fields. As the contours chosen for the KS-waveguides and PV-waveguides have different units, no direct comparison can
be made between the magenta contours across the different waveguide definitions; however, PV-waveguides on the rolling-zonalization background flow are typically more zonally extensive than those on the Fourier-filter background flow, particularly in the NH.

Both KS-waveguides and PV-waveguides typically have some increase of zonal wind speed within the waveguide region, although this is much more distinct for the PV-waveguides (regardless of the method of background flow calculation), and
it is almost completely absent for KS-waveguides in some regions (e.g. Fig. 4d and j). This is consistent with the results of Section 4.1, where we showed that KS-waveguides are not necessarily related to the strongest jets. Days of strong localized KS-waveguides also have anomalously weak zonal winds immediately poleward of the region of strong waveguides. Such anomalies act to enhance the second derivative of zonal wind in the meridional direction ($\partial^2 U/\partial$

$phi^2$), consistent with the theory of what causes KS-waveguide conditions. This pattern of zonal wind anomalies associated
with high waveguide days shifts in latitude with the waveguide region selected (not shown). The anomalies associated with strong PV-waveguides (for both background flow methodologies) are typically a region of enhanced $U$ within the waveguide region, and reduced $U$ equatorward of this region in the NH (Fig. 4), and both equatorward and poleward in the SH (Fig. 5). The differences between the KS- and PV-waveguide composites illustrate that the days with strongest KS-waveguides in a particular region are not the same as the days with strongest PV-waveguides, and thus these two waveguide definitions are not
identifying the same waveguide conditions. The similarity between the Fourier-filtered and rolling-zonalization PV-waveguides suggests that similar, although not completely identical, days are identified by the PV-waveguide methodology, regardless of the method of calculating the background flow.

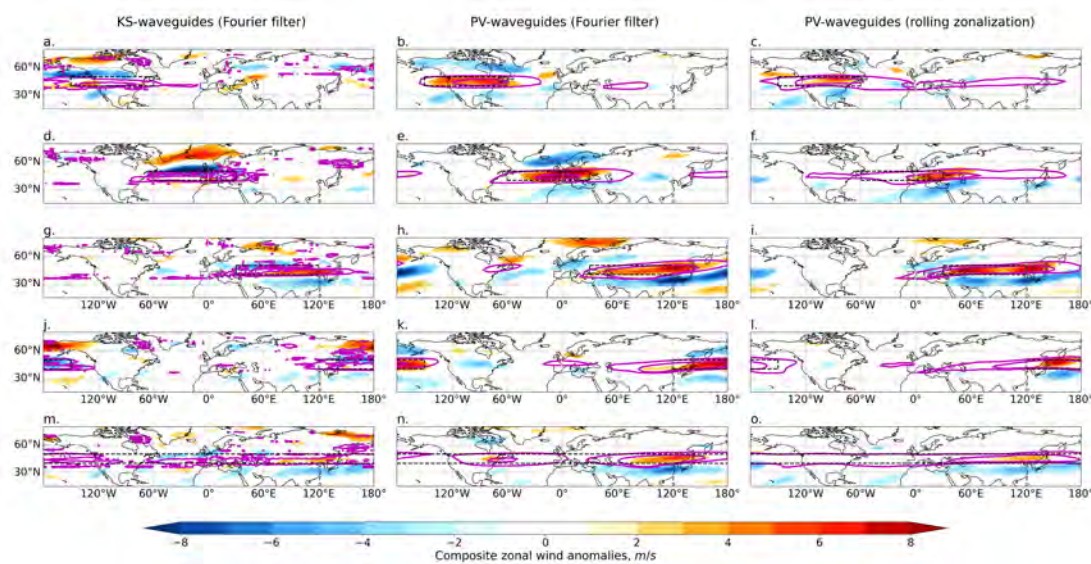

**Figure 4.** NH JJA $300hPa$ zonal wind ($U300$) composites anomalies for the 10% of days with highest total waveguide strength in JJA averaged over the black boxed region, for different regions (rows) and for KS-waveguides (left), Fourier-filtered PV-waveguides (centre) and rolling-zonalized PV-waveguides (right). Values are masked to p< 0.05 based on bootstrap resampled composites of random days. Magenta contours show composite waveguide anomalous strength, masked to zero equatorward of $35°$ N/S to focus on extratropical waveguides.

In some regions, particularly over North America (Fig. 4a), Asia (Fig. 4g, the south-east Pacific (Fig. 5a) and the south Atlantic (Fig. 5d) something of a 'double jet structure' can be seen in the KS-waveguide anomalies, with two regions of enhanced zonal winds, separated meridionally by a region of weakened zonal winds. Plots of the full zonal wind field composites for the strong waveguide days confirm that these anomalies are associated with double jets during the strong waveguide days (i.e. it is not just a feature of anomalies), with a southerly shifted mid-latitude jet and a strengthened jet at high latitudes. This result is consistent with the 'double jet' structure previously identified as associated with KS-waveguides (Petoukhov et al., 2013; Kornhuber et al., 2016; Rousi et al., 2022). This structure has been described as a strong high-latitude Arctic front jet combined with a strong and narrow subtropical jet (Mann et al., 2018). Notably, for these regionalized KS-waveguides, the double jet structure seems to be strongly localized in longitude, and there is no clear zonally symmetric double jet structure even when selecting for the zonal waveguide (Fig. 4m and Fig. 5m). Such double jet anomalies do not appear in most regions for strong PV-waveguide days, regardless of the background flow methodology; the exception is for waveguides over NH Asia (Fig. 4h and i), and upstream of the Fourier-filtered PV-waveguides over the South Atlantic (Fig. 5e. The high latitude localized $U$ anomalies on strong KS-waveguides days seen most clearly in, for example, panels 4a, d and j, are consistent with the zonal wind anomalies associated with high latitude atmospheric blocking, in which a region of high pressure leads to an acceleration of westerly winds poleward, and a deceleration equatorward. Wirth and Polster (2021) showed that anomalies associated with

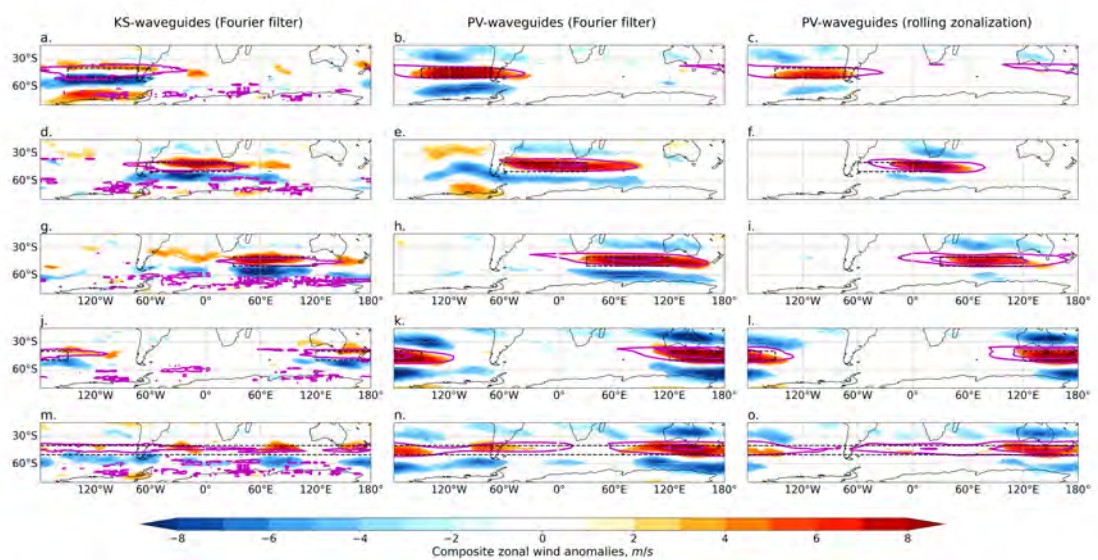

**Figure 5.** As for Fig. 4, but for SH summer (DJF). $300hPa$ zonal wind ($U300$) composites anomalies for the 10% of days with strongest DJF waveguide presence in the black boxed region, for different regions (rows) and for KS-waveguides (left), Fourier-filtered PV-waveguides (centre) and rolling-zonalized PV-waveguides (right).

idealized strong atmospheric blocking can create the spurious appearance of waveguides in the zonal mean; it is possible that a similar mechanism also occurs on more local scales, leading to the localized double jet anomalies in the KS-waveguide composites. To further explore this, we next plot composites of geopotential height.

### 4.2.2 Geopotential heights

Figs. 6 and 7 show composites of anomalous geopotential height at $500hPa$ ($Z500$) for strong KS-waveguide days (left column), strong Fourier-filtered PV-waveguides (centre) and strong rolling-zonalization PV-waveguides (right column). For KS-waveguides, in all regions except the zonally symmetric band, a region of anomalously high geopotential height can be seen poleward of the waveguide location. Such anomalies are consistent with the hypothesis that high latitude blocking creates the zonal wind anomalies seen in Figs. 4 and 5, leading to KS-waveguide conditions equatorward of the block. It is possible that the higher latitude blocking is creating local KS-waveguide conditions - blocks are known to impact the jet, and thus the subsequent movement of smaller, more transient eddies (e.g. Shutts, 1983). These results highlight that the KS-waveguide methodology is unable to effectively separate the blocking perturbation from the background flow, despite the blocks typically occurring on length scales smaller than $k = 2$, the upper bound of the spatial Fourier filter. Notably, strong PV-waveguide days (right columns of Figs. 6 and 7) do not generally show such anomalies, except for a region of high geopotential poleward and at the western edge of the waveguide region for waveguides over Asia (Fig. 6i). This indicates the improved separation

between waves and background flow for the PV-waveguides. In the PV-waveguide composites there is a broad (albeit weak) pattern of decreased geopotential height poleward of the waveguide, consistent with geostrophic balance and the enhanced

zonal winds in the waveguide region. Hemispheric wave-like patterns can be seen in some of the waveguide composites (e.g. Fig. 6b, c), suggestive of circumglobal teleconnections. The PV-waveguides calculated on time- and Fourier-filtered data (Fig. 6 and 7 centre columns) are, for most regions, very similar to those using the rolling-zonalized background data, highlighting that it is the waveguide definition itself, and not the method of separating the waves from the background flow, that is primarily responsible for the differences between the composites for the KS-waveguides and rolling-zonalization PV-waveguides. There

are, however, some regions where the background flow methodology does play more of a role, such as the South Atlantic (Fig. 7e), where a high geopotential height anomaly very similar to that seen for the KS-waveguides is found upstream and poleward of the waveguide region for Fourier-filtered PV-waveguides, but not for rolling-zonalization PV-waveguides.

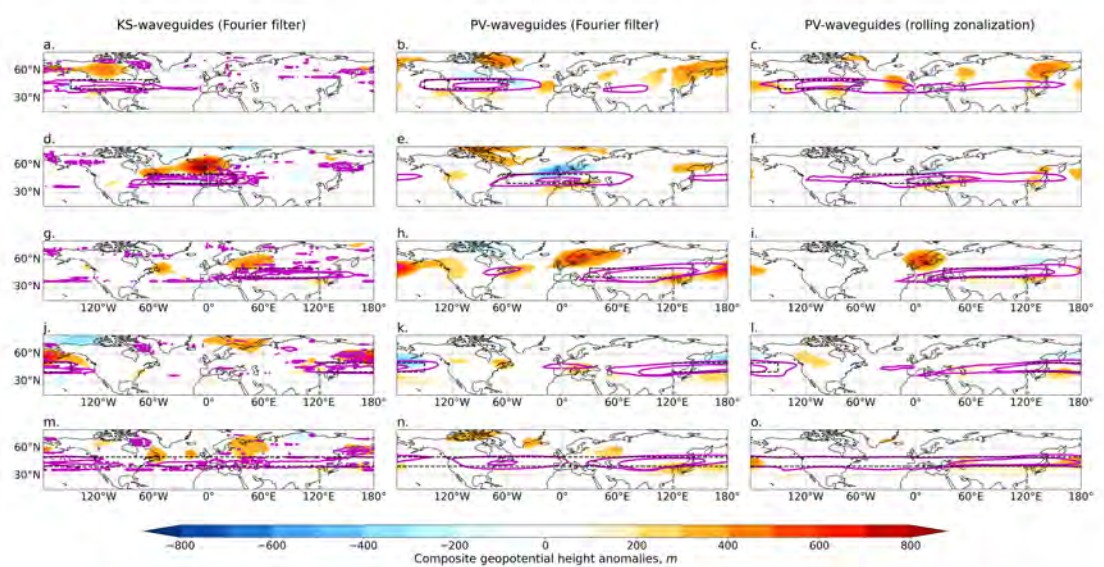

**Figure 6.** As for Fig. 4, but with geopotential height at $500hPa$ ($Z500$). NH JJA $Z500$ composites anomalies for the 10% of days with strongest JJA regionally averaged total waveguide strength in the black boxed region, for different regions (rows) and for KS-waveguides (left), Fourier-filtered PV-waveguides (centre), and rolling-zonalized PV-waveguides (right).

## 5   Waveguides and atmospheric waves

In this section we explore the relationship between waveguides and QSWs. Figures 8 and 9 show composite differences of the

QSW strength between strong waveguide days (top 10% of regional averaged total waveguide strength) and weak waveguide days (lowest 10% regional averaged total waveguide strength) for summer for the NH and SH respectively, for the same regions as in the previous figures. Here we show differences between strong and weak waveguide days to amplify the signal-

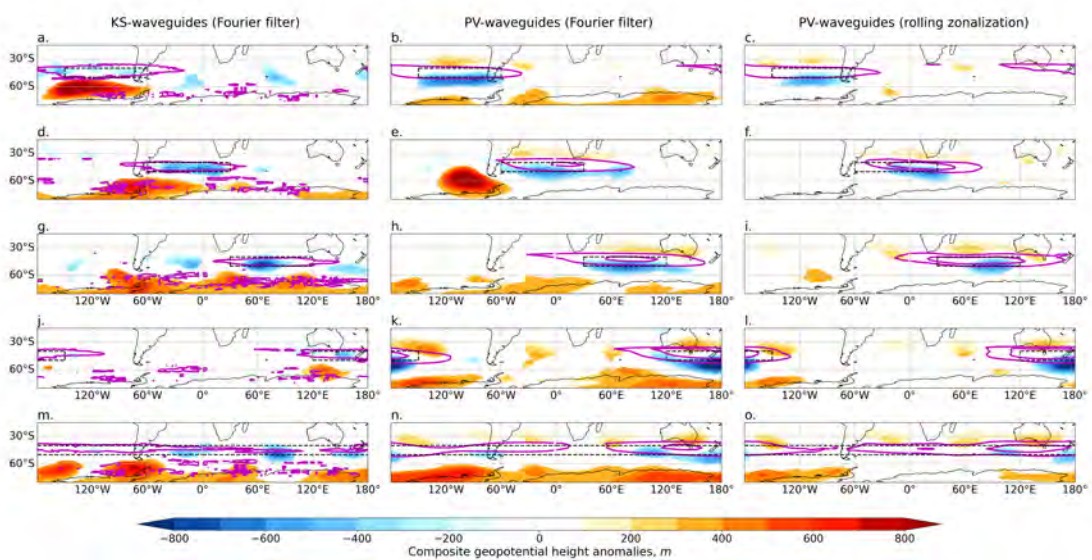

**Figure 7.** As for Fig. 6, but for SH summer (DJF). $Z500$ composites anomalies for the 10% of days with strongest DJF regionally averaged total waveguide strength in the black boxed region, for different regions (rows) and for KS-waveguides (left), Fourier-filtered PV-waveguides (centre), and rolling-zonalized PV-waveguides (right).

to-noise ratio. Increases in QSW strength are sometimes, but not always, seen within the region of strong waveguides for both KS-waveguides and PV-waveguides. For the KS-waveguides, however, many regions also show an increase in QSW strength
outside of, and predominantly poleward of, the strengthened waveguide region, sometimes stronger than the signal within the waveguide itself (e.g. Fig. 8a). This is consistent with the findings in the previous section, that high pressure anomalies consistent with atmospheric blocking are often found poleward of the enhanced waveguide, with the associated zonal wind anomalies likely helping to create waveguide conditions equatorward of the block – such atmospheric blocks would show up as positive anomalies in QSW strength. Conversely, the PV-waveguides typically show reduced QSW activity outside of
the waveguide region, for both the Fourier-filtered and rolling-zonalization PV-waveguides. For the rolling-zonalization PV-waveguides, many, but not all, regions show enhanced QSW strength co-located with the strong waveguide.

To further investigate the connection between QSWs and waveguides, Pearson correlations are calculated between total waveguide strength and co-located QSWs. Anomalies from smoothed daily climatologies were calculated to remove the impacts of any seasonal cycles on observed correlations. As a consequence of the time-filtering of the input data for the calculation
of both the waveguides and QSW envelopes, as well as intrinsic atmospheric memory, both fields exhibit auto-correlation. To reduce the impacts of this on the degrees of freedom for calculating the significance of the correlations, all fields are sub-sampled with one sample every 10 days; auto-correlation of the sub-sampled KS-waveguide dataset is under 0.1 almost everywhere.

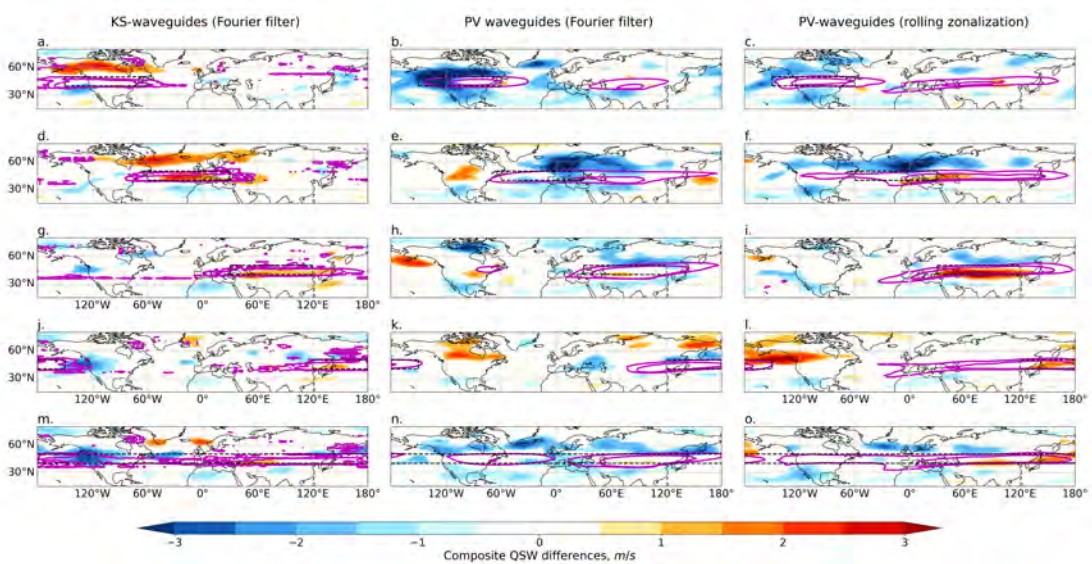

**Figure 8.** NH QSW differences between composites of the 10% of JJA days with strongest regionally averaged total waveguide strength in the black boxed region relative to composites of the 10% of JJA days with the weakest regionally averaged total waveguide strength, for different regions (rows). KS-waveguides shown on the left, Fourier-filtered PV-waveguides in the centre, and rolling-zonalized PV-waveguides on the right. Magenta contours show composite total waveguide strength differences, masked to zero equatorward of 35° N/S to focus on extratropical waveguides.

Pearson correlation coefficients are calculated over time for data in JJA and DJF separately, for QSW envelope values and total waveguide strength values, both averaged over 20° longitude by 5° latitude boxes. Correlation coefficients are shown in
Fig. 10, with hatching indicating statistical significance, where p-values have been adjusted to account for the false discovery rate of repeating multiple significance tests over space. The adjustment was performed using scipy.stats.false_discovery_control with the Benjamini-Yekutieli procedure (Benjamini and Yekutieli, 2001), as the values are not from independent tests due to spatial correlation; this is a more conservative test, and thus significance is shown for both $p < 0.1$ (diagonal hatching) and $p < 0.05$ (cross-hatching).

For both waveguide metrics, and both background flow methodologies, positive correlations are found between total waveguide strength and QSWs over much of the mid-latitudes in both hemispheres, consistent with the hypothesis that waveguides can lead to enhanced QSWs. These correlations are stronger (and more statistically significant) over much broader regions for the rolling-zonalization PV-waveguides than the KS-waveguides. The correlation strengths are more comparable between the KS-waveguides and the Fourier-filtered PV-waveguides, although the Fourier-filtered PV waveguides typically have stronger
correlations, except for NH summer (panels 10a and b). The correlations, even where statistically significant, are relatively weak, with typical correlation coefficients of $r < 0.25$; this is not unexpected - even if waveguides are indeed helping create

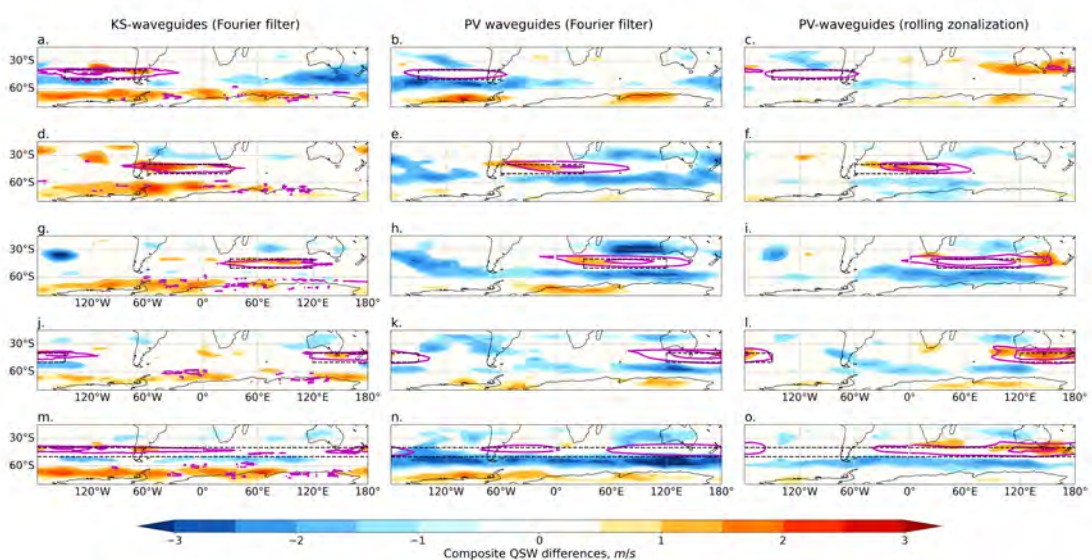

**Figure 9.** As for Fig. 8 but for SH summer (DJF). QSW differences between composites of the 10% of DJF days with strongest regionally averaged total waveguide strength in the black boxed region relative to composites of the 10% of JJA days with the weakest regionally averaged total waveguide strength, for different regions (rows). KS-waveguides shown on the left, Fourier-filtered PV-waveguides in the centre, and rolling-zonalized PV-waveguides on the right.

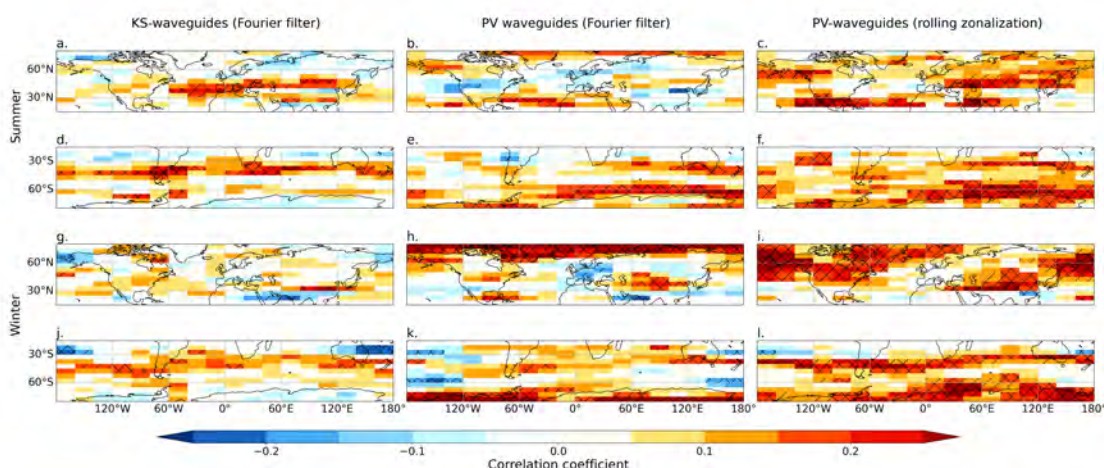

**Figure 10.** Pearson correlation coefficient between total waveguide strength and QSW strength, each averaged over $20°$ longitude $\times$ $5°$ latitude boxes for KS-waveguides (left column), Fourier-filtered PV-waveguides (centre) and rolling zonalization PV-waveguides (right column). Diagonal hatching indicates regions where the correlation is statistically significant at $p < 0.1$, and cross-hatching at $p < 0.05$.

the conditions for high amplitude QSWs, there still needs to be a source of wave energy or growth in a suitable location for the waves to become trapped within the waveguide.

The Fourier-filtered PV-waveguide correlations (centre column of Fig. 10) have a spatial pattern that more closely resembles that of the rolling-zonalization PV-waveguides (right column), than the Fourier-filtered KS-waveguides (left column), suggesting that the difference in waveguide definition, and not background flow methodology, is the dominant driver of the spatial differences between the KS- and PV-waveguides. Figure 10 does, however, illustrate differences between the two PV-waveguide datasets, with substantially weaker correlations for the Fourier-filtered PV-waveguides. This shows that the method of calculating the background flow also plays an important role in the relationship between waveguides and QSWs. Fig. 10 clearly illustrates the benefits of the rolling-zonalization PV-waveguide methodology for detecting waveguides associated with QSWs, with stronger and more widespread positive correlations than the other waveguide methods. The latitude at which the strongest correlations are found between PV-waveguides and QSWs is very sensitive to the isentrope selected, as shown in the following section, and so the spatial distribution of the positive correlations here should not be taken as indicative of which regions have strong relationships between QSWs and PV-waveguides generally, only for PV-waveguides on the 345K (summer) and 330K (winter) isentropes.

For the KS-waveguides, winter has weaker correlations between waveguides and QSWs than summer, particularly in the NH, where there is limited statistical significance in winter; conversely, the PV-waveguides, particularly with the rolling zonalization background flow, shows relatively strong correlations with QSWs in both seasons.

## 5.1 Sensitivity tests

Here we briefly explore the sensitivity of the QSW-waveguide correlations shown in Fig. 10 to some of the subjective choices made in this study, including the spatial resolution of the input data for the waveguide calculations, the surface (pressure level or isentrope) the waveguides are calculated on, the wavenumbers included in the QSW definition, and the spatial scale of the zonal filtering for the background flow. To test the impact of resolution on the detection of waveguides, the $1 \times 1°$ ERA5 data are bilinearly regridded to a $2.5 \times 2.5°$ grid using the Climate Data Operators (CDO) and the waveguide detection procedures are repeated on these data.

Fig. 11 shows the co-located QSW-waveguide correlations for various sensitivity tests for both KS- and rolling-zonalization PV-waveguides, focusing on NH summer. For both waveguide definitions, changing the resolution of the waveguide input data from $1° \times 1°$ (Figs. 11a and b) to $2.5° \times 2.5°$ (Figs. 11c and d) has little impact, suggesting that high resolution data is not required for waveguides. For KS-waveguides, the largest difference across the sensitivity tests conducted is in using $500hPa$ data for the waveguides instead of $300hPa$ (compare Fig. 11e to Fig. 11a). Using $500hPa$ data provides more positive correlations with QSWs across much of the NH. Similarly, for the PV-waveguides, there is a strong sensitivity to the isentrope on which the waveguides are calculated, from $345K$ (Fig. 11b) to $330K$ (Fig. 11f). As would be expected from consideration of the latitude at which each isentrope intersects the 2PVU contour, colder surfaces find stronger correlations further poleward. When QSWs are defined as only wavenumbers 6-15, there is little difference at low and mid- latitudes (equatorward of around $50°$N), but, for the PV-waveguides, a strong reduction in the correlation poleward of $50°$N; this may

be related to the smaller fraction of QSW power in wavenumbers $k \geq 6$ at higher latitudes (Wolf et al., 2018). Panels i. and j. illustrate the sensitivity to zonal filtering. For the KS-waveguides, we increase the wavenumber (i.e. decrease the wavelength) of the zonal filtering, allowing more spatial variation in the background flow - this leads to a very small increase in the positive correlations (compare Figs. 11a and i). For PV-waveguides we reduce the spatial variation in the background flow by conducting the rolling-zonalization procedure over $120°$ longitude regions instead of $60°$. We see a slight decrease in the strength of the spatial correlations (compare Figs. 11b and j), but the spatial pattern remains similar.

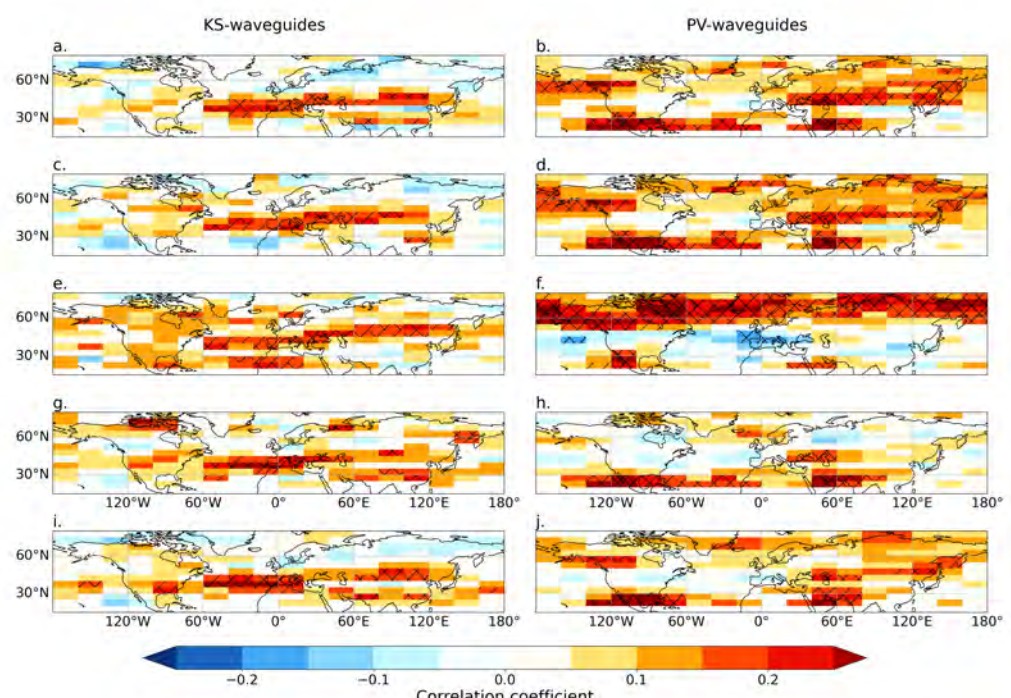

**Figure 11.** Pearson correlation coefficient between total waveguide strength and QSW strength, as in Fig. 10, but for sensitivity tests and only for NH summer. KS-waveguides (left column) and rolling-zonalization PV-waveguides (right column). Panels a. and b. are repeated from Fig. 10 panels a. and c. The second row (c. and d.) shows the sensitivity to waveguide input resolution with KS- and PV-waveguides calculated on $2.5° \times 2.5°$ data. The third row (e. and f.) shows the sensitivity to the surface on which waveguides are calculated, with KS-waveguides on $500 hPa$ and PV-waveguides on $330 K$. The fourth row (g. and h.) shows the sensitivity to QSW definition, with QSWs defined as wavenumber $6 \leq k \leq 15$. The fifth row (i. and j.) shows the sensitivity to the lengthscale of the zonal filtering, with KS-waveguides calculated using a background flow that also retains $k = 3$, and the PV-waveguides calculated on a rolling-zonalization of $120°$ longitude instead of $60°$. As in Fig. 10, diagonal hatching indicates regions where the correlation is statistically significant at $p < 0.1$, and cross-hatching at $p < 0.05$.

## 6 Discussion

In this study we compare temporally- and zonally-varying atmospheric waveguides using two definitions, KS-waveguides and PV-waveguides. We analyse climatological statistics as well as the connections between waveguides and quasi-stationary waves. The PV-waveguides are calculated using a rolling-zonalization to create the background flow conditions, as described by Polster and Wirth (2023), with waveguides then detected as the strength of the horizontal gradient of $ln(PV)$. The KS-waveguides are detected from the presence of two turning latitudes in the stationary wavenumber, $K_S$, using a detection algorithm similar to that described by Kornhuber et al. (2016), but extended to zonally varying flow. For the KS-waveguides we use a time- and wavenumber separation between the background flow and waves through a zonal Fourier transform and Butterworth time filter. To separate the impacts of the different waveguide definitions from the different background flow methodologies, we also calculate PV-waveguides using the same background flow methodology as the KS-waveguides (time- and Fourier zonal-filter). We find large differences between the two waveguide methodologies, including in composites of strong waveguide days, implying substantial differences between the two waveguide definitions on individual days. This highlights that caution is needed when detecting atmospheric waveguides, as results are likely to be sensitive to the waveguide definition. Overall, waveguide definition seems to be more important than the method of defining the background flow, but the background flow methodology also plays an important role, particularly in the connection between waveguides and QSWs.

There are substantial differences in the seasonal and hemispheric variations of the KS- and PV-waveguides, with PV-waveguides typically more frequent (for the same threshold) during winter, when jets tend to be stronger, relative to summer. Conversely, KS-waveguides are typically more frequent in summer than in winter. This can be understood by the different underlying equations of the two waveguide definitions. The strong differences we find between the two waveguide methods, combined with prior concerns over the validity of the KS-waveguides (e.g. Wirth and Polster, 2021) and the results in this study that QSWs are more strongly correlated with PV-waveguides than KS-waveguides, all suggest that the rolling-zonalization PV-waveguide definition provides a more accurate description of waveguide climatology, and the seasonal cycle of atmospheric waveguideability, and that KS-waveguides should not be used.

Through composites of 'strong waveguide days' for particular regions, the average background flow conditions associated with strong KS- and PV-waveguides are illuminated. Similar to results found in previous work on zonal mean flow (e.g. Kornhuber et al., 2016; Rousi et al., 2022), for KS-waveguides a double jet structure in the zonal wind anomalies can be seen over a number of regions, localized to the longitude region of the strong waveguide. In contrast, days with strong PV-waveguides typically do not show this double jet anomaly, instead showing a strengthened zonal wind within the waveguide region, and weaker zonal winds outside of this region. Wirth and Polster (2021) have previously cautioned that double jet anomalies in the zonal mean can be caused by non-linear high amplitude perturbations, such as those associated with atmospheric blocks, and this can lead to spurious apparent waveguide conditions. We find that, for many regions, positive Z500 and QSW anomalies are present poleward of strong KS-waveguides (see Figs. 6, 7, 8, and 9), consistent with this idea that high latitude blocks may be leading to localized waveguide conditions equatorward of the block (high latitude blocking would likely show up as enhanced QSWs in our metric). These enhanced Z500 and QSW anomalies poleward of the waveguide are not typically present

for strong PV-waveguides, even with the same background flow methodology as the KS-waveguides, further highlighting the differences between these methods of detecting atmospheric waveguides. Kornhuber et al. (2016) have shown, however, that double jets in the zonal mean can appear *prior* to amplified waves, suggesting there may still be some causality from double jets to waves; further investigation is required to fully understand this.

Both KS- and PV-waveguide strengths are positively correlated with the presence of amplified QSWs, but the strength of this connection varies with waveguide definition, and also with region, season, hemisphere, and background flow methodology (see Figs. 8, 9 and 10). PV-waveguides show generally stronger positive correlations with QSWs than KS-waveguides, particularly in the higher latitudes, consistent with recent work showing strong teleconnections along high latitude PV-waveguides (Xu et al., 2019, 2020). Note that, at high latitudes, correlations between PV-waveguides and QSWs in summer are much stronger when using the $330K$ isentrope (Fig. 11). The spatial and seasonal variations in these correlations may also be impacted by variations in QSW strength (Wolf et al., 2018). The PV-waveguides tend to have stronger correlations with QSWs than the KS-waveguides regardless of the method of calculating the background flow, but using a rolling-zonalization for background flow substantially increases the strength of the correlations. Strong PV-waveguides are typically associated with enhanced QSWs in the vicinity of the waveguide, and reduced QSW amplitude outside of this region, consistent with the hypothesis that the PV-waveguide locally enhances the probability of QSWs, leading to greater QSWs in the waveguide region, and weaker QSWs at other latitudes. KS-waveguides are also often associated with enhanced QSWs in the region of the waveguide, but also, for waveguides in many regions, with enhanced QSW activity poleward of the strong waveguide region, indicating that the KS-waveguides may not be sufficiently separated from the waves.

Correlations between co-located QSWs and total waveguide strength are substantially higher over many regions for PV-waveguides than for KS-waveguides (Fig. 10), particularly during winter and at higher latitudes. Given the result that atmospheric blocking may be creating KS-waveguide conditions, any correlations between KS-waveguides and QSWs must be interpreted cautiously. Even the co-located enhanced QSW activity for KS-waveguides, seen in Figs. 8, 9 and 10, may, in some cases, be caused by the lower-latitude cyclonic anomalies associated with higher latitude blocking of Omega, Rex or Rossby wave-breaking types (see, e.g., Woollings et al., 2018). Indeed, in the Z500 anomaly composites of strong KS-waveguide days, weak negative anomalies can be seen within the waveguide region in several regions (Figs. 6 and 7). Alternatively, the co-located positive correlations between KS-waveguides and QSWs could be indicative of a highly complex and interactive relationship between QSWs and waveguides, in which a high latitude atmospheric block leads to waveguide conditions equatorwards of the block, resulting in an increased probability of QSWs in the waveguide region. This could be consistent with the results of Ali et al. (2022), who suggest that blocks could play a role in initiating the Rossby wave packets and/or in modulating their phase to create quasi-stationary anomalies. Notably, however, the strong association with blocks mostly does not occur for PV-waveguides, particularly when the rolling-zonalization method is used to calculate the background flow.

Interestingly, the PV-waveguides from time- and Fourier-filtered background flow tend to have weaker correlations with QSWs relative to the rolling-zonalization PV-waveguides, even when a rolling-zonalization window of similar spatial scale to the zonal Fourier filter is used (compare Figs. 10b and 11j). Since the rolling-zonalization method should be able to better separate the waves from the background flow (Polster and Wirth, 2023), this enhanced correlation with better separation is

indicative that the rolling-zonalization PV-waveguides are indeed identifying background flow conditions where amplified QSWs are more likely to occur. Lag-lead studies of the co-occurrence of PV-waveguides with co-located QSWS, or using causal inference (see, e.g. Kretschmer et al., 2021) to understand causality, would be illuminating. Idealized simulations such as those performed by Segalini et al. (2024) to study the 'waveguidability' of the background flow conditions associated with strong waveguides may also help confirm the causal relationship between rolling-zonalization PV-waveguides and amplified QSWs implied by this current study.

## 7 Conclusions

Zonally-and temporally-varying KS-waveguides and PV-waveguides are identified from daily ERA5 data from 1980-2022 using two methods of calculating the background flow. The two waveguide definitions produce some broad similarities in the waveguide climatologies, but a number of notable differences, with substantial differences in hemispheric and seasonal variations in waveguide frequency. For KS-waveguides only, a double jet structure is found to be associated with strong waveguide days, particularly over the North Atlantic and European regions. Further analysis suggests this is likely at least partially related to atmospheric blocking, with blocking conditions leading to the jet anomalies that create the KS-waveguide conditions. Such an association does not occur for PV-waveguides, particularly when using the rolling-zonalization method of calculating the background flow. Positive correlations between total waveguide strength and quasi-stationary wave amplitude are found across much of the mid-latitudes; these correlations are strongest for the PV-waveguides calculated from a rolling-zonalization background flow. These correlations suggest that strong PV-waveguide conditions make amplified QSWs more likely, although causality has not been established here. Given the large differences between the two waveguide datasets, previous concerns regarding the poor separation of waves from the waveguides for the KS-waveguides (and the evidence of this presented here), and the stronger positive correlations between QSWs and PV-waveguides, we recommend that PV-waveguides on rolling-zonalized flow (Polster and Wirth, 2023) are used for detecting time-varying zonally asymmetric waveguides, particularly for studying the conditions conducive for quasi-stationary waves and related extreme weather events.

*Code and data availability.* ERA5 data were downloaded from the Copernicus Data Store (CDS):

https://cds.climate.copernicus.eu/cdsapp#!/dataset/reanalysis-era5-pressure-levels

Waveguide and QSW datasets are available for download from borealisdata.ca.

Code to reproduce the main Figures in this manuscript is available on GitHub:

https://github.com/rhwhite/SupportingInformation/blob/main/WaveguidesQSWs.ipynb

## Appendix A: Mercator Projection

$$x = a\lambda \qquad\qquad\qquad\qquad\text{(A1)}$$

$$y = aln\left[\frac{1+sin\phi}{cos\phi}\right] \tag{A2}$$

$$\frac{\partial y}{\partial \phi} = a\frac{cos\phi}{1+sin\phi}\frac{\partial}{\partial \phi}\left(\frac{1+sin\phi}{cos\phi}\right) = a\frac{cos\phi}{1+sin\phi}\left(\frac{1+sin\phi}{cos^2\phi}\right) = \frac{a}{cos\phi} \tag{A3}$$

$$\frac{\partial}{\partial y} = \frac{cos\phi}{a}\frac{\partial}{\partial \phi} \tag{A4}$$

## Appendix B: Winter composites

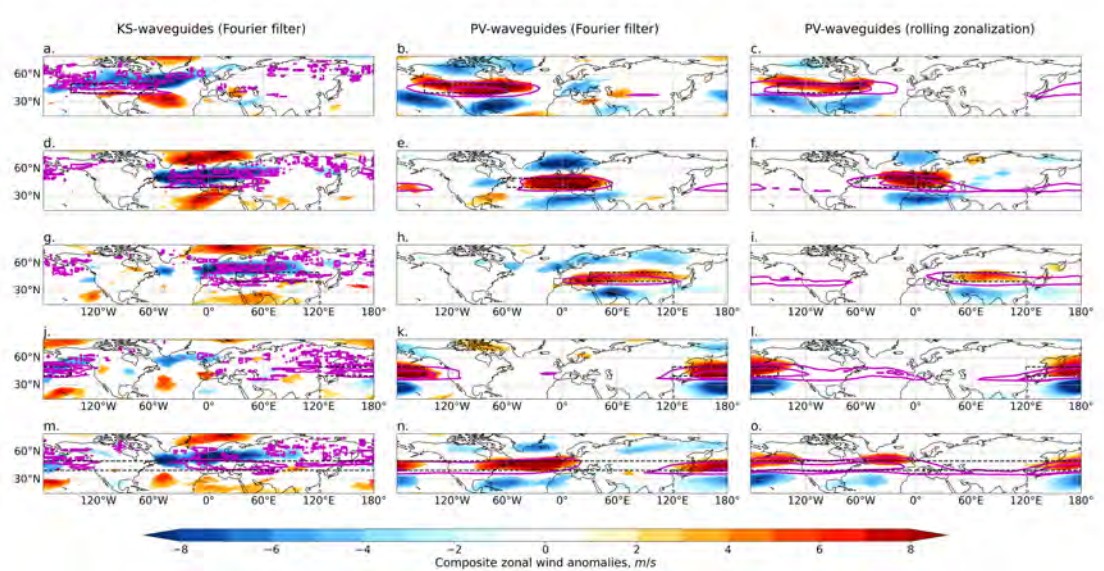

**Figure B1.** As for Fig. 4, but for NH winter (DJF). $U300$ composites anomalies for the 10% of days with highest total waveguide strength in DJF averaged over the black boxed region, for different regions (rows) and for KS-waveguides (left), Fourier-filtered PV-waveguides (centre) and rolling-zonalized PV-waveguides (right). Values are masked to $p< 0.05$ based on bootstrap resampled composites of random days. Magenta contours show composite waveguide anomalous strength, masked to zero equatorward of $35°$ N/S to focus on extratropical waveguides.

*Author contributions.* RHW designed the study, wrote and implemented the KS-waveguide algorithm, conducted the analysis, created the plots, and wrote the paper. LMA calculated the PV-waveguides and contributed to the design of the waveguide comparison, and edited the paper.

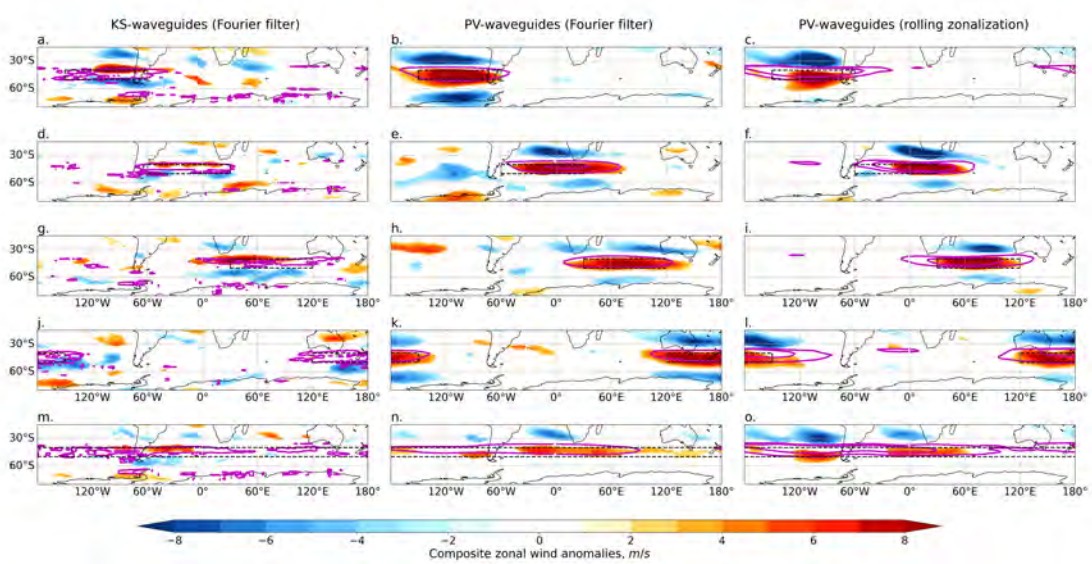

**Figure B2.** As for Fig. 5, but for SH winter (JJA). $U300$ composites anomalies for the 10% of days with strongest regionally averaged JJA total waveguide strength in the black boxed region, for different regions (rows) and for KS-waveguides (left) and PV-waveguides (right).

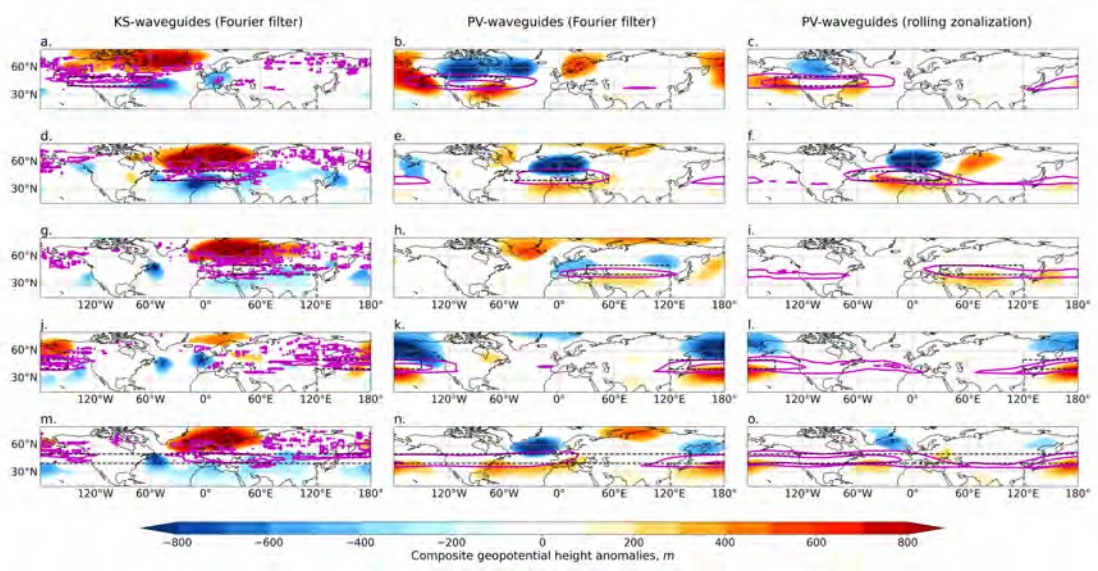

**Figure B3.** As for Fig. 6, but for NH winter (DJF). $Z500$ composites anomalies for the 10% of days with strongest regionally averaged DJF total waveguide strength in the black boxed region, for different regions (rows) and for KS-waveguides (left) and PV-waveguides (right).

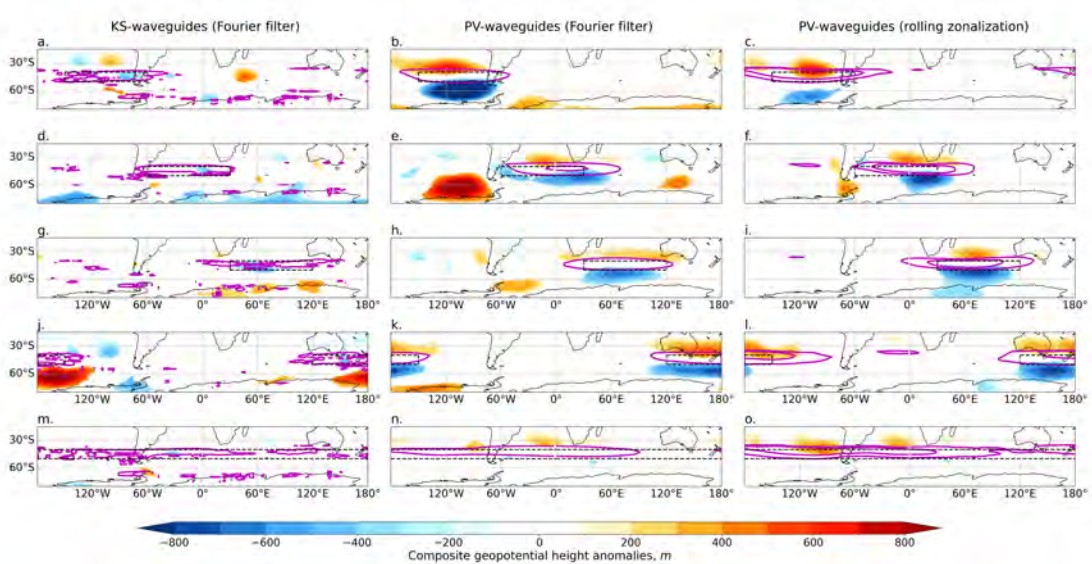

**Figure B4.** As for Fig. 7, but for SH winter (JJA). $Z500$ composites anomalies for the 10% of days with strongest regionally averaged JJA total waveguide strength in the black boxed region, for different regions (rows) and for KS-waveguides (left) and PV-waveguides (right).

520  *Competing interests.* The authors declare no competing interests

*Acknowledgements.* This research was supported in part through computational resources and services provided by Advanced Research Computing at the University of British Columbia. LMA was supported by UBC and by NSERC Discovery Grant [RGPIN-2020-05783]. The authors thank Volkmar Wirth for valuable discussions and review of this paper, Peiqiang Xu for valuable discussions, the editor and two anonymous reviewers for their helpful comments that have led to substantial improvements to this manuscript.

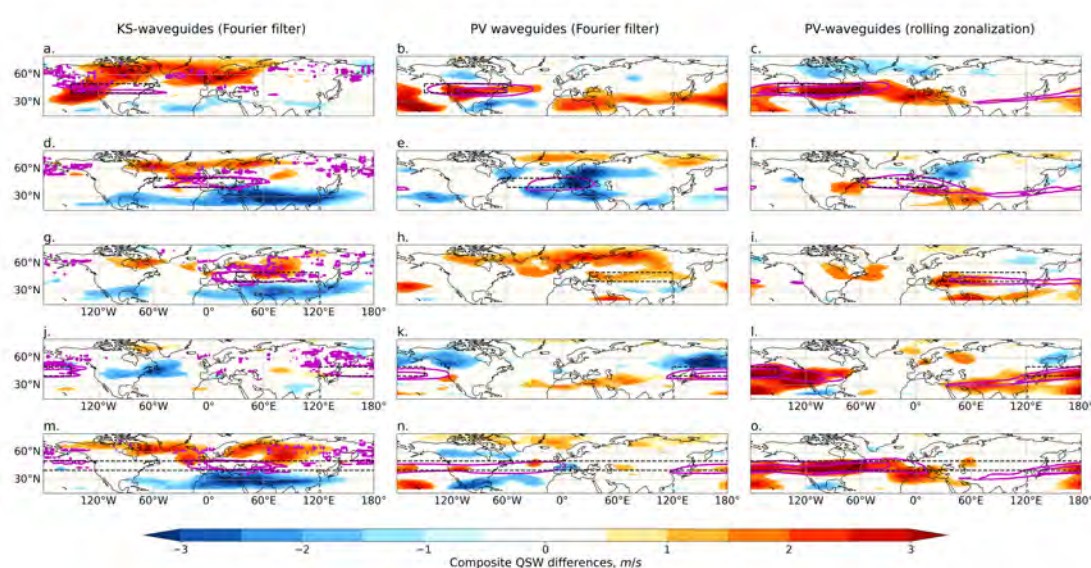

**Figure B5.** As for Fig. 8, but for NH winter (DJF). QSW composite differences between the 10% of days with strongest regionally averaged DJF total waveguide strength in the black boxed region relative to the 10% of days with weakest regional average DJF waveguides, for different regions (rows) and for KS-waveguides (left) and PV-waveguides (right). Magenta contours show composite total waveguide strength differences, masked to zero equatorward of 35° N/S to focus on extratropical waveguides.

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

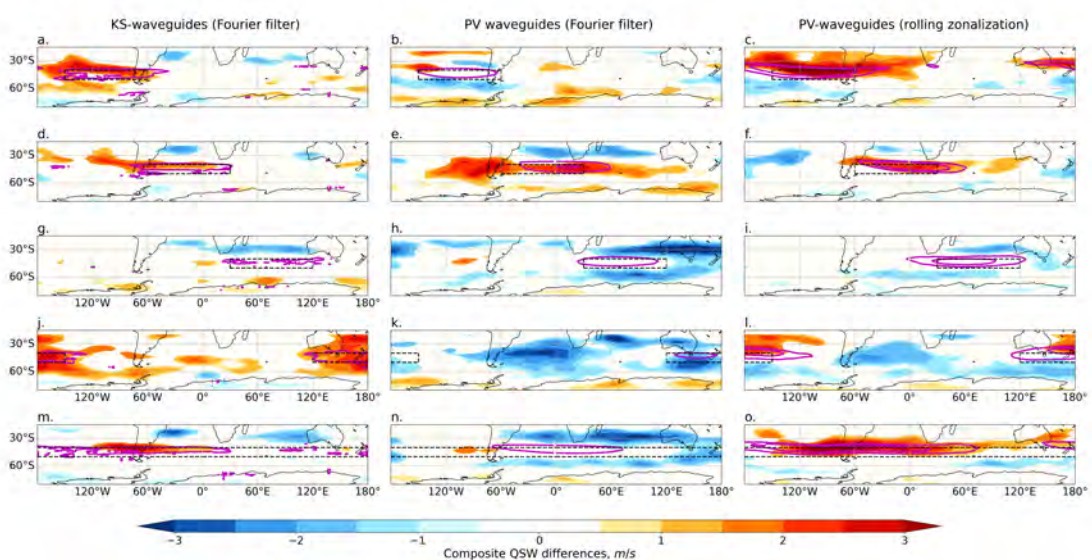

**Figure B6.** As for Fig. 9, but for SH winter (JJA). QSW composite differences between the 10% of days with strongest regionally averaged JJA total waveguide strength in the black boxed region relative to the 10% of days with weakest regional average JJA waveguides, for different regions (rows) and for KS-waveguides (left) and PV-waveguides (right). Values are masked to $p < 0.05$ based on bootstrap resampled composites of random days. Magenta contours show composite anomalous total waveguide strength.

540     Blackport, R. and Screen, J. A.: Insignificant effect of Arctic amplification on the amplitude of midlatitude atmospheric waves, Science
Advances, 6, eaay2880, https://doi.org/10.1126/sciadv.aay2880, publisher: American Association for the Advancement of Science, 2020.

Bosart, L. F., Hakim, G. J., Tyle, K. R., Bedrick, M. A., Bracken, W. E., Dickinson, M. J., and Schultz, D. M.: Large-Scale Antecedent
Conditions Associated with the 12–14 March 1993 Cyclone ("Superstorm '93") over Eastern North America, Monthly Weather Review,
124, 1865–1891, https://doi.org/10.1175/1520-0493(1996)124<1865:LSACAW>2.0.CO;2, 1996.

545     Branstator, G.: Horizontal Energy propagation in a Barotropic Atmosphere with Meridional and Zonal Structure, Journal of the Atmospheric
Sciences, 40, 1689–1708, https://doi.org/10.1175/1520-0469(1983)040<1689:HEPIAB>2.0.CO;2, 1983.

Branstator, G.: Long-Lived Response of the Midlatitude Circulation and Storm Tracks to Pulses of Tropical Heating, Journal of Climate, 27,
8809–8826, https://doi.org/10.1175/JCLI-D-14-00312.1, 2014.

Branstator, G. and Teng, H.: Tropospheric Waveguide Teleconnections and Their Seasonality, Journal of the Atmospheric Sciences, 74,
550     1513–1532, https://doi.org/10.1175/JAS-D-16-0305.1, 2017.

Chen, G., Lu, J., Burrows, D. A., and Leung, L. R.: Local finite-amplitude wave activity as an objective diagnostic of midlatitude extreme
weather, Geophysical Research Letters, 42, https://doi.org/10.1002/2015GL066959, 2015.

Coumou, D., Di Capua, G., Vavrus, S., Wang, L., and Wang, S.: The influence of Arctic amplification on mid-latitude summer circulation,
Nature Communications, 9, 2959, https://doi.org/10.1038/s41467-018-05256-8, publisher: Nature Publishing Group, 2018.

555     Ding, Q. and Wang, B.: Circumglobal Teleconnection in the Northern Hemisphere Summer, Journal of Climate, 18, 3483–3505,
https://doi.org/10.1175/JCLI3473.1, 2005.

Fei, C. and White, R. H.: Large-amplitude quasi-stationary Rossby wave events in ERA5 and the CESM2: features, precursors, and model biases in Northern Hemisphere winter, https://doi.org/10.1175/JAS-D-22-0042.1.1, section: Journal of the Atmospheric Sciences, 2023.

Francis, J. A. and Vavrus, S. J.: Evidence linking Arctic amplification to extreme weather in mid-latitudes, Geophysical Research Letters, 39, https://doi.org/10.1029/2012GL051000, _eprint: https://onlinelibrary.wiley.com/doi/pdf/10.1029/2012GL051000, 2012.

Giannakaki, P. and Martius, O.: An Object-Based Forecast Verification Tool for Synoptic-Scale Rossby Waveguides, Weather and Forecasting, 31, 937–946, https://doi.org/10.1175/WAF-D-15-0147.1, 2016.

Held, I. M., Panetta, R. L., and Pierrehumbert, R. T.: Stationary External Rossby Waves in Vertical Shear, Journal of the Atmospheric Sciences, 42, 865–883, https://doi.org/10.1175/1520-0469(1985)042<0865:SERWIV>2.0.CO;2, publisher: American Meteorological Society Section: Journal of the Atmospheric Sciences, 1985.

Held, I. M., Ting, M., and Wang, H.: Northern Winter Stationary Waves: Theory and Modeling, Journal of Climate, 15, 2125–2144, https://doi.org/10.1175/1520-0442(2002)015<2125:NWSWTA>2.0.CO;2, 2002.

Hersbach, H., Bell, B., Berrisford, P., Hirahara, S., Horányi, A., Muñoz-Sabater, J., Nicolas, J., Peubey, C., Radu, R., Schepers, D., Simmons, A., Soci, C., Abdalla, S., Abellan, X., Balsamo, G., Bechtold, P., Biavati, G., Bidlot, J., Bonavita, M., De Chiara, G., Dahlgren, P., Dee, D., Diamantakis, M., Dragani, R., Flemming, J., Forbes, R., Fuentes, M., Geer, A., Haimberger, L., Healy, S., Hogan, R. J., Hólm, E., Janisková, M., Keeley, S., Laloyaux, P., Lopez, P., Lupu, C., Radnoti, G., De Rosnay, P., Rozum, I., Vamborg, F., Villaume, S., and Thépaut, J.: The ERA5 global reanalysis, Quarterly Journal of the Royal Meteorological Society, 146, 1999–2049, https://doi.org/10.1002/qj.3803, 2020.

Hersbach, H., Bell, B., Berrisford, P., Biavati, G., Horányi, A., Muñoz Sabater, J., Nicolas, J., Peubey, C., Radu, R., Rozum, I., Schepers, D., Simmons, A., Soci, C., Dee, D., and Thépaut, J.-N.: ERA5 hourly data on pressure levels from 1940 to present, https://doi.org/10.24381/cds.bd0915c6, 2023.

Hirata, F. E. and Grimm, A. M.: The role of synoptic and intraseasonal anomalies in the life cycle of summer rainfall extremes over South America, Climate Dynamics, 46, 3041–3055, https://doi.org/10.1007/s00382-015-2751-6, 2016.

Hoskins, B. and Woollings, T.: Persistent Extratropical Regimes and Climate Extremes, Current Climate Change Reports, 1, 115–124, https://doi.org/10.1007/s40641-015-0020-8, 2015.

Hoskins, B. J. and Ambrizzi, T.: Rossby Wave Propagation on a Realistic Longitudinally Varying Flow, Journal of the Atmospheric Sciences, 50, 1661–1671, https://doi.org/10.1175/1520-0469(1993)050<1661:RWPOAR>2.0.CO;2, 1993.

Hoskins, B. J. and Karoly, D. J.: The Steady Linear Response of a Spherical Atmosphere to Thermal and Orographic Forcing, Journal of the Atmospheric Sciences, 38, 1179–1196, https://doi.org/10.1175/1520-0469(1981)038<1179:TSLROA>2.0.CO;2, 1981.

Hsu, H.-H. and Lin, S.-H.: Global Teleconnections in the 250-mb Streamfunction Field during the Northern Hemisphere Winter, Monthly Weather Review, 120, 1169–1190, https://doi.org/10.1175/1520-0493(1992)120<1169:GTITMS>2.0.CO;2, 1992.

Jiménez-Esteve, B., Kornhuber, K., and Domeisen, D. I. V.: Heat Extremes Driven by Amplification of Phase-Locked Circumglobal Waves Forced by Topography in an Idealized Atmospheric Model, Geophysical Research Letters, 49, e2021GL096 337, https://doi.org/10.1029/2021GL096337, 2022.

Kornhuber, K., Petoukhov, V., Petri, S., Rahmstorf, S., and Coumou, D.: Evidence for wave resonance as a key mechanism for generating high-amplitude quasi-stationary waves in boreal summer, Climate Dynamics, pp. 1–19, https://doi.org/10.1007/s00382-016-3399-6, 2016.

Kretschmer, M., Adams, S. V., Arribas, A., Prudden, R., Robinson, N., Saggioro, E., and Shepherd, T. G.: Quantifying Causal Pathways of Teleconnections, https://doi.org/10.1175/BAMS-D-20-0117.1, section: Bulletin of the American Meteorological Society, 2021.

Limpasuvan, V. and Hartmann, D. L.: Wave-Maintained Annular Modes of Climate Variability, Journal of Climate, 13, 4414–4429,
https://doi.org/10.1175/1520-0442(2000)013<4414:WMAMOC>2.0.CO;2, publisher: American Meteorological Society Section: Journal
of Climate, 2000.

Lorenz, D. J. and Hartmann, D. L.: Eddy–Zonal Flow Feedback in the Northern Hemisphere Winter, Journal of Climate, 16, 1212–1227,
https://doi.org/10.1175/1520-0442(2003)16<1212:EFFITN>2.0.CO;2, publisher: American Meteorological Society Section: Journal of
Climate, 2003.

Mann, M. E., Rahmstorf, S., Kornhuber, K., Steinman, B. A., Miller, S. K., and Coumou, D.: Influence of Anthropogenic Climate Change
on Planetary Wave Resonance and Extreme Weather Events, Scientific Reports, 7, 45 242, https://doi.org/10.1038/srep45242, 2017.

Mann, M. E., Rahmstorf, S., Kornhuber, K., Steinman, B. A., Miller, S. K., Petri, S., and Coumou, D.: Projected changes in persistent extreme
summer weather events: The role of quasi-resonant amplification, Science Advances, 4, eaat3272, https://doi.org/10.1126/sciadv.aat3272,
publisher: American Association for the Advancement of Science, 2018.

Manola, I., Selten, F., de Vries, H., and Hazeleger, W.: "Waveguidability" of idealized jets, Journal of Geophysical Research: Atmospheres,
118, 10,432–10,440, https://doi.org/10.1002/jgrd.50758, 2013.

Marengo, J. A., Ambrizzi, T., Kiladis, G., and Liebmann, B.: Upper-air wave trains over the Pacific Ocean and wintertime cold surges in
tropical-subtropical South America leading to Freezes in Southern and Southeastern Brazil, Theoretical and Applied Climatology, 73,
223–242, https://doi.org/10.1007/s00704-001-0669-x, 2002.

Martius, O., Schwierz, C., and Davies, H. C.: Tropopause-Level Waveguides, Journal of the Atmospheric Sciences, 67, 866–879,
https://doi.org/10.1175/2009JAS2995.1, 2010.

McKinnon, K. A., Rhines, A., Tingley, M. P., and Huybers, P.: Long-lead predictions of eastern United States hot days from Pacific sea
surface temperatures, Nature Geoscience, 9, 389–394, https://doi.org/10.1038/ngeo2687, 2016.

Methven, J. and Berrisford, P.: The slowly evolving background state of the atmosphere, Quarterly Journal of the Royal Meteorological
Society, 141, 2237–2258, https://doi.org/10.1002/qj.2518, _eprint: https://onlinelibrary.wiley.com/doi/pdf/10.1002/qj.2518, 2015.

Nakamura, N. and Huang, C. S. Y.: Atmospheric blocking as a traffic jam in the jet stream, Science, 361, 42–47,
https://doi.org/10.1126/science.aat0721, publisher: American Association for the Advancement of Science, 2018.

Nakamura, N. and Solomon, A.: Finite-Amplitude Wave Activity and Mean Flow Adjustments in the Atmospheric General Circulation. Part
II: Analysis in the Isentropic Coordinate, Journal of the Atmospheric Sciences, 68, 2783–2799, https://doi.org/10.1175/2011JAS3685.1,
publisher: American Meteorological Society Section: Journal of the Atmospheric Sciences, 2011.

Nakamura, N. and Zhu, D.: Finite-Amplitude Wave Activity and Diffusive Flux of Potential Vorticity in Eddy–Mean Flow Interaction,
Journal of the Atmospheric Sciences, 67, 2701–2716, https://doi.org/10.1175/2010JAS3432.1, 2010.

Parker, T. J., Berry, G. J., and Reeder, M. J.: The Structure and Evolution of Heat Waves in Southeastern Australia, Journal of Climate, 27,
5768–5785, https://doi.org/10.1175/JCLI-D-13-00740.1, 2014.

Petoukhov, V., Rahmstorf, S., Petri, S., and Schellnhuber, H. J.: Quasiresonant amplification of planetary waves and recent Northern Hemi-
sphere weather extremes, Proceedings of the National Academy of Sciences, 110, 5336–5341, https://doi.org/10.1073/pnas.1222000110,
2013.

Petoukhov, V., Petri, S., Rahmstorf, S., Coumou, D., Kornhuber, K., and Schellnhuber, H. J.: Role of quasiresonant planetary wave
dynamics in recent boreal spring-to-autumn extreme events, Proceedings of the National Academy of Sciences, 113, 6862–6867,
https://doi.org/10.1073/pnas.1606300113, 2016.

Polster, C.: wavestoweather/Rolling-Zonalization: Rolling-Zonalization v1.1, https://doi.org/10.5281/zenodo.10149381, 2023.

Polster, C. and Wirth, V.: A New Atmospheric Background State to Diagnose Local Waveguidability, Geophysical Research Letters, 50, e2023GL106 166, https://doi.org/10.1029/2023GL106166, _eprint: https://onlinelibrary.wiley.com/doi/pdf/10.1029/2023GL106166, 2023.

Rossby, C.-G.: Relation between Variations in the Intensity of the Zonal Circulation of the Atmosphere and the Displacements of the Semi-Permanent Centers of Action, Journal of Marine Research, 2, 38–55, 1939.

Rousi, E., Kornhuber, K., Beobide-Arsuaga, G., Luo, F., and Coumou, D.: Accelerated western European heatwave trends linked to more-persistent double jets over Eurasia, Nature Communications, 13, 3851, https://doi.org/10.1038/s41467-022-31432-y, 2022.

Röthlisberger, M., Frossard, L., Bosart, L. F., Keyser, D., and Martius, O.: Recurrent Synoptic-Scale Rossby Wave Patterns and Their Effect on the Persistence of Cold and Hot Spells, Journal of Climate, 32, 3207–3226, https://doi.org/10.1175/JCLI-D-18-0664.1, 2019.

Schubert, S., Wang, H., and Suarez, M.: Warm Season Subseasonal Variability and Climate Extremes in the Northern Hemisphere: The Role of Stationary Rossby Waves, Journal of Climate, 24, 4773–4792, https://doi.org/10.1175/JCLI-D-10-05035.1, 2011.

Screen, J. A. and Simmonds, I.: Caution needed when linking weather extremes to amplified planetary waves, Proceedings of the National Academy of Sciences, 110, E2327–E2327, https://doi.org/10.1073/pnas.1304867110, publisher: Proceedings of the National Academy of Sciences, 2013.

Screen, J. A. and Simmonds, I.: Amplified mid-latitude planetary waves favour particular regional weather extremes, Nature Climate Change, 4, 704–709, https://doi.org/10.1038/nclimate2271, 2014.

Segalini, A., Riboldi, J., Wirth, V., and Messori, G.: A linear assessment of barotropic Rossby wave propagation in different background flow configurations, Weather and Climate Dynamics, 5, 997–1012, https://doi.org/10.5194/wcd-5-997-2024, publisher: Copernicus GmbH, 2024.

Shutts, G. J.: The propagation of eddies in diffluent jetstreams: Eddy vorticity forcing of 'blocking' flow fields, Quarterly Journal of the Royal Meteorological Society, 109, 737–761, https://doi.org/10.1002/qj.49710946204, _eprint: https://onlinelibrary.wiley.com/doi/pdf/10.1002/qj.49710946204, 1983.

Teng, H., Branstator, G., Wang, H., Meehl, G. A., and Washington, W. M.: Probability of US heat waves affected by a subseasonal planetary wave pattern, Nature Geoscience, 6, 1056–1061, https://doi.org/10.1038/ngeo1988, 2013.

Vries, D. and Jan, A.: A global climatological perspective on the importance of Rossby wave breaking and intense moisture transport for extreme precipitation events, Weather and Climate Dynamics, 2, 129–161, https://doi.org/10.5194/wcd-2-129-2021, 2021.

White, R. H., Battisti, D. S., and Roe, G. H.: Mongolian Mountains Matter Most: Impacts of the Latitude and Height of Asian Orography on Pacific Wintertime Atmospheric Circulation, Journal of Climate, 30, 4065–4082, https://doi.org/10.1175/JCLI-D-16-0401.1, 2017.

White, R. H., Kornhuber, K., Martius, O., and Wirth, V.: From Atmospheric Waves to Heatwaves: A Waveguide Perspective for Understanding and Predicting Concurrent, Persistent, and Extreme Extratropical Weather, Bulletin of the American Meteorological Society, 103, E923–E935, https://doi.org/10.1175/BAMS-D-21-0170.1, publisher: American Meteorological Society Section: Bulletin of the American Meteorological Society, 2022.

Wirth, V.: Waveguidability of idealized midlatitude jets and the limitations of ray tracing theory, Weather and Climate Dynamics, 1, 111–125, https://doi.org/10.5194/wcd-1-111-2020, publisher: Copernicus GmbH, 2020.

Wirth, V. and Polster, C.: The Problem of Diagnosing Jet Waveguidability in the Presence of Large-Amplitude Eddies, Journal of the Atmospheric Sciences, 78, 3137–3151, https://doi.org/10.1175/JAS-D-20-0292.1, 2021.

Wirth, V., Riemer, M., Chang, E. K. M., and Martius, O.: Rossby Wave Packets on the Midlatitude Waveguide—A Review, https://doi.org/10.1175/MWR-D-16-0483.1, section: Monthly Weather Review, 2018.

Wolf, G. and Wirth, V.: Implications of the Semigeostrophic Nature of Rossby Waves for Rossby Wave Packet Detection, Monthly Weather Review, 143, 26–38, https://doi.org/10.1175/MWR-D-14-00120.1, 2015.

Wolf, G., Brayshaw, D. J., Klingaman, N. P., and Czaja, A.: Quasi-stationary waves and their impact on European weather and extreme events, Quarterly Journal of the Royal Meteorological Society, 144, 2431–2448, https://doi.org/10.1002/qj.3310, 2018.

Wolf, G., Brayshaw, D., and Klingaman, N.: Response of atmospheric quasi-stationary waves to La Niña conditions in Northern Hemisphere winter, Quarterly Journal of the Royal Meteorological Society, 148, 1611–1622, https://doi.org/10.1002/qj.4261, 2022.

Woollings, T., Barriopedro, D., Methven, J., Son, S.-W., Martius, O., Harvey, B., Sillmann, J., Lupo, A. R., and Seneviratne, S.: Blocking and its Response to Climate Change, Current Climate Change Reports, 4, 287–300, https://doi.org/10.1007/s40641-018-0108-z, 2018.

Xu, P., Wang, L., and Chen, W.: The British–Baikal Corridor: A Teleconnection Pattern along the Summertime Polar Front Jet over Eurasia, https://doi.org/10.1175/JCLI-D-18-0343.1, section: Journal of Climate, 2019.

Xu, P., Wang, L., Chen, W., Chen, G., and Kang, I.-S.: Intraseasonal Variations of the British–Baikal Corridor Pattern, https://doi.org/10.1175/JCLI-D-19-0458.1, section: Journal of Climate, 2020.

Xu, P., Wang, L., Vallis, G. K., Geen, R., Screen, J. A., Wu, P., Ding, S., Huang, P., and Chen, W.: Amplified Waveguide Teleconnections Along the Polar Front Jet Favor Summer Temperature Extremes Over Northern Eurasia, Geophysical Research Letters, 48, e2021GL093 735, https://doi.org/10.1029/2021GL093735, _eprint: https://onlinelibrary.wiley.com/doi/pdf/10.1029/2021GL093735, 2021.

Zimin, A. V., Szunyogh, I., Patil, D. J., Hunt, B. R., and Ott, E.: Extracting Envelopes of Rossby Wave Packets, Monthly Weather Review, 131, 1011–1017, https://doi.org/10.1175/1520-0493(2003)131<1011:EEORWP>2.0.CO;2, 2003.