# Peer review of "Time and Zonally-varying Atmospheric Waveguides - Climatologies and Connections to Quasi-Stationary Waves"

_EGUsphere, 2024_

## Referee Comment (RC2)

**Review of the manuscript** "Time-varying Atmospheric Waveguides —
Climatologies and Connections to Quasi-Stationary Waves" by R. White, submitted
for publication to Weather and Climate Dynamics

**Recommendation:** Minor revisions or a complete rewrite

**General**

The concept of a Rossby waveguide has recently found increased interest. This recent
interest may be partly due to the hypothesis of Petoukhov *et al.* (2013), who
considered circumglobal waveguides and investigated the possibility of Rossby wave
resonance during specific episodes. Their method to diagnose the existence of a
waveguide from observations was based on arguments used earlier by Hoskins and
Karoly (1981) and Hoskins and Ambrizzi (1993), which in turn start with linear wave
theory, make additional assumptions (like the so-called WKB approximation), and
finally arrive at the concept of a refractive index. Under special circumstances, the
basic state may feature two so-called turning latitudes for a given wavenumber, and
this is then interpreted as a perfect zonal waveguide for the respective wavenumber.
In the remainder of this text I will refer to this framework with the "Two Turning
Latitudes" as TTL analysis or TTL thinking.

The current paper has, broadly speaking, two parts. In the first part the author
produces and discusses, for the first time, a climatology for waveguide occurrence
based on TTL analysis. As a particular feature the author considers a background
atmosphere that is allowed to have a smooth variation in longitude, such that her
waveguides, too, include a smooth variation in longitude. In the second part, the
author goes on and correlates waveguidability as diagnosed from TTL analysis with
Rossby wave amplitude, where both waveguidability and wave amplitude are allowed
to vary smoothly with longitude. In both parts, the background atmosphere (needed
to define a waveguide) is obtained through a combination of temporal and spatial
filtering.

In the past, I have raised two major issues with the TTL analysis as a method to
diagnose waveguides and waveguidability and how it is usually applied. First, in an
idealized modeling framework I designed a method to diagnose "true"
waveguidability and compare it with TTL-based waveguidability (Wirth, 2020). My
"true waveguidability diagnostic" does not make use of the WKB assumption, which
underlies the TTL analysis but which is badly violated in realistic situations.
Therefore, in case of discrepancies between the two methods, my "true
waveguidability" would naturally be given priority. As it turns out, TTL-based
waveguidability is severly flawed in that it is unable to reproduce the gradual increase
in true waveguidability as the strength or the narrowness of a jet is increased (see
also Manola et al. 2013). In particular, the association of the existence of two turning
latitudes with a (perfect!) waveguide for the corresponding zonal wavenumber was
shown to be highly problematic. My second issue concerns the fact that the
TTL-based analysis may be subject to artefacts in the event of large wave amplitudes,
if the used background state is based on zonal averaging (Wirth and Polster 2021).

I do not contend that these two issues reduce the utility of TTL-theory to zero, but I would certainly say that TTL-theory is unable to represent certain (possibly important) aspects regarding waveguides and waveguidability. Unfortunately, we do not have a satisfying understanding yet that would allow one to distinguish those aspects for which the application of TTL-theory is appropriate from those for which it is not. This situation calls for a high level of care that needs to be exercised. Incidentally, as far as I can see, earlier papers such as Hoskins and Karoly (1981) and Hoskins and Ambrizzi (1993) never based their conclusions on TTL analysis alone; rather, their work typically included some independent statistical analysis and/or numerical modelling, and whenever all these approaches yielded consistent results, the TTL-diagnostic was used for interpretative purposes.

Despite the issues that I raised above, the TTL analysis of waveguides and waveguidability enjoys ongoing popularity, and the caveats associated with this approach are sometimes simply ignored. The current paper seems to continue along this tradition. To be sure, the author quotes all the relevant papers including the two critical ones I just mentioned. She even says explicitly that caution is needed when interpreting results that are based on TTL theory (line 90). But then she simply goes on and does not really attempt to critically discuss the implications of these papers for her own work. In my eyes this is not satisfactory. I would expect that the author gives strong arguments why she thinks that the issues I raised are irrelevant in the context of her paper. Otherwise, the reader is left alone with the question: "why learn more about a diagnostic that was shown to be severely flawed in relevant applications"?

Interestingly, when I read the paper for a second time, it occurred to me that the results from both parts can actually be interpreted in a way that further supports the criticism raised by Wirth (2020) and Wirth and Polster (2021) — see my further explanations in the major issues below. I believe that such an interpretation was not the original intention of the author.

Given this situation, I am not sure what to recommend. One option might be that the author is able to argue in favor of TTL thinking and dismiss my issues on the basis of compelling arguments. This option would probably result in minor revisions to the manuscript, like for instance adding a new paragraph that contains the relevant explanations. Personally, I cannot see how this could work, but I may be biased or miss an important point, and I am happy to get involved in a discussion and learn more. As another option I could imagine a statistical/climatological comparison of the TTL diagnostic with an alternative diagnostic like, for instance, the one of Polster and Wirth (2023). This option would make use of most of the work performed in preparing this paper, but of course it would mean a complete rewrite.

**Major issues**

1. The first set of results (section 4.1) presents climatological properties of TTL waveguidability, and it turns out that (broadly speaking) waveguidability is stronger or waveguides are more frequent in summer compared to winter. This is in striking conflict with the fact that jets are generally stronger in winter than

in summer, and the latter implies stronger (true) waveguidabiliy in winter compared to summer according to Manola *et al.* (2013) or Wirth (2020). The author mentions this conflict on line 168 and adds later on line 263 that "further research is required to fully investigate this result". I find this not very satisfying. One way to resolve this conflict would be to admit that TTL-based waveguidability is flawed and inappropriate for a reliable diagnosis in this context, consistent with the arguments of Wirth (2020).

2. Figure 6 of the paper indicates that the "black boxed regions" with strong waveguide occurrence are characterized by weaker than normal zonal wind. Again, this result is in striking conflict with the results of Manola *et al.* (2013) and Wirth (2020), who showed that strong waveguides are generally associated with stronger rather than weaker jets. As far as I can see, the author does not resolve this conflict. One way to resolve the conflict would be to acknowledge that the TTL-waveguide diagnostic is fraught by the artefact discussed in Wirth and Polster (2021, see my further comments below), and that previous authors who used the TTL-diagnostic are subject to the same artefact.

3. The second part of the paper (section 5) shows, broadly speaking, a (weak, but statistically significant) positive correlation between local Rossby wave amplitude and local TTL-waveguidability in certain regions (Fig. 10). At first sight this result was surprising to me, given that the author does not really give a motivation why one should expect such a correlation (e.g., on lines 98 and line 280ff, where such a connection is more or less assumed to be given). To be sure, in the case of circumglobal waveguides, a motivation might arise from the contested quasi-resonance arguments of Petoukhov et al. (2013). But this line or arguments cannot possibly provide a motivation for the present analysis, because Rossby wave resonance requires circumglobal waveguides, while the author here diagnoses local (even gridpointwise) waveguides. On line 210, the author mentions two papers in which such a correlation allegedly was hypothesized, but as far as I can see the waveguides in these papers were assumed to be circumglobal rather than local. Therefore, I disagree with the author's statement in the discussion section (line 279) that such a connection was hypothesized "to some extent" in previous articles.

However, after second thought I realized that this correlation may be a result of the artefact which I discussed in Wirth and Polster (2021). The author herself provides a hint in her Figs. 6 and 7, where she shows that strong TTL-waveguides are associated with tripole-like anomalies in the zonal wind corresponding to a "double jet structure". This result, in combination with the positive correlation between local TTL-waveguidability and Rossby wave amplitude, offers the following interpretation: according to the argument of Wirth and Polster (2021), these strong TTL-waveguides may simply be artefacts arising from strong Rossby wave amplitudes. Large wave amplitudes would distort the total (= background plus wave) flow pattern such that one obtains a tendency towards a double-jet structure in the zonal average (see, e.g., the schematic in Fig. 1 of Wirth and Polster 2021).

To the extent that there is no alternative plausible motivation for the correlations in Fig. 8, my argument suggests that Fig. 8 could actually be

interpreted as an independent (data-based) confirmation of the Wirth-Polster criticism.

A way to test this hyopthesis would be to use the novel zonally varying background state from Polster and Wirth (2023). In fact, the author mentions this idea on line 293. It would not be too hard to perform this analysis, since Polster and Wirth published her code along with the paper. The basic state from Polster and Wirth is based on a "rolling zonalized" background field, which is not subject to the Wirth-Polster artefact — in contrast to the background state used in the current paper. If the correlation vanished upon the use of this (presumably more appropriate) basic state, one would have produced an independent piece of evidence for the statement that, indeed, the tripole-structures in Fig. 6 and 7 essentially reflect the artefact discussed in Wirth and Polster (2021).

4. In Fig. 4, the author introduces a novel metric for "waveguide depth" $W_d$. This is an interesting idea, because $W_d$ represents a somewhat more "integral" measure for the strength of a waveguide, in contrast to the search for two turning latitudes; the latter only relies on the intersection of the $K_s$-profile with a line representing a fixed wavenumber and introduces an artificial "waveguide vs. no-waveguide" dichotomy (Wirth 2020). Fig. 4 in Wirth (2020) suggests that stronger jets are generally associated with stronger $W_d$ as defined here. However, $W_d$, too, is unlikely to represent the increase of true waveguidability with increasing jet strength beyond a certain limit. This was explicitly discussed in Wirth (2020) in connection with his Fig. 6a, where true waveguidability increases from about 48% to about 75% while jet strength was increased from 20 to 40 m/s, and this increase in true waveguidability would be completely missed by $W_d$. In fact, I believe there are better (and simpler) measures for waveguide strength, such as the horizontal PV gradient.

**Minor issues**

1. Line 15 :. . . it can be associated with extreme weather occasionally, but certainly not always.

2. Line 20: I assume that the issue with stationarity is just as severe in connection with precipitation as it is in connection with heat. E.g., the flooding events in Germany (2002), Pakistan (2010), and Germany (2021) were associated with quasi-stationary circulation patterns.

3. Line 25: are you here referring mostly to circumglobal jets? Line 209/210: again, why should a strong local waveguide be associated with strong wave amplitude? Petoukhov et al (2013) hypothesize such a connection in the case of circumglobal waveguides, but you have a very local (grid-point wise) perspective on waveguides.

4. Line 43: ". . . . theory provides qualitatively useful insights. . . .": how is this possible, if the underlying assumptions are not valid? Is this by pure chance? How about the issues in Wirth (2020), who showed that there are relevant

aspects, in which TTL-theory does not provide even qualitatively realistic results? If the theory provides sometimes useful results and sometimes not: how can one distinguish between these two alternatives?

5. Line 88: Here you seem to refer to Wirth and Polster (2021), not to Wirth (2020).

6. Line 94: to some people, a "nonlinear wave" is an oxymoron. I would prefer to speak about "nonlinear eddies" or "nonlinear perturbations" and reserve the term "waves" for linear dynamics.

7. Line 98: . . . . but this is true only if the waveguides are circuglobal!

8. Line 100: a running temporal mean?

9. Line 101: do you really mean "planetary wavenumbers of interest" . . . .? It appears to me that you are, rather, aiming to extract synoptic-scale wavenumbers here.

10. Line 103: "wave envelope": Do you really mean the wave envelope of the planetary waves?

11. Line 103: "15-day running mean": yet another temporal filter? Haven't the data already been filtered temporally (line 100)?

12. Line 104: the Hilbert transform is usually applied to compute the wave amplitude, not the wave itself. What do you mean here?

13. Line 132: A westerly jet with strength 0.5 m/s is not very impressive in my eyes, and somehow conflicts with the idea of Manola et al (2013) that a jet needs to be both narrow and strong to be a good waveguide. In addition, the desired narrowness of the jet would suggest rather a criterion that restricts to a maximum width (rather than a minimum width, as you suggest).

14. Line 135: . . . show that the results... "

15. Line 145/146: To me it seems as if you identify frequent waveguide occurrence with large waveguide amplitude, which I think is dangerous. Waveguide frequency and waveguide strength should be distinguished and not mixed together.

16. Figure 2: Can you explain the solid contours in the figure caption!?

17. Line 153: You find more waveguides in summer compared to winter, although the jet in summer is usually weaker. Isn't this inconsistent with the results of Manola et al (2013) and Wirth (2020), who show that stronger jets are usually better waveguides, hence one would expect higher waveguide frequency in winter than in summer?! Can you resolve this issue? A similar problem appears on line 167/168 when comparing Northern and Southern Hemispheres.

18. Line 170 and following: here you show that your results are consistent with the formula from the theory that you apply, but at the same time they are inconsistent with results from Manola et al. (2013) or Wirth (2020). Does this

mean that you trust the TTL-theory more than the results of Manola and Wirth? That seems dangerous, because the latter do not rely on this (somewhat questionable) theory.

19. Line 179: similar as above, you find stronger waveguide "depths" in summer compared to winter, although other work suggests that weaker jets in summer should be weaker waveguides. Can you resolve?

20. Line 210: Well, that's not quite right. Petoukhov et al. hypothesize such a correlation only for circumglobal waveguides (which is needed for Rossby wave resonance), they do not hypothesize such a correlation between local wave amplitude and local waveguidability.

21. Line 260-265: here you seem to play down the conflict between your results and those of the literature; I would expect a lucid discussion and explanation how you think that these discrepancies can be "explained"

22. Lines 280-290: you mention a few results from the literature, but it did not become clear to me how they relate to your results. In particular, it did not become clear to me how you would address the criticism formulated in some of these papers. As I argue in the first part of my review, I believe that some of your results even provide additional support for some of the criticism formulated.

23. Line 296: Can you explain how your data set potentially can shed light on causality?

Mainz, 3 May 2024

Volkmar Wirth

**References**

Hoskins, B. J., and T. Ambrizzi 1993. Rossby wave propagation on a realistic longitudinally varying flow. *J. Atmos. Sci.* **50**, 1661–1671.

Hoskins, B. J., and D. J. Karoly 1981. The Steady Linear Response of a Spherical Atmosphere to Thermal and Orographic Forcing. *J. Atmos. Sci.* **38**, 1179–1196.

Manola, I., F. Selten, H. de Vries, and W. Hazeleger 2013. "Waveguidability" of idealized jets. *J. Geophys. Res.* **118**, 10,432–10,440, doi:10.1002/jgrd.50758.

Petoukhov, V., S. Rahmstorf, S. Petri, and H.-J. Schellnhuber 2013. Quasiresonant amplification of planetary waves and recent Northern Hemisphere weather extremes. *Proceedings of the National Academy of Sciences 110*(14), 5336–5341, doi:10.1073/pnas.1222000110.

Polster, C., and V. Wirth 2023. A new atmospheric background state to diagnose local waveguidability. *GRL* **50**, DOI:10.1029/2023GL106166.

Wirth, V. 2020. Waveguidability of idealized midlatitude jets and the limitations of ray tracing theory. *Weather Clim. Dynam.* **1**, 111–125, https://doi.org/10.5194/wcd–1–111–2020.

Wirth, V., and C. Polster 2021. The problem of diagnosing jet waveguidability in the presence of large-amplitude eddies. *J. Atmos. Sci.* **78**, 3137–3151, https://doi.org/10.1175/JAS–D–20–0292.1.

---

## Author Comment (AC1)

I have copied the major comments of the reviewer in black, along with my responses in blue.

Major comments:

a) Abstract: the notion of waveguide depth is obscure there. When I read the abstract I thought it was considering the depth in altitude. It would be good to add words to qualitatively define the waveguide depth. For me, the waveguide depth is closely related to the number of zonal wavenumbers for which the waveguide exists.

Thank you for this helpful feedback – I will adjust this terminology. I am considering 'waveguide amplitude' instead of depth to avoid the confusion with altitude (feedback on this idea is welcome!), and I will give more description on what the definition of this term to provide more clarification.

b) Introduction (lines 25 to 29). It would be good to add some physical interpretation of why people think there is a link between waves amplitude and waveguides. In my opinion, this is potentially because waves are not dissipated as there are no critical latitudes in a waveguide. But I am not sure this is what the papers cited in line 28 have argued.

Yes, I agree that the lack of dissipation of wave energy within the waveguide, and channeling of wave energy into the latitude band of the waveguide is at least part of the physical interpretation. It is also possible the typically enhanced stationary wavenumber within a waveguide, allowing waves of higher wavenumber to become (quasi-)stationary, also plays a role. In addition, if the waveguides are quasi-stationary, this would allow quasi-stationary channeling of wave energy into this particularly latitude band. I will expand on this part of the introduction in a revised manuscript.

c) Method: it would be good to highlight that the background flow and the waves are separated by the spatial scale (k<2 for the background flow and k between 4 and 15 for the quasi-stationary waves). Since the 15-day running mean is used for the detection of both the background flow and the waves, the reader might be confused by the separation between these two parts of the flow.

Thank you, I will add clarification text on this. I have also repeated the quasi-stationary wave analysis using k between 6-15 to provide even greater separation, and results are very similar. I will add this to the sensitivity analysis.

d) Section 4.1 and waveguide frequencies:

- before starting that section it would have been nice to show Ks for the time mean flow of JJA and DJF, i.e repeat Figures 3c of Hoskins and Ambrizzi (1993) and 11c of Ambrizzi et al (1995). Maybe it would be good to do it by considering the climatological flow for k<2 to be close to what is done in the present paper for the time-evolving waveguides. Such additional figures could help to better visualize the difference between summer and winter and between SH and NH. The argument made lines 160-165 to qualitatively explain why the summer NH has more frequent waveguides than the winter NH could be better understood by showing the time mean U and Ks for both seasons. Furthermore, maybe the additional argument is the fact that the jet is probably narrower in summer than winter and both the planetary vorticity gradient and relative vorticity gradient play a role in the difference between summer and winter.

This is a nice idea, thank you. I will add climatologies for the time mean flow; although, because Ks is non-linear, the Ks for the time mean flow may be quite different from the time mean Ks. But I agree that plots of the climatological fields may help provide more physical interpretation of the

results, particularly in the context of the discussion for the next point.

- I am surprised that the SH has less waveguides than the NH as the double jet structure (separation between subtropical and eddy-drivent jet) is more marked there, at least in the climatologies.

Yes, I agree that this result is interesting, although slightly stronger waveguide frequency in Northern vs Southern hemispheres can also be seen in Polster and Wirth (2023) in Fig. 3, and so this result is consistent with previously published analysis. I will add references to this paper and add more discussion around this topic in a revised manuscript. It is also worth noting that the waveguide frequencies are giving the frequency of a waveguide existing at that latitude, not the frequency of a waveguide at that longitude in that hemisphere – thus jets with more variable latitude may show a reduced waveguide frequency at any one latitude. Some evidence in support of this idea can be seen in Manola et al. 2013 (Fig. 5 e and f), although this is comparing SH summer with NH winter. There may also be an influence of differences in the width of the jets between the northern and southern hemisphere. For example. Manola et al. 2013 (Fig. 5 c and d) show some differences between the jet widths (again, comparing SH summer with NH winter), and in the SH jet widths of up to 16-18 degrees latitude are more common than in the NH. This importance of jet width can also be seen in the new figure you suggested, which I have added below, looking at composites of zonal winds (not just anomalies) for strong vs low/no waveguide days - the differences are often more in the narrowness of the jet, and not always in the maximum strength. Lastly, Hoskins and Woollings show the climatological stationary wavenumber for both hemispheres and both seasons; poleward of 25S/N, the climatological stationary wavenumber in the jets tends to be stronger in the NH than in the SH. It is possible that the jets in the SH are in fact "too" separated, such that the meridional gradients of zonal wind are not as stronger as in the NH. As suggested above, I will plot the climatological stationary wavenumber distributions, and discuss the hemispheric and seasonal differences of waveguide frequency in the context of the climatological distributions of stationary wavenumber.

e) Section 4.2: this is the part of the results where I am less convinced by the conclusions. For instance, Figure 6a shows an anomalous tripolar pattern in zonal wind when computing the difference between high and low waveguide strengths. This anomaly could be the result of different changes: a more pronounced double jet structure is one possibility but it could result from a widening of the jet or some latitudinal shifts. So it would be very nice to compare composites of high waveguide strengths and low waveguide strengths separately before (or rather than) showing the difference.

Thank you for this suggestion. I have made these plots of the composites (see below) and will add these to the paper to allow more physical interpretation of the anomalies currently shown. As discussed above, these composites do show a double jet structure in many regions, however this is also related to a southward shift in the lower latitude jet relative to the no/low waveguide condition.

[Figure]

Zonal wind (m/s) for NH summer composites. Left column: no/low waveguide strength; right column: high waveguide strength. Different rows denote waveguides in the different boxed region, ad in Fig. 6 in the paper.

f) Section 5: It is surprising that the correlations are strong in the Atlantic and over Asia and not in the Pacific while the waveguide depths are similar in the North Atlantic and North Pacific. What would be a possible explanation for that? Or if you do not have hypothesis it would be nice to comment these results by referring to other studies. Were the studies on the relationship waveguide-wave amplitude focused in the North Atlantic and Asian regions. Do you know studies that also considered that relationship in the North Pacific?

I agree that this is a somewhat surprising result, and may perhaps be some combination of longitudinal variations in waveguide strength, longitudinal waveguide extent, and the location of wave sources relative to the waveguides, however I do not have a clear hypothesis. I will look into this more for a revision. There is still a positive correlation across much of the Pacific region in most sensitivity analyses - the correlation is weaker and not statistically significant, however it may be physically real.

g) Sensitivity tests: I think it would be good to have a sensitivity test by changing the mean pressure level (e.g. 500 hPa?). Held et al. (1985) computed a barotropic equivalent level near 425 hPa and Charney (1949, see section 6) found a barotropic equivalent level closer to 550-600 hPa.

This is an interesting idea, thank you – I will add this to the sensitivity tests.

Thank you for your minor comments. I will add in these suggestions and corrections.

---

## Author Comment (AC2)

I have copied the main comments and major concerns of the reviewer in black, along with my responses in blue.

The atmospheric waveguide has a profound influence on the propagation path of stationary Rossby waves, thereby affecting when and where these waves impact surface weather and climate. In recent years, studies on the atmospheric waveguide have gained popularity among the climate community due to its significant connection to extreme events. The investigation of waveguides can be traced back to early works, notably Hoskins and Karoly (1981), followed by Hoskins and Ambrizzi (1993) and Ambrizzi et al. (1995). This current study aims to extend previous research by examining the waveguide in the context of spatially and temporally varying mean flow. Most of the analysis is focusd on this issue.

This is a nice manuscript that uses refractive index as a perspective to understand atmospheric waveguide and its connection to stationary Rossby waves. In my opinion, it holds the potential to be considered for publication in a WCD. However, I have several major concerns about the methodology and interpretation of the results. I have listed my major comments below and would like to invite the authors to address them:

I thank the reviewer for their thoughtful comments. Please see my responses below.

1. The separation of mean flow and perturbations is always a controversial issue when studying wave-mean flow interactions. This issue becomes even more critical when large-amplitude eddies appear in the mean flow (e.g., Wirth and Polster, 2021). However, the present study heavily relies on the separation method, and most of the findings are based on the assumption that the waveguide and Rossby waves are well-separated and independent. Therefore, I question the significance of the results, as many intraseasonal waveguide behaviors are actually reflected by long-lasting waves.

I agree that sub-seasonal changes in waveguides can be caused by long-lasting waves; however the changes that are caused by those waves, particularly waves of low zonal wavenumber, can cause local changes in waveguides, which may influence higher wavenumber waves. In this work I attempt to separate out the low and high wavenumber waves through wavelength decomposition, with wavenumbers 2 and lower included in the background flow, and wavenumbers 4 and higher considered as waves. In response to the reviews, I have repeated the quasi-stationary wave analysis (correlation with co-located waveguides) using wavenumbers 6 and higher for the waves (retaining the definition of 2 and lower for the background flow); the results are relatively insensitive to this, suggesting that overlap between what is counted as background flow and what is counted as waves likely contributes little to the results shown in Figs. 8 and 10. That is not to say that there is no influence on non-linear perturbations (e.g. blocking) on the waveguides; indeed, I think this work shows a potential connection between blocks and waveguides. New results created in response to the reviewer comments, looking at composites of geopotential height anomalies, show that, at some (but likely not all) longitudes, high latitude blocks likely play a role in *creating* the conditions for a waveguide to exist at lower latitudes through their influence on the localized zonal winds. Such a waveguide may then play a role in trapping quasi-stationary waves of higher wavenumbers at lower latitudes. This may explain some of the 'double quasi-stationary wave' pattern (waves of different wavenumber seen

in different latitudes) seen in several extreme months (e.g. White et al. 2022 Fig. 1), and the coincidence between blocks and recurrent Rossby waves (which would show up in the quasi-stationary wave metric used here) found by Mubashshir Ali et al. 2022.

2. Regarding the methodology, using the traditional turning point perspective to identify the waveguide could be misleading, despite its extensive use in recent studies such as Petoukhov et al. (2013) and many subsequent papers. The limitations of this method have been thoroughly discussed by Wirth (2020). Therefore, the authors need to demonstrate the limitation of the method used here is nontrivial and confirm the appropriateness of the method.

The limitations discussed by Wirth (2020) primarily concern the zonal mean perspective in the presence of very idealized, high amplitude non-linear perturbations to the flow. In this work I move away from the zonal mean perspective, partially due to some of these limitations. As discussed above, composites of geopotential height anomalies suggest that, at some longitudes, the presence of waveguides is indeed associated with the presence of a block (high geopotential height anomalies); however, the relative location, with the block located to the north of the waveguide, suggests that it is more likely that the block acts to increase the probability of the waveguide through its impact on the zonal winds. This new analysis will be added to a revised manuscript.

3. I doubt about the characterization of the atmospheric circulation associated with the waveguide strength as "double jet streams" (Figure 6), as the zonal wind anomalies are only confined to a local scale. Additionally, as related to my major comment 1, long-lasting waves might play a role in this structure. Therefore, it is possible that the pattern seen in Figure 6 is not "double jet streams", but prominant Rossby wave activity itself.

I did not mean to imply that these double jet stream existed on a hemispheric scale; indeed, Figure 6 shows clearly that they do not, and they are associated with localized enhanced waveguide presence. I also agree that the composites of zonal wind in this current paper support the idea that nonlinear anomalies such as block may in fact lead to the waveguide conditions, rather than causality in the opposite direction.  I will put more emphasis on this result in a revised manuscript, and, as discussed above, intend to include composites of geopotential height anomalies, showing that, in some regions, the presence of strong waveguides is associated with a region of enhanced geopotential height **poleward** of the waveguide location, consistent with the zonal wind changes that create the double jet structure shown in the previous composites. This result is consistent with the idea of an atmospheric block helping to create the local double jet which helps create the waveguide conditions, supportive of some of the ideas of Wirth and Polster (2021). Taking a zonal mean of the zonal composites shown in this paper, would very likely show a double jet structure; however, importantly, the 'double jet' found in these composites is a localized double jet, not one with hemispheric extent, and a hemispheric average would likely be misleading, as stated by Wirth and Polster (2021).

4. The current study primarily focuses on the waveguide effect along the subtropical jet, based on the refractive index, which essentially represents the gradient of absolute vorticity. However, recent studies (e.g., Xu et al. 2019, doi: 10.1175/JCLI-D-18-0343.1; Xu et al., 2020, doi:

10.1175/JCLI-D-19-0458.1) have presented compelling evidence of the existence of stationary Rossby waves along the eddy-driven jet, where the waveguide effect arises due to the gradient of potential vorticity. As mentioned by the author herself, this important aspect has been neglected due to the limitations of the methodology used in this manuscript. The authors briefly touch upon this issue in the manuscript, but in my opinion, more in-depth discussion is required.

The frequency of higher latitude waveguides is not zero in this dataset (see Fig 2); however, it is notable that the positive correlation between waveguides and quasi-stationary waves is not present poleward of 50N. It is possible that stability plays a more important role in the presence of waveguides at higher latitudes, as demonstrated in the two papers suggested by the reviewer. As suggested, I will include a more in-depth discussion of this, including those papers, in a revised manuscript.

Notably, even the PV-based methodology of Polster and Wirth (2023) shows weak waveguide frequency at high latitudes; however, the Polster and Wirth (2023) definition of waveguides could be easily adjusted to have a lower PV gradient threshold, that may detect more frequent waveguides at higher latitudes – this would need to be explored, but is outside the scope of the current work. It would be interesting to repeat the correlation analysis between waveguides and quasi-stationary waves on such PV-detected waveguides, to see if positive correlations are found further polewards; I will suggest this in the section on future research.

---

## Author Comment (AC3)

I have copied the main comments and major concerns of the reviewer in black, along with my responses in blue.

The concept of a Rossby waveguide has recently found increased interest. This recent interest may be partly due to the hypothesis of Petoukhov et al. (2013), who considered circumglobal waveguides and investigated the possibility of Rossby wave resonance during specific episodes. Their method to diagnose the existence of a waveguide from observations was based on arguments used earlier by Hoskins and Karoly (1981) and Hoskins and Ambrizzi (1993), which in turn start with linear wave theory, make additional assumptions (like the so-called WKB approximation), and finally arrive at the concept of a refractive index. Under special circumstances, the basic state may feature two so-called turning latitudes for a given wavenumber, and this is then interpreted as a perfect zonal waveguide for the respective wavenumber. In the remainder of this text I will refer to this framework with the "Two Turning Latitudes" as TTL analysis or TTL thinking. The current paper has, broadly speaking, two parts. In the first part the author produces and discusses, for the first time, a climatology for waveguide occurrence based on TTL analysis. As a particular feature the author considers a background atmosphere that is allowed to have a smooth variation in longitude, such that her waveguides, too, include a smooth variation in longitude. In the second part, the author goes on and correlates waveguidability as diagnosed from TTL analysis with Rossby wave amplitude, where both waveguidability and wave amplitude are allowed to vary smoothly with longitude. In both parts, the background atmosphere (needed to define a waveguide) is obtained through a combination of temporal and spatial filtering.

In the past, I have raised two major issues with the TTL analysis as a method to diagnose waveguides and waveguidability and how it is usually applied. First, in an idealized modeling framework I designed a method to diagnose "true" waveguidability and compare it with TTL-based waveguidability (Wirth, 2020). My "true waveguidability diagnostic" does not make use of the WKB assumption, which underlies the TTL analysis but which is badly violated in realistic situations. Therefore, in case of discrepancies between the two methods, my "true waveguidability" would naturally be given priority. As it turns out, TTL-based waveguidability is severly flawed in that it is unable to reproduce the gradual increase in true waveguidability as the strength or the narrowness of a jet is increased (see also Manola et al. 2013). In particular, the association of the existence of two turning latitudes with a (perfect!) waveguide for the corresponding zonal wavenumber was shown to be highly problematic. My second issue concerns the fact that the TTL-based analysis may be subject to artefacts in the event of large wave amplitudes, if the used background state is based on zonal averaging (Wirth and Polster 2021).

I do not contend that these two issues reduce the utility of TTL-theory to zero, but I would certainly say that TTL-theory is unable to represent certain (possibly important) aspects regarding waveguides and waveguidability. Unfortunately, we do not have a satisfying understanding yet that would allow one to distinguish those aspects for which the application of TTL-theory is appropriate from those for which it is not. This situation calls for a high level of care that needs to be exercised. Incidentally, as far as I can see, earlier papers such as Hoskins and Karoly (1981) and Hoskins and Ambrizzi (1993) never based their conclusions on TTL analysis alone; rather, their work typically included some independent statistical analysis and/or numerical modelling, and whenever all these approaches yielded consistent results, the TTL-diagnostic was used for interpretative purposes.

Despite the issues that I raised above, the TTL analysis of waveguides and waveguidability enjoys ongoing popularity, and the caveats associated with this approach are sometimes simply ignored. The current paper seems to continue along this tradition. To be sure, the author quotes all the relevant papers including the two critical ones I just mentioned. She even says explicitly that caution is needed when interpreting results that are based on TTL theory (line 90). But then she simply goes on and does not really attempt to critically discuss the implications of these papers for her own work. In my eyes this is not satisfactory. I would expect that the author gives strong arguments why she thinks that the issues I raised are irrelevant in the context of her paper. Otherwise, the reader is left alone with the question: "why learn more about a diagnostic that was shown to be severely flawed in relevant applications"?

Interestingly, when I read the paper for a second time, it occurred to me that the results from both parts can actually be interpreted in a way that further supports the criticism raised by Wirth (2020) and Wirth and Polster (2021) — see my further explanations in the major issues below. I believe that such an interpretation was not the original intention of the author.

Given this situation, I am not sure what to recommend. One option might be that the author is able to argue in favor of TTL thinking and dismiss my issues on the basis of compelling arguments. This option would probably result in minor revisions to the manuscript, like for instance adding a new paragraph that contains the relevant explanations. Personally, I cannot see how this could work, but I may be biased or miss an important point, and I am happy to get involved in a discussion and learn more. As another option I could imagine a statistical/climatological comparison of the TTL diagnostic with an alternative diagnostic like, for instance, the one of Polster and Wirth (2023). This option would make use of most of the work performed in preparing this paper, but of course it would mean a complete rewrite.

I thank the reviewer for his time reviewing this manuscript. I agree that care needs to be taken with interpretation of the TTL definition of waveguides, and I tried to convey this in the manuscript, but I will add in more discussion of some of the points raised by the reviewer. I believe that, whilst the method certainly has limitations (as, I would argue, do all research methods trying to study waves and waveguides in a time-varying and spatially-varying context), its usefulness has not been disproven, and the research I present shows some of the potential uses of this theory in understanding connections between blocking and localized waveguides (stepping away from a zonal mean perspective), and between waveguides and co-located quasi-stationary waves. I believe this research provides a useful step forward, and publication of this paper and dataset will allow the community to further investigate this methodology and the topic of interactions between waves and waveguides.

I agree that there are some advantages to the PV-gradient methodology as raised by the reviewer, however, given that daily zonal wind on upper tropospheric pressure levels is available directly from many CMIP6 climate models (https://github.com/cmip6dr/data_request_snapshots/blob/main/Release/dreqPy/docs/CMIP6_MIP_tables.xlsx) the TTL method provides some advantages for studying future changes, if the methodology is found to be valid and useful. Given this, I believe that continued community investigation, through peer-reviewed papers on the use, and similarities/differences between the two methods of waveguide detection, will be useful in furthering our understanding of this complex topic.

The reviewer has indeed shown, in Wirth (2020) that, under very specific circumstances, with idealized high amplitude perturbations, the zonal mean background flow does not represent a 'true' background flow and waveguides in the zonal mean flow can appear as an artefact of the methodology. However, the research in my paper is an explicit attempt to move away from the zonal mean, partly for this reason. Waves of sufficiently long wavelength may alter the flow such that they then impact waves of higher wavenumber. This, I believe, is suggested by the research in the current paper and is an important result.

I will certainly include more of a comparison with the waveguides of Polster and Wirth (2023), particularly as there are some similarities between these two methods in terms of northern vs southern hemisphere asymmetries in waveguide frequency. Unfortunately Polster and Wirth (2023) only studied wintertime waveguides, and so a comparison across seasons is not possible. I agree that a more complete statistical comparison of these two datasets of waveguide frequency would be valuable, but is outside the scope of this current paper; I will suggest this in the section on future research.

I also agree with the reviewer that some of the results from this work are supportive of the results of Wirth and Polster (2021), namely that strong blocks may create waveguide conditions. The reviewer argues that, when looking at zonal waveguides the causal relationship is from block to waveguide. I agree that this may indeed be the case in some circumstances (although I do not think it has been proven that it is always the case), and I also agree that the composites of zonal wind in this current paper support this idea. I agree that more emphasis should be put on this result, and intend to include composites of geopotential height anomalies, showing that, in some regions, the presence of strong waveguides is associated with a region of enhanced geopotential height **poleward** of the waveguide location, consistent with the zonal wind changes that create the double jet structure shown in the previous composites. This result is consistent with the idea of an atmospheric block helping to create the local double jet which helps create the waveguide conditions. Taking a zonal mean of the zonal composites shown in this paper, would very likely show a double jet structure; however, importantly, the 'double jet' found in these composites is a localized double jet, not one with hemispheric extent, and a hemispheric average would likely be misleading, as stated by Wirth and Polster (2021).

The goal of this study, however, is to look at localized waveguides. Whilst indeed it seems that blocks may well cause an increased probability of a waveguide equatorward of the block, in the final section I show the relationship between **co-located** quasi-stationary waves and waveguides. Because I use the wave envelope of meridional wind to define the wave amplitude, one would expect any wave amplitude associated with the block to be poleward of where the waveguide exists, and thus this cannot be the explanation for the positive correlations I show. The results presented here show that, at some (but likely not all) longitudes, high latitude blocks likely play a role in *creating* the conditions for a waveguide to exist at lower latitudes through their influence on the localized zonal winds. The positive correlations with co-located waveguides suggest that such waveguides may then play a role in trapping quasi-stationary waves of higher wavenumbers at lower latitudes. This may help explain some of the 'double quasi-stationary wave' pattern (waves of different wavenumber seen in different latitudes) seen in several extreme months (e.g. White et al. 2022), and the coincidence between blocks and recurrent Rossby waves (which would show up in the quasi-stationary wave metric used here) found by Mubashshir Ali et al. 2022.

I understand the reviewer's concern about the idea that, as soon as there are two turning latitudes, there is a 'perfect waveguide'. I agree that this perspective would be problematic, but I do not think that this is the only interpretation of the turning latitude theory. Inspired by the reviewer's previous work (Wirth 2020), I chose to define a 'waveguide amplitude' (previously 'waveguide depth') metric to acknowledge that waveguides can exist with a spectrum of strengths, even in the two turning latitude perspective.

Major Comments
The first set of results (section 4.1) presents climatological properties of TTL waveguidability, and it turns out that (broadly speaking) waveguidability is stronger or waveguides are more frequent in summer compared to winter. This is in striking conflict with the fact that jets are generally stronger in winter than 2 in summer, and the latter implies stronger (true) waveguidabiliy in winter compared to summer according to Manola et al. (2013) or Wirth (2020). The author mentions this conflict on line 168 and adds later on line 263 that "further research is required to fully investigate this result". I find this not very satisfying. One way to resolve this conflict would be to admit that TTL-based waveguidability is flawed and inappropriate for a reliable diagnosis in this context, consistent with the arguments of Wirth (2020).

The waveguides depend on the stationary wavenumber, $K_s$, which is related to the 2nd meridional gradient of the zonal wind (see Eqs 1 and 2 in the manuscript) rather than the strength of the zonal wind in the jet (and, indeed, the zonal wind strength appears in the denominator of the $K_s$ equation), and so the width of the jet plays a key, and indeed, perhaps stronger, role relative to the strength of the jet. This point was also highlighted by Manola et al. 2013. I therefore don't believe the results found in this paper are inconsistent with theoretical expectations. Indeed, in Hoskins and Woollings (2015) Fig. 2, a stronger climatological stationary wavenumber can be seen in the summer vs winter seasons, which implies (although the link is not direct) higher waveguide frequency in summer. As suggested by reviewer 1, I will include climatological maps of the stationary wavenumber in a revised manuscript, allowing a more complete discussion of the hemispheric and seasonal differences.

Hoskins, Brian, and Tim Woollings. 2015. "Persistent Extratropical Regimes and Climate Extremes." *Current Climate Change Reports* 1 (3): 115–24. https://doi.org/10.1007/s40641-015-0020-8.

Manola, I., Frank Selten, Hylke de Vries, and Wilco Hazeleger. 2013. "'Waveguidability' of Idealized Jets." *Journal of Geophysical Research: Atmospheres* 118 (18): 10,432-10,440. https://doi.org/10.1002/jgrd.50758.

I also believe there is more of a difference between localized waveguide presence and 'hemispheric waveguidability' than the reviewer is considering here, with the latter 'waveguidability' defined as the ability of a background flow to trap waves within a certain latitude band across the whole longitude range, i.e. considering hemispheric waveguides. Such hemispheric waveguidability is important for circumglobal Rossby waves, but I am interested in whether localized waveguides can be important for localized Rossby wave amplitude. I recognise that this distinction could be made more apparent in the paper, and will adjust the text to do so.

2. Figure 6 of the paper indicates that the "black boxed regions" with strong waveguide occurrence are characterized by weaker than normal zonal wind. Again, this result is in striking conflict with the results of Manola et al. (2013) and Wirth (2020), who showed that strong waveguides are generally associated with stronger rather than weaker jets. As far as I can see, the author does not resolve this conflict. One way to resolve the conflict would be to acknowledge that the TTL-waveguide diagnostic is fraught by the artefact discussed in Wirth and Polster (2021, see my further comments below), and that previous authors who used the TTL-diagnostic are subject to the same artefact.

The winds are only weaker in the northern half of the boxes, and stronger in the south, consistent with enhanced second meridional gradients in zonal wind, which is consistent with the theory of waveguides. The waveguides in these composites are typically in the southern half of the boxes, as shown by the pink contours. To try to make this point clearer, I will replot the figures using waveguide regions with a smaller latitudinal extent, and further emphasise the importance of the meridional gradient over just the maximum strength of the jet.

3. The second part of the paper (section 5) shows, broadly speaking, a (weak, but statistically significant) positive correlation between local Rossby wave amplitude and local TTL-waveguidability in certain regions (Fig. 10). At first sight this result was surprising to me, given that the author does not really give a motivation why one should expect such a correlation (e.g., on lines 98 and line 280ff, where such a connection is more or less assumed to be given). To be sure, in the case of circumglobal waveguides, a motivation might arise from the contested quasi-resonance arguments of Petoukhov et al. (2013). But this line or arguments cannot possibly provide a motivation for the present analysis, because Rossby wave resonance requires circumglobal waveguides, while the author here diagnoses local (even gridpointwise) waveguides. On line 210, the author mentions two papers in which such a correlation allegedly was hypothesized, but as far as I can see the waveguides in these papers were assumed to be circumglobal rather than local. Therefore, I disagree with the author's statement in the discussion section (line 279) that such a connection was hypothesized "to some extent" in previous articles. However, after second thought I realized that this correlation may be a result of the artefact which I discussed in Wirth and Polster (2021). The author herself provides a hint in her Figs. 6 and 7, where she shows that strong TTL-waveguides are associated with tripole-like anomalies in the zonal wind corresponding to a "double jet structure". This result, in combination with the positive correlation between local TTL-waveguidability and Rossby wave amplitude, offers the following interpretation: according to the argument of Wirth and Polster (2021), these strong TTL-waveguides may simply be artefacts arising from strong Rossby wave amplitudes. Large wave amplitudes would distort the total (= background plus wave) flow pattern such that one obtains a tendency towards a double-jet structure in the zonal average (see, e.g., the schematic in Fig. 1 of Wirth and Polster 2021).

To the extent that there is no alternative plausible motivation for the correlations in Fig. 8, my argument suggests that Fig. 8 could actually be 3 interpreted as an independent (data-based) confirmation of the Wirth-Polster criticism. A way to test this hyopthesis would be to use the novel zonally varying background state from Polster and Wirth (2023). In fact, the author mentions this idea on line 293. It would not be too hard to perform this analysis, since Polster

and Wirth published her code along with the paper. The basic state from Polster and Wirth is based on a "rolling zonalized" background field, which is not subject to the Wirth-Polster artefact — in contrast to the background state used in the current paper. If the correlation vanished upon the use of this (presumably more appropriate) basic state, one would have produced an independent piece of evidence for the statement that, indeed, the tripole-structures in Fig. 6 and 7 essentially reflect the artefact discussed in Wirth and Polster (2021).

Waveguides are regions where wave energy is more likely to be trapped within a particular latitude range (e.g. Hoskins and Karoly, 1981; Hoskins and Ambrizzi, 1993). Hoskins and Karoly also propose that waves will be refracted towards regions of higher stationary wavenumber, i.e. towards and into a waveguide. The presence of waveguides is thus indicative of regions where wave energy is more likely to be guided into, and trapped within this latitude. In addition, regions of higher stationary wavenumber would allow higher wavenumber quasi-stationary waves to be present. I will add these arguments into the introduction of the paper, making it clear that the hypothesised connection between waveguides and waves does not require zonally symmetric waveguides in this case, as I am not studying circumglobal waves.

As discussed above (see a more complete response there), in the section you refer to here, I show the relationship between **co-located** quasi-stationary waves and waveguides. Because I use meridional wind to define the waves, one would expect any wave energy associated with blocks that may be helping to create waveguide conditions to be poleward of where the waveguide exists. Thus, I consider the co-located quasi-stationary waves seen in the correlations to not be related to the block itself, and therefore the positive correlations found suggest that waveguides do lead to a higher probability of high amplitude quasi-stationary waves (although the direction of causality is not proven)

It may be the case that a block poleward of the waveguide leads to the changes in zonal wind, which helps create the waveguide conditions, which leads to a higher probability of quasi-stationary waves equatorward of the block itself – this idea will be hypothesised in the revised manuscript, but it is outside the scope of this current paper to prove this connection.

4. In Fig. 4, the author introduces a novel metric for "waveguide depth" Wd. This is an interesting idea, because Wd represents a somewhat more "integral" measure for the strength of a waveguide, in contrast to the search for two turning latitudes; the latter only relies on the intersection of the Ks-profile with a line representing a fixed wavenumber and introduces an artificial "waveguide vs. no-waveguide" dichotomy (Wirth 2020). Fig. 4 in Wirth (2020) suggests that stronger jets are generally associated with stronger Wd as defined here. However, Wd, too, is unlikely to represent the increase of true waveguidability with increasing jet strength beyond a certain limit. This was explicitly discussed in Wirth (2020) in connection with his Fig. 6a, where true waveguidability increases from about 48% to about 75% while jet strength was increased from 20 to 40 m/s, and this increase in true waveguidability would be completely missed by Wd. In fact, I believe there are better (and simpler) measures for waveguide strength, such as the horizontal PV gradient.

Indeed, my metric for waveguide strength (renamed from depth at the suggestion of reviewer 1) was inspired by your work in Wirth (2020). However, I believe that this metric DOES capture some of the increasing waveguidability for stronger/narrower jets. For example, in Fig. 4 in Wirth (2020), the "waveguide strength" (difference between k and maximum Ks in the

waveguide) for a wave of wavenumber 5 increases from around 1.6 for a 10m/s jet to approximately 2.1 for a 40m/s jet, and so this increase in waveguidability is not entirely miseed by this method.

I thank the reviewer for the minor comments, and will address these in a revision.

---

## Author Response (AR1)

I thank the reviewer for their comments and suggestions for improving the paper. I agree with your suggestions, and have implemented most of them. After more analysis based on the comments of other reviewers, the paper now compares Ks and PV-waveguides, and thus some details of the double jets in the KS waveguides have now been removed to make room for the comparison. Reviewer comments are given below in black, with my responses in blue.

Major comments:

a) Abstract: the notion of waveguide depth is obscure there. When I read the abstract I thought it was considering the depth in altitude. It would be good to add words to qualitatively define the waveguide depth. For me, the waveguide depth is closely related to the number of zonal wavenumbers for which the waveguide exists.

Thank you for this helpful feedback – I have adjusted this terminology to be 'waveguide strength'.

b) Introduction (lines 25 to 29). It would be good to add some physical interpretation of why people think there is a link between waves amplitude and waveguides. In my opinion, this is potentially because waves are not dissipated as there are no critical latitudes in a waveguide. But I am not sure this is what the papers cited in line 28 have argued.

Yes, I agree that the lack of dissipation of wave energy within the waveguide, and channeling of that wave energy is at least part of the physical interpretation, along with the typically enhanced stationary wavenumber within a waveguide, allowing waves of higher wavenumber to become quasi-stationary. In addition, if the waveguides are quasi-stationary, this would allow quasi-stationary channeling of wave energy into this particularly latitude band. I have added: "Waveguides provide a region where wave energy is more meridionally confined, and thus wave dissipation may be low; they may therefore provide conditions for high-amplitude quasi-stationary waves to develop." Before the line: "Improved understanding of atmospheric waveguides and their role in the existence and amplification of quasi-stationary waves is therefore of great interest."

c) Method: it would be good to highlight that the background flow and the waves are separated by the spatial scale (k<2 for the background flow and k between 4 and 15 for the quasi-stationary waves). Since the 15-day running mean is used for the detection of both the background flow and the waves, the reader might be confused by the separation between these two parts of the flow.

Thank you, I have added clarification text on this. I have also repeated the quasi-stationary wave analysis using k between 6-15 to provide even greater separation, and results are very similar – this is now added to the sensitivity analysis.

"In this work, both the waveguides and the QSWs employ the same 15-day low-pass filter; the waveguides and waves are therefore separated only by their spatial scale, with waveguides using k<=2, and the QSWs (4<=k<=15). Key results are repeated with the QSWs defined as (6<=k<=15) to increase the degree of separation (see Section 5.1).

d) Section 4.1 and waveguide frequencies:

- before starting that section it would have been nice to show Ks for the time mean flow of JJA and DJF, i.e repeat Figures 3c of Hoskins and Ambrizzi (1993) and 11c of Ambrizzi et al (1995). Maybe it would be good to do it by considering the climatological flow for k<2 to be close to what is done in the present paper for the time-evolving waveguides. Such additional figures could help to better visualize the difference between summer and winter and between SH and NH. The argument made lines 160-165 to qualitatively explain why the summer NH has more frequent waveguides than the winter NH could be better understood by showing the time mean U and Ks for both seasons. Furthermore, maybe the additional argument is the fact that the jet is probably narrower in summer than winter and both the planetary vorticity gradient and relative vorticity gradient play a role in the difference between summer and winter.

This is a nice idea, thank you. Instead of showing Ks for the climatological U, I am showing the climatological median (the mean is too impacted by extreme values) Ks on the time and zonally filtered data. This is slightly different from the Ks of the climatological flow, because Ks is non-linear, but a comparison to the suggested papers can still be made. This plot is now referenced when thinking about the differences in climatological waveguide frequency between different seasons and hemispheres.

- I am surprised that the SH has less waveguides than the NH as the double jet structure (separation between subtropical and eddy-drivent jet) is more marked there, at least in the climatologies.

Yes, I agree that this result is interesting, and is in contrast to the PV-waveguides (now compared directly in the paper) I have added more discussion around this topic in the revised manuscript. It is also worth noting that the waveguide frequencies are giving the frequency of a waveguide existing at that latitude, not the frequency of a waveguide at that longitude in that hemisphere – thus jets with more variable latitude may show a reduced waveguide frequency at any one latitude. There is likely also an influence of differences in the width of the jets between the northern and southern hemisphere, particularly for the KS-waveguides. For example. Manola et al. 2013 (Fig. 5 c and d) show some differences between the jet widths (again, comparing SH summer with NH winter), and in the SH jet widths of up to 16-18 degrees latitude are more common than in the NH. This importance of jet width can also be seen in the new figure you suggested, which I have added below, looking at composites of zonal winds (not just anomalies) for strong vs low/no waveguide days - the differences are often more in the narrowness of the jet, and not always in the maximum strength. Given the shift of the paper towards a comparison of KS and PV-waveguides, this figure is not included in the paper, however the results are used in the discussion. Whilst in idealized situations, one can create stronger meridional gradients by changing the strength of the maximum jet, that is not the only way, and these results suggest, in fact, a stronger role for the narrowness of the jet (as also highlighted in Manola et al. 2013)

e) Section 4.2: this is the part of the results where I am less convinced by the conclusions. For instance, Figure 6a shows an anomalous tripolar pattern in zonal wind when computing the difference between high and low waveguide strengths. This anomaly could be the result of different changes: a more pronounced double jet structure is one possibility but it could result from a widening of the jet or some latitudinal shifts. So it would be very nice to compare composites of high waveguide strengths and low waveguide strengths separately before (or rather than) showing the difference.

Thank you for this suggestion. I have made these plots of the composites (see below). As discussed above, these composites do show a double jet structure in many regions, however this is also related to

a southward shift in the lower latitude jet relative to the no/low waveguide condition. As noted above, however, these plots are not included in the revised manuscript, as they would focus too much on the KS-waveguides. As the double jets are found to be likely related to high latitude blocking in the new composites of geopotential height, we have decided to not focus too heavily on the double jets themselves, and more on the mechanism of whether blocks are causing the waveguide conditions.

[Figure]

f) Section 5: It is suprising that the correlations are strong in the Atlantic and over Asia and not in the Pacific while the waveguide depths are similar in the North Atlantic and North Pacific. What would be a possible explanation for that ? Or if you do not have hypothesis it would be nice to comment these results by referring to other studies. Were the studies on the relationship waveguide-wave amplitude focused in the North Atlantic and Asian regions. Do you know studies that also considered that relationship in the North Pacific ?

I agree that this is a somewhat surprising result, and may perhaps be some combination of the waveguide strength and the location of wave sources; however, we find for the PV-waveguides very different correlation patterns. This could be because the QSWs co-located with KS-waveguides are more to do with low pressure anomalies associated with higher latitude blocking, and thus the spatial distribution of correlations is related to blocking strength and the type of blocks that typically happen in that region. Note that there are still positive correlations across much of the Pacific region in most sensitivity analyses - the correlation is weaker and not statistically significant, however it may be physically real.

g) Sensitivity tests: I think it would be good to have a sensitivity test by changing the mean pressure level (e.g. 500 hPa?). Held et al. (1985) computed a barotropic equivalent level near 425 hPa and Charney (1949, see section 6) found a barotropic equivalent level closer to 550-600 hPa.

This is an interesting idea, thank you – this has been added to the sensitivity tests.

Minor comments:

Thank you for your minor comments. I will add in these suggestions and corrections.

1) Line 125: maybe add "temporally and zonally filtered U following the method described in section 2.1"

Thank you; added.

2) Line 130: it would be good to have a qualitative description of what the waveguide depth means. We understand mathematically in the main text but this would be useful for the abstract.

Thank you – this is now added in the introduction: "in this study we extend the binary approach, using the stationary wavenumber to define a metric of `waveguide strength', allowing a continuous range of `waveguidability' for the KS-waveguides, once the threshold of waveguide presence is reached."

3) Line 134 and thereafter: why is cut-off latitude used rather than turning latitude ? I think turning latitude is the classical term

Thanks for noting the confusion – here I intend to refer to the latitude at which the algorithm starts looking for waveguides, and so use a different term to differentiate from the turning latitudes. I have re-phrased to make this clearer.

4) Figure 2: what are the black contours. Zonal wind at 300 mb ? Same question for Figure 4 but they disappear in Figure 5.

Thanks for noting these missing pieces. You're correct that they were the (zonally filtered) seasonal mean zonal winds – these figures have now been altered, but the zonal winds in what is now Fig. 3 are explained in the caption.

5) Line 178: I do not understand the end of the sentence "latitudinal cut-off ... latitude of the jet".

This was in reference to the lowest latitude that the algorithm looks for turning latitudes. This discussion has been mostly removed from the revised manuscript.

6) Lines 187-188: I do not understand what is meant by "Latitudes are weighted equally". Do you mean that multiplication by the cosine of latitude is applied to do the regional averages?

This has been amended, and all figures are now shown with latitude-weighting, noted in the 'area-weighted regional averages'

7) Line 196: I would add "a 'double jet structure' in anomalies is present

This has been reworded: "In all regions and seasons analysed in Fig. 7 a consistent anomalous jet structure is present at the longitude of the waveguide, with a region of anomalously low zonal wind immediately to the north of the region of high waveguide strength. Typically there are also either anomalously strong zonal winds at, or just south of, the waveguide location, and/or poleward of region of low zonal wind."

8) Lines 204-206: here again it would be better to see both composites rather the difference between composites

Because of the shift towards the PV- and KS-waveguide comparison, we continue to use differences to be more concise. For U and Z500 I now use differences between strong waveguide days and climatology, so this may be easier to interpret.

9) Figure 7: the magenta contour is difficult to see in the red-brown areas.

Thanks for this comment – I have made the magenta contour thicker in all plots so it is easier to see.

10) Lines 260-265: Here the importance of narrow jet is highlighted but this has not been the main argument mentioned in the main text when describing the difference between summer and winter in the NH (lines 160-165). The story of the equatorward displacement of the jet was emphasized. So please be more precise why there is a difference between summer and winter. How important are the jet width and latitude in that story ?

In lines 160-165 I was trying to explain why the peak in the waveguide is displaced equatorward of the peak in the jet, and not the difference between summer and winter. I believe the difference between summer and winter is indeed to do with jet width (now also seen in the Ks figure you suggested). I have re-phrased these sections to hopefully be less confusing.

11) Caption Figure 9: please provide units for dimensional parameters.

Thank you. This figure and caption have been substantially changed; all dimensional parameters in the caption have units now.

Reviewer 2

I thank the reviewer for his time reviewing the manuscript, and the helpful comments and suggestions that have substantially improved the manuscript. The additional analysis performed in response to the reviews led us to, as suggested, include a comparison of the two types of waveguide definition (the 'two-turning-latitude' approach, or KS-waveguides, and the PV-gradient approach, following Polster and Wirth 2023). We now show that some of the problems the reviewer hypothesised regarding the 'two-turning-latitude problem' are likely correct – these issues are explored further in new plots looking at composites of geopotential heights and QSWs for the two waveguide types. Ultimately, we end up recommending the PV-waveguides for future research. This recommendation is further supported by the finding that the correlations with co-located QSWs are much higher and widespread with the PV-waveguides than with the Ks-waveguides. I have copied below the comments of the reviewer in black, along with my responses in blue.

The concept of a Rossby waveguide has recently found increased interest. This recent interest may be partly due to the hypothesis of Petoukhov et al. (2013), who considered circumglobal waveguides and investigated the possibility of Rossby wave resonance during specific episodes. Their method to diagnose the existence of a waveguide from observations was based on arguments used earlier by Hoskins and Karoly (1981) and Hoskins and Ambrizzi (1993), which in turn start with linear wave theory, make additional assumptions (like the so-called WKB approximation), and finally arrive at the concept of a refractive index. Under special circumstances, the basic state may feature two so-called turning latitudes for a given wavenumber, and this is then interpreted as a perfect zonal waveguide for the respective wavenumber. In the remainder of this text I will refer to this framework with the "Two Turning Latitudes" as TTL analysis or TTL thinking. The current paper has, broadly speaking, two parts. In the first part the author produces and discusses, for the first time, a climatology for waveguide occurrence based on TTL analysis. As a particular feature the author considers a background atmosphere that is allowed to have a smooth variation in longitude, such that her waveguides, too, include a smooth variation in longitude. In the second part, the author goes on and correlates waveguidability as diagnosed from TTL analysis with Rossby wave amplitude, where both waveguidability and wave amplitude are allowed to vary smoothly with longitude. In both parts, the background atmosphere (needed to define a waveguide) is obtained through a combination of temporal and spatial filtering.

In the past, I have raised two major issues with the TTL analysis as a method to diagnose waveguides and waveguidability and how it is usually applied. First, in an idealized modeling framework I designed a method to diagnose "true" waveguidability and compare it with TTL-based waveguidability (Wirth, 2020). My "true waveguidability diagnostic" does not make use of the WKB assumption, which underlies the TTL analysis but which is badly violated in realistic situations. Therefore, in case of discrepancies between the two methods, my "true waveguidability" would naturally be given priority. As it turns out, TTL-based waveguidability is severly flawed in that it is unable to reproduce the gradual increase in true waveguidability as the strength or the narrowness of a jet is increased (see also Manola et al. 2013). In particular, the association of the existence of two turning latitudes with a (perfect!) waveguide for the corresponding zonal wavenumber was shown to be highly problematic. My second issue concerns the fact that the TTL-based analysis may be subject to artefacts in the event of large wave amplitudes, if the used background state is based on zonal averaging (Wirth and Polster 2021).

I do not contend that these two issues reduce the utility of TTL-theory to zero, but I would certainly say that TTL-theory is unable to represent certain (possibly important) aspects regarding waveguides and waveguidability. Unfortunately, we do not have a satisfying understanding yet that would allow one to distinguish those aspects for which the application of TTL-theory is appropriate from those for which it is not. This situation calls for a high level of care that needs to be exercised. Incidentally, as far as I can see,

earlier papers such as Hoskins and Karoly (1981) and Hoskins and Ambrizzi (1993) never based their conclusions on TTL analysis alone; rather, their work typically included some independent statistical analysis and/or numerical modelling, and whenever all these approaches yielded consistent results, the TTL-diagnostic was used for interpretative purposes.

Despite the issues that I raised above, the TTL analysis of waveguides and waveguidability enjoys ongoing popularity, and the caveats associated with this approach are sometimes simply ignored. The current paper seems to continue along this tradition. To be sure, the author quotes all the relevant papers including the two critical ones I just mentioned. She even says explicitly that caution is needed when interpreting results that are based on TTL theory (line 90). But then she simply goes on and does not really attempt to critically discuss the implications of these papers for her own work. In my eyes this is not satisfactory. I would expect that the author gives strong arguments why she thinks that the issues I raised are irrelevant in the context of her paper. Otherwise, the reader is left alone with the question: "why learn more about a diagnostic that was shown to be severely flawed in relevant applications"?

Interestingly, when I read the paper for a second time, it occurred to me that the results from both parts can actually be interpreted in a way that further supports the criticism raised by Wirth (2020) and Wirth and Polster (2021) — see my further explanations in the major issues below. I believe that such an interpretation was not the original intention of the author.

Given this situation, I am not sure what to recommend. One option might be that the author is able to argue in favor of TTL thinking and dismiss my issues on the basis of compelling arguments. This option would probably result in minor revisions to the manuscript, like for instance adding a new paragraph that contains the relevant explanations. Personally, I cannot see how this could work, but I may be biased or miss an important point, and I am happy to get involved in a discussion and learn more. As another option I could imagine a statistical/climatological comparison of the TTL diagnostic with an alternative diagnostic like, for instance, the one of Polster and Wirth (2023). This option would make use of most of the work performed in preparing this paper, but of course it would mean a complete rewrite.

Thank you for your comments and suggestions. Having completed further analysis of the composites of the Ks-waveguides, based on the comments of your and other reviewers, the work in this paper now confirms that there is a connection between high pressures/atmospheric blocking poleward of the waveguide and the existence of KS-waveguides, as suggested by the reviewer, and hinted at in the previous results. It is likely (although not proven here) that this relationship is in the direction of the atmospheric blocking causing the waveguide conditions, as shown in the zonal mean perspective in Wirth (2021). Given this, I have now included a comparison with the PV-gradient waveguides as calculated using the method of Polster and Wirth (2023). The main structure of the paper remains the same, but the figures now show comparisons between the KS-waveguides and the PV-waveguides (for summer), rather than summer vs winter; plots for winter are shown in the Appendix.

I have also added more explicit information about the limitations of the KS-waveguide method, including: "The theory may therefore provide some use in understanding underlying mechanisms; however, care should be taken not to over-interpret results in a quantitative manner. One key limitation, shown clearly by Wirth (2020), is when the theory is interpreted in a binary manner: either a waveguide exists and waves are 100% trapped within the waveguide, or there is no waveguide and 0% wave confinement; in reality a range of `waveguidability' exists depending on the strength of the meridional gradients within the jet. Motivated by this, I extend the binary approach, and use the

stationary wavenumber to define a metric of `waveguide strength', allowing a continuous range of `waveguidability'."

The reviewer has indeed shown, in Wirth (2020) that, under specific circumstances, with idealized high amplitude perturbations, the zonal mean background flow does not represent a 'true' background flow and waveguides in the zonal mean flow can appear as an artefact of the methodology. The research in this current paper is an explicit attempt to move away from the zonal mean, partly for this reason. Waves of sufficiently long wavelength may alter the flow such that they then impact waves of higher wavenumber. This, I believe, is suggested by the research in the current paper (mostly by the KS-waveguides) and could be an important result – certainly worthy of further investigation.

Overall, the paper now compares the two waveguide approaches, showing that the PV-waveguides tend to be: a. less likely to be associated with local higher latitude blocking, and b. more strongly correlated with co-located QSWs. We therefore recommend the rolling-zonalization PV-waveguides of Polster and Wirth (2023) for future studies of waveguides.

Major Comments
The first set of results (section 4.1) presents climatological properties of TTL waveguidability, and it turns out that (broadly speaking) waveguidability is stronger or waveguides are more frequent in summer compared to winter. This is in striking conflict with the fact that jets are generally stronger in winter than 2 in summer, and the latter implies stronger (true) waveguidabiliy in winter compared to summer according to Manola et al. (2013) or Wirth (2020). The author mentions this conflict on line 168 and adds later on line 263 that "further research is required to fully investigate this result". I find this not very satisfying. One way to resolve this conflict would be to admit that TTL-based waveguidability is flawed and inappropriate for a reliable diagnosis in this context, consistent with the arguments of Wirth (2020).

The waveguides depend on the stationary wavenumber, Ks, which is related to the 2$^{nd}$ meridional gradient of the zonal wind (see Eqs 1 and 2 in the manuscript) rather than the strength of the zonal wind in the jet (and, indeed, the zonal wind strength appears in the denominator of the Ks equation), and so the width of the jet plays a key, and indeed, perhaps stronger, role relative to the strength of the jet. This point was also highlighted by Manola et al. 2013. I therefore don't believe the results found in this paper are inconsistent with theoretical expectations. Indeed, in Hoskins and Woollings (2015) Fig. 2, a stronger climatological stationary wavenumber can be seen in the summer vs winter seasons, which implies (although the link is not direct) higher waveguide frequency in summer. As suggested by reviewer 1, I now include climatological maps of the stationary wavenumber in a revised manuscript, allowing a more complete discussion of the hemispheric and seasonal differences. It is interesting, however, that the PV-waveguides frequency (now also shown in Fig. 3) is higher in winter, and more equally distributed between the hemispheres. We do not consider this necessarily to be a weakness of the KS-waveguides, however, as the results are consistent with the mathematics, that the zonal wind strength appears in the denominator of the Ks equation. I also believe there is more of a difference between localized waveguide presence and 'hemispheric waveguidability' than the reviewer is considering here, with the latter 'waveguidability' defined as the ability of a background flow to trap waves within a certain latitude band across the whole longitude range, i.e. considering hemispheric waveguides. Such hemispheric waveguidability is important for circumglobal Rossby waves, but here we are interested in whether localized waveguides can be important for localized Rossby wave amplitude, a hypothesis which seems to be confirmed in Fig 10, with positive correlations between waveguide strength and co-located QSW activity, particularly for the PV-waveguides. Confirming the seasonality

and hemispheric variations of 'true waveguidability' from observations will be confounded by seasonal and hemispheric variations in the zonal continuity of the waveguides as well as wave sources. We now discuss this: "Some of these differences, such as seasonal and hemispheric variations can be at least partially explained by the different formulations of the waveguide definition, with the strength of the zonal wind U appearing in the denominator of KS-waveguides, but not for PV-waveguides. It is unclear which is the more `accurate' description, as so many other factors, including the continuity of waveguide conditions along a longitude circle, and strength of local waviness, may influence observed hemispheric and seasonal variations in wave strength within waveguides. Idealized simulations such as those performed by Segalini et al. 2024, analysing how waves behave with different background flows, could help understand these differences, and confirm the relative importance of jet width vs jet strength."

Hoskins, Brian, and Tim Woollings. 2015. "Persistent Extratropical Regimes and Climate Extremes." *Current Climate Change Reports* 1 (3): 115–24. https://doi.org/10.1007/s40641-015-0020-8.

Manola, I., Frank Selten, Hylke de Vries, and Wilco Hazeleger. 2013. "'Waveguidability' of Idealized Jets." *Journal of Geophysical Research: Atmospheres* 118 (18): 10,432-10,440. https://doi.org/10.1002/jgrd.50758.

Segalini, A., Riboldi, J.,Wirth, V., and Messori, G. 2024. A linear assessment of barotropic Rossby wave propagation in different background flow configurations, Weather and Climate Dynamics, 5, 997–1012, https://doi.org/10.5194/wcd-5-997-2024

2. Figure 6 of the paper indicates that the "black boxed regions" with strong waveguide occurrence are characterized by weaker than normal zonal wind. Again, this result is in striking conflict with the results of Manola et al. (2013) and Wirth (2020), who showed that strong waveguides are generally associated with stronger rather than weaker jets. As far as I can see, the author does not resolve this conflict. One way to resolve the conflict would be to acknowledge that the TTL-waveguide diagnostic is fraught by the artefact discussed in Wirth and Polster (2021, see my further comments below), and that previous authors who used the TTL-diagnostic are subject to the same artefact.

The winds are only weaker in the northern half of the boxes, and stronger in the south, consistent with enhanced second meridional gradients in zonal wind, which is consistent with the theory of waveguides. The waveguides in these composites are typically in the southern half of the boxes, as shown by the pink contours. To try to make this point clearer, I have recreated this analysis with a smaller latitudinal extent for the waveguide region so the stronger waveguides more accurately match with the black boxes. At the suggestion of reviewer 1, I have also made the magenta contours, showing the waveguide strength anomalies, thicker and thus more obvious. Figs. 4 and 5 now typically show enhanced winds in the waveguide region, but we agree that the decrease in zonal winds poleward is the more prominent feature – this, and its connection to high latitude blocking, is now discussed explicitly in the paper, with new figures (6 and 7) showing the high pressures in the composites consistent with high latitude blocking creating the zonal wind anomalies that create the waveguide conditions.

3. The second part of the paper (section 5) shows, broadly speaking, a (weak, but statistically significant) positive correlation between local Rossby wave amplitude and local TTL-waveguidability in certain regions (Fig. 10). At first sight this result was surprising to me, given that the author does not really give a motivation why one should expect such a correlation (e.g., on lines 98 and line 280ff, where such a

connection is more or less assumed to be given). To be sure, in the case of circumglobal waveguides, a motivation might arise from the contested quasi-resonance arguments of Petoukhov et al. (2013). But this line or arguments cannot possibly provide a motivation for the present analysis, because Rossby wave resonance requires circumglobal waveguides, while the author here diagnoses local (even gridpointwise) waveguides. On line 210, the author mentions two papers in which such a correlation allegedly was hypothesized, but as far as I can see the waveguides in these papers were assumed to be circumglobal rather than local. Therefore, I disagree with the author's statement in the discussion section (line 279) that such a connection was hypothesized "to some extent" in previous articles. However, after second thought I realized that this correlation may be a result of the artefact which I discussed in Wirth and Polster (2021). The author herself provides a hint in her Figs. 6 and 7, where she shows that strong TTL-waveguides are associated with tripole-like anomalies in the zonal wind corresponding to a "double jet structure". This result, in combination with the positive correlation between local TTL-waveguidability and Rossby wave amplitude, offers the following interpretation: according to the argument of Wirth and Polster (2021), these strong TTL-waveguides may simply be artefacts arising from strong Rossby wave amplitudes. Large wave amplitudes would distort the total (= background plus wave) flow pattern such that one obtains a tendency towards a double-jet structure in the zonal average (see, e.g., the schematic in Fig. 1 of Wirth and Polster 2021).

To the extent that there is no alternative plausible motivation for the correlations in Fig. 8, my argument suggests that Fig. 8 could actually be 3 interpreted as an independent (data-based) confirmation of the Wirth-Polster criticism. A way to test this hyopthesis would be to use the novel zonally varying background state from Polster and Wirth (2023). In fact, the author mentions this idea on line 293. It would not be too hard to perform this analysis, since Polster and Wirth published her code along with the paper. The basic state from Polster and Wirth is based on a "rolling zonalized" background field, which is not subject to the Wirth-Polster artefact — in contrast to the background state used in the current paper. If the correlation vanished upon the use of this (presumably more appropriate) basic state, one would have produced an independent piece of evidence for the statement that, indeed, the tripole-structures in Fig. 6 and 7 essentially reflect the artefact discussed in Wirth and Polster (2021).

Waveguides are regions where wave energy is more likely to be trapped within a particular latitude range (e.g. Hoskins and Karoly, 1981; Hoskins and Ambrizzi, 1993). Hoskins and Karoly (1981) also propose that waves will be refracted towards regions of higher stationary wavenumber, i.e. towards and into a waveguide. The presence of waveguides is thus indicative of regions where wave energy is more likely to be guided into, and trapped within this latitude. In addition, regions of higher stationary wavenumber would allow higher wavenumber quasi-stationary waves to be present. I will add these arguments into the introduction of the paper, making it clear that the hypothesised connection between waveguides and waves does not require zonally symmetric waveguides in this case, as we are not studying circumglobal waves.

This correlation between QSWs and waveguides is now shown for both KS- and PV-waveguides, and in fact the correlations are stronger for the PV-waveguides. This is an important result of this revised paper, as it seems to confirm the hypothesis that local waveguides can provide the conditions for amplified QSWs.

For the KS-waveguides, it may be the case that a block poleward of the waveguide leads to the changes in zonal wind, which helps create the waveguide conditions, which then leads to a higher probability of quasi-stationary waves equatorward of the block itself, or it may be that the co-located QSWs in this case are just the low pressure systems associated with the higher latitude block, in the case of Rex,

Omega blocks or Rossby wave breaking. These ideas are now hypothesised in the revised manuscript, but it is outside the scope of this current paper to prove this connection.

4. In Fig. 4, the author introduces a novel metric for "waveguide depth" Wd. This is an interesting idea, because Wd represents a somewhat more "integral" measure for the strength of a waveguide, in contrast to the search for two turning latitudes; the latter only relies on the intersection of the Ks-profile with a line representing a fixed wavenumber and introduces an artificial "waveguide vs. no-waveguide" dichotomy (Wirth 2020). Fig. 4 in Wirth (2020) suggests that stronger jets are generally associated with stronger Wd as defined here. However, Wd, too, is unlikely to represent the increase of true waveguidability with increasing jet strength beyond a certain limit. This was explicitly discussed in Wirth (2020) in connection with his Fig. 6a, where true waveguidability increases from about 48% to about 75% while jet strength was increased from 20 to 40 m/s, and this increase in true waveguidability would be completely missed by Wd. In fact, I believe there are better (and simpler) measures for waveguide strength, such as the horizontal PV gradient.

Indeed, my metric for waveguide strength (renamed from depth at the suggestion of reviewer 1) was inspired by your work in Wirth (2020). However, I believe that this metric does capture some of the increasing waveguidability for stronger/narrower jets. For example, in Fig. 4 in Wirth (2020), the "waveguide strength" (difference between k and maximum Ks in the waveguide) for a wave of wavenumber 5 increases from around 1.6 for a 10m/s jet to approximately 2.1 for a 40m/s jet, and so this increase in waveguidability is not entirely miseed by this method.

Minor Comments
1. Line 15 :. . . it can be associated with extreme weather occasionally, but certainly not always.
Yes, I agree. Reworded to: The circulation associated with large scale atmospheric Rossby waves (Rossby 1939) can have a strong influence on the weather we experience at the Earth's surface, particularly in the extra-tropics. Indeed, high-amplitude Rossby waves can be associated with extreme weather….

2. Line 20: I assume that the issue with stationarity is just as severe in connection with precipitation as it is in connection with heat. E.g., the flooding events in Germany (2002), Pakistan (2010), and Germany (2021) were associated with quasi-stationary circulation patterns.
Good point; the specific reference to temperature has been removed.

3. Line 25: are you here referring mostly to circumglobal jets? Line 209/210: again, why should a strong local waveguide be associated with strong wave amplitude? Petoukhov et al (2013) hypothesize such a connection in the case of circumglobal waveguides, but you have a very local (grid-point wise) perspective on waveguides.
Added: "Waveguides provide a region where wave energy is more meridionally confined, and thus wave dissipation may be low; they may therefore provide conditions for high-amplitude quasi-stationary waves to develop."

4. Line 43: ". . . . theory provides qualitatively useful insights. . . .": how is this possible, if the underlying assumptions are not valid? Is this by pure chance? How about the issues in Wirth (2020), who showed that there are relevant aspects, in which TTL-theory does not provide even qualitatively realistic results? If the theory provides sometimes useful results and sometimes not: how can one distinguish between these two alternatives?

The paper now shows that the PV-waveguides are likely better than the KS-waveguides for studying connections with waves. However, I generally believe that, even if underlying assumptions are not 100% valid, is does not mean that the theory has no use at all, it does mean it should be used with care. However, as we now have a, seemingly better, alternative in the rolling-zonalized PV-waveguides, I think this is a useful comparison to have. This section has now been re-written as:

Despite the limitations of the theory behind KS-waveguides (Wirth 2020), including questionable validity of the underlying assumptions (limitations articulated clearly in the original papers), the theory has previously provided qualitatively useful insights into the behaviour of waves in both idealized simulations and with realistic flow conditions (Hoskins and Karoly 1981, Hoskins and Ambrizzi 1993, Hsu and Lin 1992, Hoskins and Woollings 2015, White et al. 2017). Here, I compare waveguides created using each of these methods, referring to them as `PV-waveguides' and `KS-waveguides'.

5. Line 88: Here you seem to refer to Wirth and Polster (2021), not to Wirth (2020).
Thank you, corrected.

6. Line 94: to some people, a "nonlinear wave" is an oxymoron. I would prefer to speak about "nonlinear eddies" or "nonlinear perturbations" and reserve the term "waves" for linear dynamics.
To me, non-linear wave helps clarify that the perturbation started off as a (near) linear wave, and has grown non-linearly; however, I see your point. Happy to rephrase to non-linear perturbations, here and in the discussion.

7. Line 98: . . . . but this is true only if the waveguides are circuglobal!
As now discussed in the introduction, the reduced dissipation of wave energy within waveguides could potentially lead to enhanced QSWs, even locally. The start of this paragraph has, however, now been rephrased to read:
"Quasi-stationary waves (QSWs) can lead to extreme weather, and thus potential connections to atmospheric waveguides are worth exploring."
As we find strong correlations between QSWs and PV-waveguides, this provides some confirmation that this hypothesised connection may be real.

8. Line 100: a running temporal mean?
Yes, now clarified, thank you.

9. Line 101: do you really mean "planetary wavenumbers of interest" . . . .? It appears to me that you are, rather, aiming to extract synoptic-scale wavenumbers here.
Yes, good point, thank you. Corrected to specify synoptic-scale wavenumbers.

10. Line 103: "wave envelope": Do you really mean the wave envelope of the planetary waves?
With the previous clarification earlier in this sentence that we are isolating the synoptic-scale waves, hopefully this is now clear that it is the wave envelope of the synoptic scale quasi-stationary waves.

11. Line 103: "15-day running mean": yet another temporal filter? Haven't the data already been filtered temporally (line 100)?
The original text was giving an overview, with the following text clarifying the exact specifications of the filters used, but I see this was a little confusing. This has been re-phrased:

12. Line 104: the Hilbert transform is usually applied to compute the wave amplitude, not the wave itself. What do you mean here?
This sentence is now removed to remove the repetition in this section.

13. Line 132: A westerly jet with strength 0.5 m/s is not very impressive in my eyes, and somehow conflicts with the idea of Manola et al (2013) that a jet needs to be both narrow and strong to be a good waveguide. In addition, the desired narrowness of the jet would suggest rather a criterion that restricts to a maximum width (rather than a minimum width, as you suggest).
These are just criteria on the detected waveguides, not ways to detect waveguides. The waveguides are still detected by the presence of turning latitudes, which will be impacted by the narrowness of the jet. The >0.5m/s criterium simply requires winds to be westerly within the waveguide region, i.e. it excludes regions of easterly winds, requiring that strong meridional wind gradients are not created by a gradient between easterlies and weak westerlies, but rather between a strong westerly jet. The 0.5m/s wind criteria is likely only applicable at the edges of the waveguide, not in the centre. This has been re-phrased to:
"In the main results the following thresholds are used, with waveguides removed from the dataset if they do not meet all criteria…. results are found to be insensitive to changes in these thresholds of up to 50%.

14. Line 135: . . . show that the results... "
Corrected, thanks.

15. Line 145/146: To me it seems as if you identify frequent waveguide occurrence with large waveguide amplitude, which I think is dangerous. Waveguide frequency and waveguide strength should be distinguished and not mixed together.
In the waveguide frequency section, I am identifying waveguide occurrence as any waveguide, regardless of strength. In the composites subsection I am presenting composites of days with high waveguide strength, and so this is isolating strong waveguides over a particular region.
This section has now been re-written, as the maps of climatological K_S are included, so the corresponding text has been removed.

16. Figure 2: Can you explain the solid contours in the figure caption!?
Yes, thank you, added to the captions.

17. Line 153: You find more waveguides in summer compared to winter, although the jet in summer is usually weaker. Isn't this inconsistent with the results of Manola et al (2013) and Wirth (2020), who show that stronger jets are usually better waveguides, hence one would expect higher waveguide frequency in winter than in summer?! Can you resolve this issue? A similar problem appears on line 167/168 when comparing Northern and Southern Hemispheres.
The waveguides depend on the 2nd meridional gradient of the winds, not the maximum strength, and so the width of the jet plays a key role, as also highlighted by Manola et al. 2013. I therefore don't believe the results found in this paper are necessarily inconsistent. Additionally, Manola et al. 2013 (Fig. 5) also show some differences between the jet widths (comparing SH summer with NH winter), and in the SH jet widths of up to 16-18 degrees latitude are more common. I therefore believe that the results found by the KS-waveguide metric highlight the importance of the narrowness of the jet, in addition to the strength. Further research on this is needed. This is now clarified in the revised paper as:
"The results in this paper highlight some strong differences between temporally and zonally varying KS-waveguides and PV-waveguides. Some of these differences, such as seasonal and hemispheric variations

can be at least partially explained by the different formulations of the waveguide definition, with the strength of the zonal wind $U$ appearing in the denominator of KS-waveguides, but not for PV-waveguides. It is unclear which is the more `accurate' description, as so many other factors, including the continuity of waveguide conditions along a longitude circle, and strength of local waviness, may influence observed wave strength within waveguides. Idealized simulations such as those performed by Segalini et al. (2024), analysing how waves behave with different background flows, could help understand these differences, specifically confirming the relative importance of jet width vs jet strength."

18. Line 170 and following: here you show that your results are consistent with the formula from the theory that you apply, but at the same time they are inconsistent with results from Manola et al. (2013) or Wirth (2020). Does this mean that you trust the TTL-theory more than the results of Manola and Wirth? That seems dangerous, because the latter do not rely on this (somewhat questionable) theory. As discussed above, I don't believe these results are necessarily inconsistent with the results of Manola et al. When they looked at waveguidability, they also took into account the continuity of waveguide conditions around a longitude circle, which was part of the reason for studying the southern hemisphere jet. Here, we are studying regional waveguides. Indeed, Manola et al. do also highlight the importance of jet width, which is what we highlight here. For me, it remains unclear which (Ks or PV gradients) provides a more accurate description of the waveguidability – I think experiments such as those by Segalini et al. 2024 would be illuminating, and this is now suggested in the discussion section. However, what is clear is that the background flow is better separated using the rolling-zonalization method, and thus, until there is a way to calculate Ks-waveguides on a rolling-zonalized flow, we recommend the PV waveguides.

19. Line 179: similar as above, you find stronger waveguide "depths" in summer compared to winter, although other work suggests that weaker jets in summer should be weaker waveguides. Can you resolve?
Again, whilst the jets are weaker in the summer, the stationary wavenumber, KS is not, as seen in the new Fig. 1 (and in Hoskins and Woollings, 2015). See response to above comment.

20. Line 210: Well, that's not quite right. Petoukhov et al. hypothesize such a correlation only for circumglobal waveguides (which is needed for Rossby wave resonance), they do not hypothesize such a correlation between local wave amplitude and local waveguidability.
Thanks for noting this. I have now re-written the hypothesis between waveguides and QSWs in the introduction, and here it simply reads: "In this section we explore the relationship between waveguides and QSWs." Interesintly, however, the relationship between QSWs and waveguides is stronger for the PV-waveguides (see new Fig. 10).

21. Line 260-265: here you seem to play down the conflict between your results and those of the literature; I would expect a lucid discussion and explanation how you think that these discrepancies can be "explained"
Now that the PV-waveguide comparison has been included, there is a more in-depth discussion about the differences between the two datasets, and what might be causing that. I do not remain certain that the KS-waveguide strength is incorrect, relative to the PV-waveguide strength, and suggest further work that may help understand this in the discussion section.

22. Lines 280-290: you mention a few results from the literature, but it did not become clear to me how they relate to your results. In particular, it did not become clear to me how you would address the

criticism formulated in some of these papers. As I argue in the first part of my review, I believe that some of your results even provide additional support for some of the criticism formulated.

This section has been substantially re-written to include the results of the comparison between the two waveguide datasets.

23. Line 296: Can you explain how your data set potentially can shed light on causality?

This section has been re-written following the inclusion of the PV-waveguides, but a similar section now reads: "Studies of lag-lead correlations, or using causal inference (see, e.g., Kretschmer et al. 2021), would be valuable in illuminating the direction of any causality between QSWs and both KS- and PV-waveguides."

Reviewer 3

I thank the reviewer for their time reviewing the manuscript, and their helpful comments that have led to substantial improvements to the paper. After more analysis based on the comments of this review and other reviewers, the paper now compares Ks and PV-waveguides, an extension which I feel provides a significant improvement. Below I have copied the comments of the reviewer in black, with my responses in blue.

The atmospheric waveguide has a profound influence on the propagation path of stationary Rossby waves, thereby affecting when and where these waves impact surface weather and climate. In recent years, studies on the atmospheric waveguide have gained popularity among the climate community due to its significant connection to extreme events. The investigation of waveguides can be traced back to early works, notably Hoskins and Karoly (1981), followed by Hoskins and Ambrizzi (1993) and Ambrizzi et al. (1995). This current study aims to extend previous research by examining the waveguide in the context of spatially and temporally varying mean flow. Most of the analysis is focusd on this issue.

This is a nice manuscript that uses refractive index as a perspective to understand atmospheric waveguide and its connection to stationary Rossby waves. In my opinion, it holds the potential to be considered for publication in a WCD. However, I have several major concerns about the methodology and interpretation of the results. I have listed my major comments below and would like to invite the authors to address them:

1. The separation of mean flow and perturbations is always a controversial issue when studying wave-mean flow interactions. This issue becomes even more critical when large-amplitude eddies appear in the mean flow (e.g., Wirth and Polster, 2021). However, the present study heavily relies on the separation method, and most of the findings are based on the assumption that the waveguide and Rossby waves are well-separated and independent. Therefore, I question the significance of the results, as many intraseasonal waveguide behaviors are actually reflected by long-lasting waves.

This is a very valid concern, and the revised manuscript now illuminates some of the potential problems of this separation method more clearly, with the comparison to PV-waveguides. I agree that long-lasting waves can influence the waveguides, and this is part of the motivation for this work, to look at longitudinal variations in waveguides. Further analysis looking at composites of geopotential heights (Figs. 6 and 7) now show that KS-waveguides in some regions do indeed seem to be related to the presence of a higher latitude blocking high. This may also be influencing the co-located correlations

between the KS-waveguides and the QSWs, which is now discussed in the Discussion section. Notably, the PV-waveguides also show positive correlations with QSWs, and these are mostly stronger than those found for the KS-waveguides.

2. Regarding the methodology, using the traditional turning point perspective to identify the waveguide could be misleading, despite its extensive use in recent studies such as Petoukhov et al. (2013) and many subsequent papers. The limitations of this method have been thoroughly discussed by Wirth (2020). Therefore, the authors need to demonstrate the limitation of the method used here is nontrivial and confirm the appropriateness of the method.

I agree that care needs to be taken with interpretation of the TTL definition of waveguides, and the revised manuscript provides a comparison of the KS- and PV-waveguides, and in fact now recommends the PV-waveguides, although we believe there may be some interesting results illuminated by the KS-waveguides that may be worthy of further study. We have now also added in more discussion of some of the points raised by the reviewer regarding the KS-waveguide limitations, e.g.:
"The theory may therefore provide some use in understanding underlying mechanisms; however, care should be taken not to over-interpret results in a quantitative manner. One key limitation, shown clearly by Wirth (2020), is when the theory is interpreted in a binary manner: either a waveguide exists and waves are 100% trapped within the waveguide, or there is no waveguide and 0% wave confinement; in reality a range of `waveguidability' exists depending on the strength of the meridional gradients within the jet. Motivated by this, I extend the binary approach, and use the stationary wavenumber to define a metric of `waveguide strength', allowing a continuous range of `waveguidability'."

3. I doubt about the characterization of the atmospheric circulation associated with the waveguide strength as "double jet streams" (Figure 6), as the zonal wind anomalies are only confined to a local scale. Additionally, as related to my major comment 1, long-lasting waves might play a role in this structure. Therefore, it is possible that the pattern seen in Figure 6 is not "double jet streams", but prominant Rossby wave activity itself.

This is an excellent point, and is now explored in more detail in the composites of geopotential height, and the comparison between KS- and PV-waveguides. Indeed, we find no such double jet anomalies for the PV-waveguides, and the geopotential height composites suggest that presence of atmospheric blocks at high latitudes, creating the conditions for the KS-waveguides. This supports the argument made by the reviewer, and by Wirth and Polster (2021), which is now highlighted in the paper, including in the conclusions: "For KS-waveguides only, a double jet structure is found to be associated with strong waveguide days, particularly over the North Atlantic and European regions. Further analysis suggests this is likely at least partially related to atmospheric blocking, with blocking conditions leading to the jet anomalies that create the KS-waveguide conditions. Such an association does not occur for PV-waveguides."

4. The current study primarily focuses on the waveguide effect along the subtropical jet, based on the refractive index, which essentially represents the gradient of absolute vorticity. However, recent studies (e.g., Xu et al. 2019, doi: 10.1175/JCLI-D-18-0343.1; Xu et al., 2020, doi: 10.1175/JCLI-D-19-0458.1) have presented compelling evidence of the existence of stationary Rossby waves along the eddy-driven jet, where the waveguide effect arises due to the gradient of potential vorticity. As mentioned by the author

herself, this important aspect has been neglected due to the limitations of the methodology used in this manuscript. The authors briefly touch upon this issue in the manuscript, but in my opinion, more in-depth discussion is required.

The manuscript now compares the KS- and PV-waveguides, and indeed, find stronger correlations between QSWs and PV-waveguides, particularly at high latitudes. This comparison is a focus of the revised manuscript, and the differences are discussed in the discussion section: "PV-waveguides show generally stronger positive correlations with QSWs than KS-waveguides, particularly in the higher latitudes, consistent with recent work showing strong teleconnections along high latitude PV-waveguides (Xu et al. 2019; Xu et al. 2020)"

---

## Author Response (AR2)

**Editor**

The revised paper now compares two waveguide diagnostics, and the reviewer is not happy with the discussion and conclusions drawn from the results.

Having gone over the revised paper, I tend to agree with the reviewer on this, that more explicit statements are needed.

Here are some of my thoughts that might help -

there are two main differences between the two diagnostics - one is the zonalization process- a low frequency filtering vs a wave-activity based zonalization of PV contours. The other difference is due to the difference between PV gradients and the stationary wavenumber which is essentially grad(PV)/U. In fact for a zonal mean the relation between the two should be:

Ks^2=-fo*grad(ln(|PV|))/U (Bukenberger et al 2023).

It might help for the comparison to actually calculate this quantity and compare to the standard Ks waveguide, to more directly see any effects of the zonalization method.

As for the direct comparison between using Ks^2 and PV gradients, the two in my mind show different things - Ks^2 shows, for a given zonal wavenumber, the meridional wavenumber that satisfies the dispersion relation, in regions of meridional wave propagation. The relation of the wave amplitude to this is complex. Strong PV gradients indicate regions in which meridional displacements will most directly create strong PV perturbations.

Theoretically I expect both are a proxy of where waves will propagate zonally and of the latitude where we expect most waves to occur at each longitude. The analysis in this paper can more closely and explicitly examine this in real data

Thank you for these comments and suggestions. Overall, we are very grateful to both yourself and all reviewers for comments that have led to substantial improvements to this manuscript. We agree that separating out the impacts of the background flow differences with the impacts of the waveguide definition differences is valuable, and so have added additional analysis on this. To keep the analysis zonally asymmetric, we decided to calculate the PV-waveguides using, instead of the rolling-zonalization method of smoothing, the same time- and zonal-filter that we use for the KS-waveguides. This analysis shows that it is predominantly the waveguide-definition that matters the most, although the background flow is also important, with the rolling-zonalization PV-waveguides showing the strongest correlations with QSWs.

I agree that both methods theoretically tell us something about where we would expect most waves at each longitude, and this is a good phrasing – we have added sentences on this in the introduction, and in the discussion/conclusions.

**Reviewer 1**

The paper has been deeply modified since the first submission. The paper now contains a systematic comparison between two notions of waveguides: one based on the classical stationary wavenumber, which was present in the initial submission, and the other on potential vorticity gradient that has been recently introduced by Wirth and Polster papers and inserted in the revised version of the paper. I think this comparison made the paper much more attractive and I enjoyed reading it. Additionally, all my comments have been carefully considered by the authors. Therefore, I recommend publication of the paper once the following minor suggestions have been taken into account:

1) The waveguide strength is a new diagnostic on which many figures rely on but its definition is not entirely easy to find. Line 165, the maximum waveguide strength $W\_s=K\_s-k$ is introduced. But in order to build composites from Figures 4 to 9, the text says (line 218) that the waveguide strength is the sum of all $K\_s-k$ for k between 4 and 9. Why did you choose the sum of $K\_s-k$ for the composites and not the maximum $K\_s-k$ ? Since all the composites rely on that parameter it would be good to introduce a mathematical notation for that parameter. This would help to clarify the captions of Figures 4 to 9. Different wordings are used to say the same thing: in Figures 4 to 7 "strongest waveguide presence" is used, in Figure 8 "strongest average waveguide strength", Figures B1-B5 "strongest ... average waveguides". Please keep the same wording or use a specific notation.

Thank you for this comment. The results are not sensitive to using the maximum $W\_s$ versus using the sum/average $W\_s$ across wavenumbers (shown below for a sample of the plots). Because waves can exist in a range of wavenumbers, and not necessarily in the wavenumber coinciding with the maximum strength, we choose to show the sum. We have now referred to this as total $W_S$, and noted that the results are not sensitive to this choice/definition. Captions have been updated to be consistent throughout.

[Figure]

Zonal wind composites for strong waveguide days for two regions (black boxes) based on total $W_S$ (left column) and maximum $W_S$ (right column)

2) Line 227: "in the appendix B"
Corrected, thank you.

3) Line 270: I do not understand the end of the sentence "however, these results ....
background flow"
This has now been reworded to clarify: "It is possible that the higher latitude blocking is
creating local KS-waveguide conditions - blocks are known to impact the jet, and thus the
subsequent movement of smaller, more transient eddies (e.g. Shutts 1983). These results
highlight that the KS-waveguide methodology is unable to effectively separate the blocking
perturbation from the background flow, despite the blocks typically occurring on length scales
smaller than k=2, the upper bound of the spatial Fourier filter.

4) There are two groups of sentences in the discussion (lines 418-422) and in the conclusion
(lines 434-438) saying roughly the same thing. I would suggest to suppress one of the two.
These sentences have been substantially changed given the addition of the PV-waveguides
using the Fourier-transform background flow, and we have removed the repetition.

**Reviewer 2**

We thank the reviewer for his continued critique of this paper, and whilst we may not agree on
all points, we know that the revised paper is substantially better because of his reviews, and we
appreciate the time he has spent on this. Since he chose to not be anonymous, we have added
a statement to the acknowledgements to acknowledge his contribution.

Major comments:

A key result of the revised manuscript is the fact that the two different waveguide definitions
are associated with rather strong differences in their climatological/ statistical properties. As I
said, that's interesting and worth a publication, given the fact that the Ks-waveguide has been
used extensively in the past. However, I disagree with the author's interpretation saying that "it
is unclear which [of the two] is the more 'accurate' description [of reality]"; such or related
statements appears several times in the text (occasionally somewhat indirectly). First of all, the
Ks-definition is based on assumptions that are badly violated in reality (Wirth 2020). It does,
therefore, not come as a surprise that the waveguidability diagnosed with the Ks-algorithm is
unable to explain properties of true waveguidability (Wirth, 2020). Moreover, Wirth and Polster
(2021) have shown that the Ks-method is prone to artefacts in the presence of large-amplitude
waves. In fact, the current work corroborates the existence of such artifacts by showing that
certain "double jet features" co-occur with blocking-like flow patterns, and the abstract of this
manuscript says that the "Ks-methodology…. does not sufficiently separate waves from the
background". Given this body of earlier work and the authors own statement in the abstract, a
more logical interpretation of the substantial differences between the two methods would be
saying that the current work corroborates earlier indications that the Ks-definition may be
inappropriate to diagnose waveguides.

To test the reviewers hypothesis, we have now completed additional analysis looking at the impacts of the background flow methodology, with conclusions that, in most cases, it is the KS-vs PV- definition, and not the method of calculating the background flow, that dominates in the differences between the KS-waveguides and the rolling-zonalization PV-waveguides. We therefore now agree with the reviewer that this works corroborates earlier indications that the KS-definition may be inappropriate to diagnose waveguides. This is reflected in several changes throughout the manuscript, examples given below:

"We recommend PV-waveguides using rolling-zonalized background flow for the study of zonally varying waveguides and their connections to waves. This study adds further caution against using KS-waveguides on time- and/or zonally-varying scales."

"Given the two waveguide definitions have large differences in the relative climatological waveguide frequency between different hemispheres and seasons, it is clear that the two waveguide methods cannot both be accurately quantifying the waveguidability of the atmosphere. Existing concerns over the KS-waveguide methodology suggest that the PV-waveguide climatology is likely a more accurate description of the atmosphere's waveguidability; in the following sub-sections we further explore the differences in the waveguide definitions."

"Given the large differences between the two waveguide datasets, the concerns around separation of waves from the waveguides for the KS-waveguides, and the stronger positive correlations between QSWs and PV-waveguides, we recommend that PV-waveguides on rolling-zonalized flow \citep{polster_new_2023} are used for detecting time-varying zonally asymmetric waveguides, particularly for studying the conditions conducive for quasi-stationary waves and related extreme weather events."

My second issue is related to the first issue. It concerns the authors "explanation" ("this can be understood…..") of the differences between the two waveguide detection algorithms, most notably the fact that the definition of Ks implies a division by the wind speed U, while the PV-gradient method does not. Obviously, the division by U explains some of the differences between the two statistics on an algorithmic level. However, the current text conveys the impression that both methods are equally valid, suggesting that the jury is out which of the two methods is better suited (see above, my first major issue). More explicitly, in her reply the author says: "We do not consider this [i.e., the differences in the seasonal and hemispheric behavior between the two waveguide definitions] necessarily to be a weakness of the Ks-waveguides, however, as the results are consistent with the mathematics that the zonal wind strength appears in the denominator of the Ks equation". I disagree, because this argument lacks logic. Of course, the differences in the statistics can be traced back to differences in the algorithms, but this exercise does not make any statement about the appropriateness of an algorithm to represent reality. For the reasons given above, I think that the Ks-algorithm is not a good waveguide diagnostic. A proof that an algorithm does what is it supposed to do cannot save it from being inappropriate.

I accept that the reviewer has already come to a conclusion that the KS-waveguides are, under all circumstances, not valid; however, I do not agree that the scientific community has agreed that KS-waveguides are not valid under any circumstances. I agree that the reviewer has shown in his earlier work that there are certain situations in which the background flow used for the KS-waveguides includes aspects of the waves, but, to my understanding, it has not been proven that these situations dominate, and therefore the KS-waveguide method should be considered invalid at all time. I therefore think the differences between the KS-waveguides and the PV-waveguides are interesting, and worthy of explanation. The end conclusion of the paper is indeed that the KS-waveguides seem less valid and less useful than the PV-waveguides, and this is made clear in the abstract and conclusions. The new results show that most of the differences between the PV-waveguides and KS-waveguides actually come from the different waveguide definitions, not the different methods of calculating the background flow, and thus I continue to believe that explaining how and why these two methods are different is useful to readers. We have strengthened statements about the inappropriateness of using the KS-waveguides (see response to above comment). But I continue to think that understanding the differences, rather than completely dismissing the KS-waveguides from the beginning, is useful.

Minor comments:

Line 5: what is an "objective algorithm"?
Fair comment – the word objective has been removed.

Lines 6,7: given that the concept of "waveguidability" is more appropriate than a binary decision between "waveguide" or "no waveguide", such a number ("40% of all days") is meaningless, because it sensitively depends on the threshold chosen in the definition of a waveguide.
Agreed – this sentence has been removed.

Line 34: does this argument apply only to quasi-stationary waves, or does it also apply to travelling waves?
Unsure – I suspect wave dissipation may play a larger role for quasi-stationary waves remaining at a strong amplitude than for travelling waves passing through at a strong amplitude? Perhaps we should discuss!

Line 44, "there are two main methods….": this formulation suggests that both methods are well established and equally valid. I would argue that this is not the case: the Ks-method was shown in earlier papers to be fraught by a number of issues (see my first major issue)
Reworded to better highlight the concerns that have been raised about the KS-waveguides.

Line 61, "without this clear separation….": I would argue that without a scale separation, the entire concept of a waveguide breaks down, which is much worse than what is suggested by the simple phrase "care must be taken".

The new analysis shows that it is not the method of background flow separation that determines most of the differences between the two methods, but actually the waveguide definitions themselves. From my perspective a lack of 'clear' separation does not equal no separation, and thus the phrase 'much care' is reasonable.

Line 180, "however": this sentence lacks logic (see my second major issue). The word "however" suggests that the argument that follows solves the riddle that was given in the previous sentence. The previous sentence refers to the fact that summer has generally weaker jets than winter, and that according to previous work weaker jets are weaker waveguides. The argument given in the following sentence CANNOT resolve this issue, as it only shows why the Ks theory yields a stronger waveguide in summer than in winter; one might just as well conclude that for this reason Ks-theory is inappropriate, because it contradicts the previous wisdom that weaker jets are weaker waveguides.

I argue that the KS-theory states that narrower jets are stronger waveguides, and that this was shown by Manola et al. (2013), and thus the however is reasonable: the basic idea that stronger jets = stronger waveguides is not necessarily true as the width of the jet also needs to be taken into account. However (!), I have reworded this paragraph now to remove this clause. The paragraph is simply trying to explain the results seen, not argue that therefore the KS-waveguide method is OK. I think it is still worthwhile to understand why a method shows the results that it does, particularly when they are different from other methods. I have added a short paragraph at the end of the waveguide frequency sub-section saying:
"Given the two waveguide definitions have large differences in the relative climatological waveguide frequency between different hemispheres and seasons, it it clear that the two waveguide methods cannot both be accurately quantifying the waveguidability of the atmosphere. Existing concerns over the KS-waveguide methodology suggest that the PV-waveguide climatology is potentially a more accurate description of the atmosphere's waveguidability; in the following sub-sections we further explore the differences in the waveguide definitions.."

Line 183, these idealized tests: I do not quite understand the result of these tests, because they contradict the following simple asymptotic argument. If you increase the strength of the jet and leave the width of the jet constant, I expect that Ks saturates to some finite value, but it should not decrease. Basically, in this limit one can neglect the first term on the right-hand side of (2), and Uyy divided by U essentially yields a meridional wavenumber that characterizes the meridional width of the jet.

Agreed, but I don't think we can make the assumption that we are acting in the limit of being able to neglect the first term in (2), i.e. the planetary vorticity – and in that case, increasing U, whilst keeping the jet half width the same, does indeed decrease K_S. This has been now specified: "Idealised tests of calculating $K_S$ on linear multiples of different realistic $U$ profiles confirms that, unless the planetary vorticity (first term on the right hand side of Eq. \ref{eq:betaM}) is negligible, then if $U$ increases in strength but the latitudinal shape of the $U$ profile (i.e. the jet half-width) remains constant, $K_S$ decreases in magnitude."

Line 213: I suggest to change "much more" to "more"
Changed.

Line 214, "U in the denominator...": again, this only makes plausible that algorithmically the differences are due to the U in the denominator, but it does not make any statement about which of the two options is more realistic (see my second major issue). Similarly on line 345 and 409, "…. This can be understood…": yes, algorithmically the difference can be "understood", but the more important question would be which of the two definitions is more realistic. You simply write that "further study is required"; however, I would argue that the Ks-definition has already accumulated a significant amount of criticism, including the fact that the underlying assumptions are badly violated (see my first major issue). For that reason, one could conclude from these differences that the Ks-definition is less realistic. In your text you make the reader believe that the jury is out (see also lines 356, 357, or on line 411 "it is unclear which is the more 'accurate' description…."), but in my eyes is really isn't.

As noted above, I still think it is helpful to understand why they are different, and so most of this text remains. To address the reviewers comment about which is more realistic, we have now added the short paragraph:
"Given the two waveguide definitions have large differences in the relative climatological waveguide frequency between different hemispheres and seasons, it it clear that the two waveguide methods cannot both be accurately quantifying the waveguidability of the atmosphere. Existing concerns over the KS-waveguide methodology suggest that the PV-waveguide climatology is potentially a more accurate description of the atmosphere's waveguidability; in the following sub-sections we further explore the differences in the waveguide definitions.."
Following the analysis on the background flow impacts, showing that much of the differences comes from the waveguide definition, and the stronger QSW correlations for the PV-waveguides, we have also removed the comments about it being unclear which is more realistic, and hopefully the text is now clearer that the PV-waveguides are preferred:
"This study adds further caution against using KS-waveguides on time- and/or zonally-varying scales, and we recommend using rolling-zonalized PV-waveguides for the study of time- and zonally-varying waveguides and their connections to quasi-stationary atmospheric waves."
"The results in this paper highlight some strong differences between temporally and zonally varying KS-waveguides and PV-waveguides. We can understand many of these differences, such as seasonal and hemispheric variations, at least partially by the different formulations of the waveguide definition, with the strength of the zonal wind $U$ appearing in the denominator of KS-waveguides, but not for PV-waveguides. \cite{wolf_quasistationary_2018} show that QSWs are stronger in NH winter than in NH summer, in line with the seasonal cycle of PV-waveguides. Given the stronger correlations found with QSWs for the PV-waveguides, we conclude that the PV-waveguides likely give a more accurate description of the seasonal cycle of waveguidability."
"Given the large differences between the two waveguide datasets, the concerns around separation of waves from the waveguides for the KS-waveguides, and the stronger positive

correlations between QSWs and PV-waveguides, we recommend that PV-waveguides on rolling-zonalized flow \citep{polster_new_2023} are used for detecting time-varying zonally asymmetric waveguides, particularly for studying the conditions conducive for quasi-stationary waves and related extreme weather events."

Line 237, "…. Enhanced zonal wind inside the waveguide region": I cannot verify this statement on panels 4c, g, i
Text edited to note that it is not true for all regions for the KS-waveguides, highlighting panels c and g (I would argue it is true to some extent in panel i).

Line 248: could you point to those panels in Fig 4 where this "double jet feature" is clearly visible?!
Added direct references to panels a (North America) and e (Asia), where it is most clear. Removed the reference for Europe, as it is weaker (although clearer when the statistical significance masking is removed).

Line 303: suggest to change "much stronger" ◊ "stronger"
Amended as suggested.

Line 432,433: I disagree. I think here one can dare a statement about causality. One important advantage of the rolling zonalization approach is the fact that the so-obtained background state is (almost) independent of the waves that may be present on the background state (see Wirth and Polster 2021). For this reason, one may at least hypothesize that the strength of the waveguide has an impact on the strength of the waves, but not vice versa!

Agreed, although this study has not proven causality, which is what I meant to imply. Given that the positive correlations also exist (albeit slightly weaker) for the PV-waveguides with the zonal and time filtering, I am hesitant to claim causality any stronger than is done in the previous part of the sentence. However, I agree that it is unlikely in the PV-waveguide definition that causality is in the opposite direction. This has therefore been reworded to: "suggesting that strong PV-waveguide conditions make amplified QSWs more likely, although causality has not been established here"

**Reviewer 3**
The authors have significantly revised the manuscript by adding a new discussion on the waveguide defined using the gradient of PV, in addition to the original definition based on the gradient of absolute vorticity (i.e., refractive index). This addition offers a more comprehensive exploration of the waveguide and provides new insights into understanding the behavior of waveguides variations from different perspectives. The authors have fully addressed my previous concerns, and thus I recommend accepting the current version of the manuscript

We thank the reviewer for their time and comments that helped improve the manuscript.

---

## Author Response (AR3)

Thanks again to the editor and reviewer for their helpful comments, we are grateful to them for their time spent helping improve this manuscript. Please see our responses below.

Editor comments

Specifically, the comment about lines 6 and 86/87 - I agree with the reviewer that these sentences are confusing, and have a feeling that the main message is that the rolling zonalization requires PV on theta surfaces, which requires a further high-level diagnostic of calculating the PV on theta surfaces. This can add a lot of numerical noise if there is only daily data on pressure surfaces.

The reason for not using rolling-zonalization for the KS waveguides is as Volkmar noted, that inverting PV to get the zonal winds required for the Ks waveguides is a complex task. However, your point is also valid. The text has been reworded, as noted below for the specific sentences.

Abstract sentence - the bit after the ; "however, this can only be performed on PV, and so the KS-waveguides use time- and zonal- filtering" is not clear, especially if read before reading the paper itself. Also, why "and so"?

Thank you for this comment. This has now been reworded as follows:

We compare waveguides from potential vorticity (PV) gradients (`PV-waveguides') with barotropic waveguides based on what is known as the stationary wavenumber, or $K_S$ (`KS-waveguides'), which is calculated from the zonal wind. The PV-waveguides use a PV-rolling-zonalization method to separate the waves from the background flow. The background flow for the KS-waveguides is calculated using time- and zonal- filtering.

Also lines 86-87: "This method requires PV, however, and cannot be applied directly to the zonal wind data required for the KS-waveguides." is confusing. You can calculate PV from the zonal flow, no? is it indeed a matter of not having enough layers in the daily output to capture the contribution of the vertical derivatives of the flow to PV, so that the numerical errors are too large?

Yes, we could calculate PV from the zonal flow, and then zonalize it, but the issue is (as Volkmar noted) that inverting PV to get the zonal winds required for the Ks waveguides is a complex task. We have now made this clearer in the text:

To avoid this issue, Polster and Wirth (2023) develop a method of calculating a `locally-zonalized' flow for detection of the PV-waveguides on isentropic surfaces, an extension of the zonally symmetric zonalization method (Nakamura and Zhu, 2010; Nakamura and Solomon, 2011; Methven and Berrisford, 2015}. This method cannot be applied directly to the zonal wind data required for the KS-waveguides, and inverting the zonalized PV to produce (zonalized) zonal winds would be a complex task. Given that daily zonal wind on upper tropospheric pressure levels are available directly from many CMIP6 climate model simulations, KS-waveguides on time- and zonally-filtered zonal winds are easier to calculate for future climates than PV-waveguides. The interpolation of this daily pressure level data onto the isentropic surfaces required for the PV-waveguides could potentially introduce significant numerical noise.

Other comments I have on top of the ones raised by the reviewer:
The rolling zonalization is not quite a smoothing (c.f. line 127). I would maybe phrase:
"Zonalization is a method of straightening out the wavy PV contours, essentially zonally smoothing.."
A good point, thanks – text changed as suggested.

The diagnostic description is missing details about the levels used for the calculation. Is it 300mb U for Ks? What theta surface is used in figures 2-3? I would also explicitly note that the climatological U is different between Ks and Pv based subplots in these figures because of the differences between isentropic and pressure level U (assuming this is the cause for the difference).
These details were given at the beginning of section 2 (300hPa for Ks, 330K for PV in winter, 345K for PV in summer). The zonal wind contours in the PV panels are still at 300hPa (as noted above, inverting the PV to get the zonal winds is not done in this study. These details have now been added to the captions of Figs 2 and 3.

Reviewer comments

Abstract line 6 and main text lines 86/87: The text reads as if the technique of rolling zonalization precludes the computation of Ks-waveguides. This is not really true. Assuming a balance condition, the knowledge of PV allows one to compute the wind through so-called PV inversion. Of course, PV inversion in the framework of the primitive equations is cumbersome (see e.g., Nakamura and Solomon 2011), but at least in principle it would be possible. Having said this, I do NOT suggest that this should be done here.

Agreed – I have reworded the text to make this clearer:

Abstract: We compare waveguides from potential vorticity (PV) gradients (`PV-waveguides') with barotropic waveguides based on what is known as the stationary wavenumber, or $K\_S$ (`KS-waveguides'), which is calculated from the zonal wind. The PV-waveguides use a PV-rolling-zonalization method to separate the waves from the background flow. The background flow for the KS-waveguides is calculated using time- and zonal- filtering.

Line 86/87: To avoid this issue, Polster and Wirth (2023) develop a method of calculating a `locally-zonalized' flow for detection of the PV-waveguides on isentropic surfaces, an extension of the zonally symmetric zonalization method (Nakamura and Zhu, 2010; Nakamura and Solomon, 2011; Methven and Berrisford, 2015}. The zonalization method cannot be applied directly to the zonal wind data required for the KS-waveguides. Calculating KS-waveguides on zonalized background flow would therefore require inverting zonalized PV to produce zonalized zonal winds; this PV inversion on daily data would be a complex task. Given that daily zonal wind on upper tropospheric pressure levels are available directly from many CMIP6 climate

model simulations, KS-waveguides on time- and zonally-filtered zonal winds are easier to calculate for future climates than PV-waveguides, or KS-waveguides on zonalized data. The interpolation of daily pressure level data onto the isentropic surfaces required for the PV-waveguides could potentially introduce significant numerical noise, adding further complexity to the PV-waveguide calculation for CMIP data.

Figure caption Fig 2: Units of \nabla ln PV must be m-1
Good point, thanks – corrected.
Line 185, the definition of w_s: this seems somewhat unclear, since K_s is a function of latitude. Are you considering the maximum value of K_s-k over the waveguide, i.e., the range of latitudes where K_s>k?
This has been reworded to hopefully make it clearer that it is indeed the maximum value of Ks-k over the range of latitudes within the waveguide:
the maximum waveguide strength, $w_S = \max(K_S - k)$ across all latitudes within the waveguide

Line 257: the formula given for W_s is somewhat ambiguous: do you mean the sum of (K_s-k) or do you mean the (sum of K_s) minus k?
Added parentheses to clarify:

Line 283, what is the "second meridional gradient"? Do you mean the second derivative of U in the meridional direction?
Yes – text clarified.

Line 348: what is an "in-season seasonal cycle"?
We meant differences between June and August, for example, distinct from the larger differences between December and July, however it seems perhaps that is confusing. Changed to just seasonal cycle.